# DexTrack: Towards Generalizable Neural Tracking Control for Dexterous Manipulation from Human References

**Xueyi Liu**[1,2], **Jianibieke Adalibieke**[2], **Qianwei Han**[2], **Yuzhe Qin**[4], **Li Yi**[1,3,2]
[1]Tsinghua University  [2]Shanghai Qi Zhi Institute  [3]Shanghai AI Laboratory  [4]UC San Diego
`Project website:` meowuu7.github.io/DexTrack

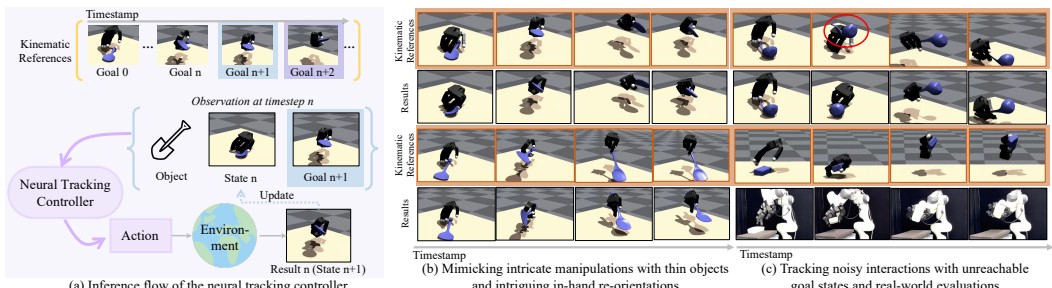

(a) Inference flow of the neural tracking controller
(b) Mimicking intricate manipulations with thin objects and intriguing in-hand re-orientations
(c) Tracking noisy interactions with unreachable goal states and real-world evaluations

Figure 1: DexTrack learns a generalizable neural tracking controller for dexterous manipulation from human references. It generates hand action commands from kinematic references, ensuring close tracking of input trajectories (*Fig. (a)*), generalizes to novel and challenging tasks involving thin objects, complex movements and intricate in-hand manipulations (*Fig. (b)*), and demonstrates robustness to large kinematics noise and utility in real-world scenarios (*Fig. (c)*). Kinematic references are illustrated in orange rectangles and background.

## Abstract

We address the challenge of developing a generalizable neural tracking controller for dexterous manipulation from human references. This controller aims to manage a dexterous robot hand to manipulate diverse objects for various purposes defined by kinematic human-object interactions. Developing such a controller is complicated by the intricate contact dynamics of dexterous manipulation and the need for adaptivity, generalizability, and robustness. Current reinforcement learning and trajectory optimization methods often fall short due to their dependence on task-specific rewards or precise system models. We introduce an approach that curates large-scale successful robot tracking demonstrations, comprising pairs of human references and robot actions, to train a neural controller. Utilizing a data flywheel, we iteratively enhance the controller's performance, as well as the number and quality of successful tracking demonstrations. We exploit available tracking demonstrations and carefully integrate reinforcement learning and imitation learning to boost the controller's performance in dynamic environments. At the same time, to obtain high-quality tracking demonstrations, we individually optimize per-trajectory tracking by leveraging the learned tracking controller in a homotopy optimization method. The homotopy optimization, mimicking chain-of-thought, aids in solving challenging trajectory tracking problems to increase demonstration diversity. We showcase our success by training a generalizable neural controller and evaluating it in both simulation and real world. Our method achieves over a 10% improvement in success rates compared to leading baselines. The project website with animated results is available at DexTrack.

## 1 Introduction

Robotic dexterous manipulation refers to the ability of a robot hand skillfully handling and manipulating objects for various target states with precision and adaptability. This capability has attracted

significant attention because adept object manipulation for goals such as tool use is vital for robots to interact with the world. Many efforts have been devoted previously to push the ability of a dexterous hand toward human-level dexterity and versatility (Rajeswaran et al., 2017; Chen et al., 2023; 2021; Akkaya et al., 2019; Christen et al., 2022; Zhang et al., 2023; Qin et al., 2022; Liu et al., 2022; Wu et al., 2023; Gupta et al., 2016; Wang et al., 2023; Mordatch et al., 2012; Liu et al., 2024a; Li et al., 2024). This also aligns with our vision.

Achieving human-level robotic dexterous manipulation is challenging due to two main difficulties: the intricate dynamics of contact-rich manipulation, which complicates optimization (Pang & Tedrake, 2021; Pang et al., 2023; Liu et al., 2024a; Jin, 2024), and the need for robots to master a wide range of versatile skills beyond specific tasks. Previous approaches mainly resort to model-free reinforcement learning (RL) (Chen et al., 2023; 2021; Akkaya et al., 2019; Christen et al., 2022; Zhang et al., 2023; Qin et al., 2022; Liu et al., 2022; Wu et al., 2023; Gupta et al., 2016; Wang et al., 2023) or model-based trajectory optimization (TO) (Pang & Tedrake, 2021; Pang et al., 2023; Jin, 2024; Hwangbo et al., 2018). While RL requires task-specific reward designs, limiting its generalization, TO depends on accurate dynamics models with known contact states, restricting adaptability to new objects and skills. A promising alternative is to leverage human hand-object manipulation references, widely available through videos or motion synthesis, and focus on controlling a dexterous hand to track these references. This approach separates high-level task planning from low-level control, framing diverse skill acquisition as the development of a universal tracking controller. However, challenges remain due to noisy kinematic references, differences in morphology between human and robotic hands, complex dynamics with rich contacts, and diverse object geometry and skills. Existing methods struggle with these issues, often limiting themselves to simple tasks without in-hand manipulation (Christen et al., 2022; Zhang et al., 2023; Wu et al., 2023; Xu et al., 2023; Luo et al., 2024; Singh et al., 2024; Chen et al., 2024) or certain specific skills (Qin et al., 2022; Liu et al., 2024a; Rajeswaran et al., 2017).

In this work, we aim to develop a general-purpose tracking controller that can follow hand-object manipulation references across various skills and diverse objects. In particular, given a collection of kinematics-only human hand-object manipulation trajectories, the controller is optimized to drive a robotic dexterous hand to manipulate the object so that the resulting hand and object trajectories can closely mimic their corresponding kinematic sequences. We expect the tracking controller to exhibit strong versatility, generalize well to precisely track novel manipulations, and have strong robustness towards large kinematics noises and unexpected reference states.

To achieve the challenging goal above, we draw three key observations: 1) learning is crucial for handling heterogeneous reference motion noises and transferring data prior to new scenarios, supporting robust and generalizable tracking control; 2) leveraging large-scale, high-quality robot tracking demonstrations that pair kinematic references with tracking action sequences can supervise and significantly empower neural controllers, as demonstrated by data-scaling laws in computer vision and NLP (OpenAI, 2023; Brown et al., 2020); 3) acquiring large and high-quality tracking demonstrations is challenging but we could utilize the data flywheel (Chiang et al., 2024; Bai et al., 2023) to iteratively improve the tracking controller and expand the demonstrations in a bootstrapping manner.

Based upon the previous observations, we propose DexTrack, a novel neural tracking controller for dexterous manipulation, guided by human references. Specifically, given a collection of human hand-object manipulation trajectories, we first retarget the collection to kinematic robotic dexterous hand sequences to form a set of reference motions as data preparation. Our method then alternates between mining successful robot tracking demonstrations and training the controller with the mined demonstrations. To make sure the data flywheel functions effectively, we introduce two key designs. First, we carefully integrate reinforcement and imitation learning techniques to train a neural controller, ensuring its performance improves with more demonstrations while maintaining robustness against unexpected states and noise. Second, we develop a per-trajectory tracking scheme that uses the trained controller to mine diverse and high-quality tracking demonstrations through a homotopy optimization method. The scheme transfers the tracking prior from the controller to individual trajectories to ease per-trajectory tracking for better demonstration quality. Moreover, the scheme will convert a tracking reference into a series of gradually simplified reference motions so that tracking these references from simple to complex could help better track the original reference motion. This is akin to chain-of-thought and is very suitable for tracking complex reference motions to increase the demonstration diversity. The two designs above together with the iterative training enable DexTrack to successfully track novel and challenging human references.

We demonstrate the superiority of our method and compare it with previous methods on challenging manipulation tracking tasks in two datasets, describing expressive hand-object interactions in daily and functional tool-using scenarios, involving complex object movements, difficult and subtle in-hand re-orientations, interactions with thin objects, and frequent hand-object rich contact variations. We conduct both extensive experiments in the simulator, *i.e.,* Isaac Gym (Makoviychuk et al., 2021), and evaluations in the real world, to demonstrate the efficacy, generalization ability, and robustness of our tracker to accomplish a wide range of manipulation tracking tasks and even excellently track novel manipulation trajectories (Figure 1). Our approach successfully surpasses the previous methods both quantitatively and qualitatively, achieving more than 10% success rate than the previous best-performed method. Besides, we conduct further analysis and demonstrate the various recovery behaviors of our controller, demonstrating its robustness to unexpected situations. Thorough ablations are conducted to validate the efficacy of our designs.

Our contributions are threefold:

- We present a generalizable neural tracking controller that progressively improves its performance through iterative mining and incorporating high-quality tracking demonstrations.
- We introduce a training method that synergistically combines reinforcement learning and imitation learning. This approach leverages abundant high-quality robot tracking demonstrations to produce a controller that is generalizable, versatile, and robust.
- We develop a per-trajectory optimization scheme that employs our tracking controller within a homotopy optimization framework. We propose a data-driven way to generate homotopy paths, enabling solving challenging tracking problems.

## 2 RELATED WORK

Equipping robots with human-level dexterous manipulation skills is crucial for future advancements. Previous approaches either rely on model-based trajectory optimization or model-free reinforcement learning (RL). Model-based methods face challenges due to the complexity of dynamics, often requiring approximations (Pang et al., 2023; Jin, 2024; Pang & Tedrake, 2021). Model-free approaches, using RL (Rajeswaran et al., 2017; Chen et al., 2023; 2021; Christen et al., 2022; Zhang et al., 2023; Qin et al., 2022; Liu et al., 2022; Wu et al., 2023; Gupta et al., 2016; Wang et al., 2023; Mordatch et al., 2012), focus on goal-driven tasks with task-specific rewards, limiting their generalization across diverse tasks. Our work explores a general controller for dexterous manipulations. Besides, learning via mimicking kinematic trajectories has recently become a popular train to equip the agent with various expressive skills (Jenelten et al., 2023; Luo et al., 2023b;a). DTC (Jenelten et al., 2023) proposes a strategy that can combine the power of model-based motion planning and RL to overcome the sample inefficiency of RL. In the humanoid motion tracking space, PHC (Luo et al., 2023b) proposes an effective RL-based training strategy to develop a general humanoid motion tracker. Recently, OmniGrasp (Luo et al., 2024) proposes to train a universal grasping and trajectory following policy. The policy can generalize to unseen objects as well as track novel motions. However, their considered motions are still restricted in grasping and trajectory following, leaving the problem of tracking more interesting and difficult trajectories such as those with subtle in-hand manipulations largely not explored. In this paper, we focus on these difficult and challenging manipulations. Moreover, we are also related to recent trials on combining RL with imitation learning. To overcome the sample inefficiency problem of RL and to facilitate the convergence, various approaches have been developed aiming to augment RL training with demonstrations (Sun et al., 2018; Hester et al., 2017; Booher et al., 2024; Liu et al., 2023). In our work, we wish to leverage high-quality demonstrations to guide the agent's explorations. Unlike previous work where demonstrations are readily available, acquiring a sufficient amount of high-quality robot tracking demonstrations remains a significant challenge in our task.

## 3 METHOD

**Terminologies and notations.** Dexterous manipulation "tracking" involves controlling a robotic hand to mimic a kinematic hand-object state sequence, the goal trajectory, denoted as $\{\hat{\mathbf{s}}_n\}_{n=0}^{N}$. These "kinematic references" are retargeted from human manipulation trajectories, with $\hat{\mathbf{s}}_n$ representing the robot hand state and object pose at timestep $n$. A "tracking demonstration" pairs a kinematic reference $\{\hat{\mathbf{s}}_n\}$ with an expert action sequence $\{\mathbf{a}_n^L\}$, guiding the robot from $\mathbf{s}_0 = \hat{\mathbf{s}}_0$ to

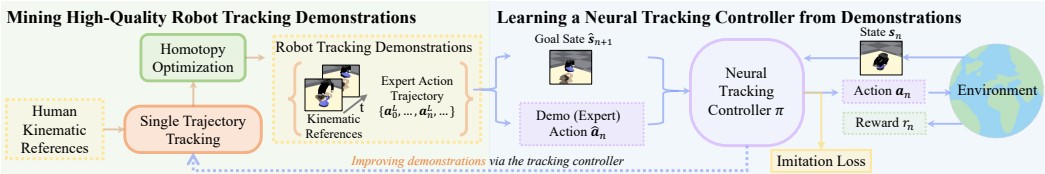

Figure 2: **DexTrack** learns a generalizable neural tracking controller for dexterous manipulation from human references. It alternates between training the tracking controller using abundant and high-quality robot tracking demonstrations and improving the data via the tracking controller through a homotopy optimization scheme.

achieve a state sequence $\{\mathbf{s}_n\}_{n=0}^N$ aligned with $\{\hat{\mathbf{s}}_n\}_{n=0}^N$. A "robust" controller can tolerate disturbances like kinematics noise and unreachable states. It demonstrates high "generalization ability" if it performs well on unseen scenarios such as new objects and novel motions, and has "adaptivity" if it maintains effectiveness despite shifting contexts, such as changing contacts and dynamics.

Given a collection of kinematic human-object manipulation trajectories, we wish to learn a generalizable neural tracking controller for a dexterous robotic hand. The problem is challenging due to the difficulty in precise dexterous manipulation, which is challenged by the underlying complex dynamics, and the high demands for the controller's generalizability and robustness.

We address these challenges by combining reinforcement learning (RL) and imitation learning (IL) to train a generalizable tracking controller, jointly easing the difficulty in solving the complex problem via supervision from high-quality and diverse robot tracking demonstrations and improving the policy's robustness via RL explorations. We introduce a single trajectory tracking scheme to mine tracking demonstrations, composed of paired kinematic references and action sequences. For each kinematic reference, we use RL to train a trajectory-specific policy that generates actions to track the reference. To overcome RL's limitations, we propose to leverage the tracking controller through a homotopy optimization scheme to enhance the demonstration quality and diversity. By iteratively mining better demonstrations and refining the controller in a bootstrapping manner, we develop an effective, generalizable tracking controller. We will explain how we learn the neural tracking controller from demonstrations in Section 3.1, how we mine high-quality demonstrations in Section 3.2, and how we iterate between learning controller and mining demonstrations in Section 3.3.

### 3.1 LEARNING A NEURAL TRACKING CONTROLLER FROM DEMONSTRATIONS

Given a collection of human-object manipulation trajectories and a set of high-quality robot tracking demonstrations, our goal is to learn an effective and generalizable neural tracking controller. In the beginning, we retarget human hand-object manipulation to a robotic hand as a data preprocessing step. We combine RL and IL to develop a generalizable and robust neural tracking controller. Taking advantage of imitating diverse and high-quality robot tracking demonstrations, we can effectively let the tracking controller master diverse manipulation skills and also equip it with high generalization ability. Jointly leveraging the power of RL, the controller avoids overfitting to successful tracking results limited to a narrow distribution, thereby maintaining robust performance in the face of dynamic state disturbances. Specifically, we design an RL-based learning scheme for training the tracking controller, including the carefully-designed action space, observations, and the reward tailored for the manipulation tracking task. We also introduce an IL-based strategy that lets the tracking controller benefit from imitating high-quality demonstration data. By integrating these two approaches, we effectively address the complex problem of generalizable tracking control.

**Neural tracking controller.** In our formulation, the neural tracking controller acts as an agent that interacts with the environment according to a tracking policy $\pi$. At each timestep $n$, the policy observes the observation $\mathbf{o}_n$ and the next goal $\hat{\mathbf{s}}_{n+1}$ (designated as the target state for the robotic hand and the object). The policy then computes the distribution of the action. Then the agent sample an action $\mathbf{a}_n$ from the policy, *i.e.,* $\mathbf{a}_n \sim \pi(\cdot|\mathbf{o}_n, \hat{\mathbf{s}}_{n+1})$. The observation $\mathbf{o}_n$ primarily contains the current state $\mathbf{s}_n$ and the object geometry. Upon executing the action with the robotic dexterous hand, the hand physically interacts with the object, leading both the hand and the object to transition into the next state according to the environment dynamics, represented as $\mathbf{s}_{n+1} \sim p(\cdot|\mathbf{s}_n, \mathbf{a}_n)$. An effective tracking controller should ensure that the resulting hand and object states closely align with their respective next goal states.

**Reinforcement learning.** In the RL-based training scheme, the agent receives a reward $r_n = r(\mathbf{s}_n, \mathbf{a}_n, \hat{\mathbf{s}}_{n+1}, \mathbf{s}_{n+1})$ after each transition. The training objective is to maximize the discounted

cumulative reward:

$$J = \mathbb{E}_{p(\tau|\pi)} \left[ \sum_{n=0}^{N-1} \gamma^n r_n \right], \tag{1}$$

where $p(\tau|\pi) = p(\mathbf{s}_0) \prod_{n=0}^{N-1} p(\mathbf{s}_{n+1}|\mathbf{o}_n, \mathbf{a}_n) \pi(\mathbf{a}_n|\mathbf{s}_n, \hat{\mathbf{s}}_{n+1})$ is the likelihood of a transition trajectory of the agent $\tau = (\mathbf{s}_0, \mathbf{a}_0, r_0, ..., \mathbf{s}_{N-1}, \mathbf{a}_{N-1}, r_{N-1}, \mathbf{s}_N)$. The discount factor $\gamma \in [0, 1)$ determines the effective horizon length of the policy. In the tracking control problem, the next goal state $\hat{\mathbf{s}}_{n+1}$ typically comprises the subsequent hand state and the object state from the kinematic reference sequence. We control the robotic hand using a proportional derivative (PD) controller, following previous literature (Luo et al., 2024; 2023b; Christen et al., 2022; Zhang et al., 2023). The action $\mathbf{a}_n$ contains the target position commands for all hand joints. To enhance the sample efficiency of RL, rather than allowing the tracking policy to learn the absolute positional targets, we introduce a residual action space. Specifically, we introduce a baseline hand trajectory and train the policy to learn the residual relative target $\Delta \mathbf{a}_n$ at each timestep. The baseline trajectory is consistently available for the tracking problem and can be trivially set to the kinematic reference trajectory. In each timestep $n$, we compute the position target via $\mathbf{a}_n = \mathbf{s}_n^b + \sum_{k=0}^{n} \Delta \mathbf{a}_k$, where $\mathbf{s}_n^b$ is the $n$-th step hand state in the baseline trajectory. The observation at each timestep $n$ encodes the current hand and object state, the baseline trajectory, actions, velocities, and the object geometry:

$$\mathbf{o}_n = \{\mathbf{s}_n, \dot{\mathbf{s}}_n, \mathbf{s}_n^b, \mathbf{a}_n, \text{feat}_{\text{obj}}, \text{aux}_n\}, \tag{2}$$

where $\text{feat}_{\text{obj}}$ is the object feature generated by a pre-trained object point cloud encoder. We also introduce an auxiliary feature $\text{aux}_n$, computed based on available states, to provide the agent with more informative context. Further details will be explained in Appendix A. Our reward for manipulation tracking encourages the transited hand state and the object state to closely match their respective reference states, as well as promoting hand-object affinity:

$$r = w_{o,p} r_{o,p} + w_{o,q} r_{o,q} + w_{\text{wrist}} r_{\text{wrist}} + w_{\text{finger}} r_{\text{finger}} + w_{\text{affinity}} r_{\text{affinity}}, \tag{3}$$

where $r_{o,p}, r_{o,q}, r_{\text{wrist}}, r_{\text{finger}}, r_{\text{affinity}}$ represent rewards for object position, object orientation, hand wrist, hand fingers, and hand-object affinity, respectively, while $w_{o,p}, w_{o,q}, w_{\text{wrist}}, w_{\text{finger}}, w_{\text{affinity}}$ are their corresponding weights. Details regarding the reward computation are deferred to Appendix A.

**Imitation learning.** The RL-based learning scheme, hindered by sample inefficiency and its inability to handle multiple tracking problems, struggles to solve the generalizable tracking control problem, as demonstrated by our early experiments. Therefore, we propose an IL-based strategy to distill successful, abundant, and diverse "tracking knowledge" into the tracking controller. Specifically, we train the tracking agent to imitate a large number of high-quality robot tracking demonstrations. This approach effectively guides the agent to produce "expert actions" that can successfully track the reference state. Additionally, by imitating diverse tracking demonstrations, the agent can avoid repeatedly encountering low rewards in challenging tracking scenarios, while also preventing over-exploitation of easier tasks. Formally, a robot tracking demonstration of length $N$ consists of a kinematic reference sequence $(\hat{\mathbf{s}}_0, ..., \hat{\mathbf{s}}_N)$ and an expert's state-action trajectory $(\mathbf{s}_0^L, \mathbf{a}_0^L, ..., \mathbf{s}_{N-1}^L, \mathbf{a}_{N-1}^L, \mathbf{s}_N^L)$. In addition to the actor loss, we incorporate an action supervision loss to bias the policy's predictions at each timestep towards the corresponding expert action in the demonstration:

$$\mathcal{L}_a = \mathbb{E}_{\mathbf{a}_n \sim \pi(\cdot|\mathbf{o}_n, \hat{\mathbf{s}}_{n+1})} \|\mathbf{a}_n - \mathbf{a}_n^L\|. \tag{4}$$

This guidance allows the policy's exploration to be informed by these demonstrations, ultimately speeding up convergence and enhancing performance in complex problems. From the perspective of IL, the RL exploration introduces noise into the states, making the imitation more robust, in a similar flavor to DART (Laskey et al., 2017). Since the agent should not and will not explore states too far from the reference states in the tracking control task, it is feasible to optimize the policy using both the imitation loss and RL reward simultaneously.

### 3.2 MINING HIGH-QUALITY ROBOT TRACKING DEMONSTRATIONS USING NEURAL CONTROLLER THROUGH A HOMOTOPY OPTIMIZATION SCHEME

To prepare a demonstration from a kinematic reference trajectory $(\hat{\mathbf{s}}_0, ..., \hat{\mathbf{s}}_N)$ for training the tracking controller, we need to infer the action sequence $(\mathbf{a}_0^L, ..., \mathbf{a}_{N-1}^L)$ that successfully tracks the reference sequence. A straightforward approach is to leverage RL to train a single-trajectory tracking

policy $\pi$ and use its resulting action sequence directly. However, relying solely on this strategy often fails to provide a diverse and high-quality dataset of tracking demonstrations, as RL struggles with the inherent challenges of dexterous manipulation. To enhance both the diversity and quality of robot tracking demonstrations, we propose utilizing the tracking controller in conjunction with a homotopy optimization scheme to improve the per-trajectory tracking results.

**RL-based single trajectory tracking.** A basic approach to acquiring demonstrations is to leverage RL to address the per-trajectory tracking problem, using the resulting action sequence as the demonstration. Given a kinematic reference trajectory, our goal is to optimize a tracking policy $\pi$ that can accurately track this reference trajectory. At each timestep, $\pi$ observes the current state $\mathbf{s}_n$ and the next goal state $\hat{\mathbf{s}}_{n+1}$, and it predicts the distribution of the current action $\mathbf{a}_n$. The policy $\pi$ is optimized to minimize the difference between the transited state $\mathbf{s}_{n+1} \sim p(\cdot|\mathbf{s}_n, \mathbf{a}_n)$ at each timestep $n$ and the goal state $\hat{\mathbf{s}}_{n+1}$. Similar to our design for the RL-based learning scheme for the tracking controller, we adopt a residual action space. For each tracking task, we introduce a baseline trajectory $(\mathbf{s}_0^b, ..., \mathbf{s}_N^b)$ and only learn a residual policy using RL. The observations and rewards follow the same design as for the tracking controller (see Eq. 8 and Eq. 10). Once $\pi$ has been optimized, we can sample the action $\mathbf{a}_n$ from $\pi$ at each timestep. By iteratively transitioning to the next state using the predicted action and querying the policy $\pi$ to generate a new action, we can obtain the expert action sequence $(\mathbf{a}_0^L, ..., \mathbf{a}_{N-1}^L)$ for the input kinematic reference trajectory.

**Transferring "tracking prior".** To improve the demonstrations, we devise a strategy that takes advantage of the tracking controller, which has already encoded "knowledge" that can track lots of trajectories (named as "tracking prior"), to improve trajectory-specific tracking policies. Specifically, to track a reference trajectory $(\hat{\mathbf{s}}_0, ..., \hat{\mathbf{s}}_N)$, we first utilize the tracking controller to track it, with the baseline trajectory set to thew reference sequence. We then adjust the baseline trajectory to the resulting action sequence and re-optimize the residual policy using the RL-based single trajectory tracking method. This approach can help us find a better baseline trajectory, facilitating single-trajectory tracking policy learning and improving the per-trajectory tracking results.

**A homotopy optimization scheme.** Training the controller with data mined from itself can introduce biases and reduce diversity, hindering its generalization capabilities. To address this issue, we propose a homotopy optimization scheme to improve the per-trajectory tracking performance and to tackle previously unsolvable single-trajectory tracking problems. For the tracking problem $T_0$, instead of solving it directly, the homotopy optimization iteratively solves each tracking task in an optimization path, *e.g.*, $(T_K, T_{K-1}, ..., T_0)$, and finally solves $T_0$, in a similar flavor to "chain-of-thought" (Wei et al., 2022). In the beginning, we leverage the RL-based tracking method to solve $T_K$. After that, we solve each task $T_m$ via the RL-based tracking method with the baseline trajectory set to the tracking result of $T_{m+1}$. This transfer of tracking results from other tasks helps us establish better baseline trajectories, ultimately yielding higher-quality tracking outcomes.

**Finding effective homotopy optimization paths.** While leveraging the homotopy optimization scheme to solve previously unsolvable problems has proven effective for many tasks, it depends on identifying effective optimization paths. A straightforward approach is brute-force searching. Specifically, given a set of kinematic references, we first optimize their per-trajectory tracking outcomes. We then identify neighbors for each task based on the similarity between pairs of kinematic reference trajectories. Next, we iteratively transfer optimization results from neighboring tasks and re-optimize the residual policy for each tracking task. For a specific task, we consider a neighbor that provides a better baseline trajectory than its kinematic trajectory as an effective "parent task". After reaching the maximum number of iterations, $K$, we can find effective homotopy paths for a task, *i.e.*, $T_0$, by starting from $T_0$ and backtracing effective "parent tasks". We define $(T_K, ..., T_0)$ as an effective homotopy path for $T_0$ if, for each $0 \leq m \leq K$, $T_{m+1}$ is an effective parent task for $T_m$.

**Learning a homotopy generator for efficient homotopy path planning.** Finding effective homotopy optimization paths for tracking each trajectory is computationally expensive and impractical during inference. To address this, we propose learning a homotopy path generator $\mathcal{M}$ from a small dataset, enabling efficient generation of effective homotopy paths for other tracking tasks. The key problem in identifying homotopy paths lies in finding effective "parent tasks". We reformulate this problem as a tracking task transformation problem, aiming for a generator $\mathcal{M}$ that provides a distribution of effective "parent tasks" for each tracking task $T_0$: $\mathcal{M}(\cdot|T_0)$, considering the fact that a single tracking task may have multiple effective "parent tasks". Once $\mathcal{M}$ is trained, we can find a homotopy path by iteratively finding parent tasks. We propose training a conditional diffusion model

as the tracking task transformer, leveraging its strong distribution modeling capability. Given a set of tracking tasks, characterized by the kinematic reference trajectories of the hand and object, as well as object geometry, we first train a diffusion model to capture the distribution of tracking tasks. To fine-tune this diffusion model into a conditional model, we first search for effective homotopy paths within the tracking task dataset. This yields a set of paired data of the tracking task $T_c$ and their corresponding effective "parent task" $T_p$. We then use this data to tune the diffusion model into a conditional diffusion model, such that $T_p \sim \mathcal{M}(\cdot|T_c)$. A well-trained $\mathcal{M}$ can efficiently propose an effective homotopy path for a tracking task, *i.e.*, $T_0$, by starting from $T_0$ and recursively finding the parent task, resulting in $(T_K, ..., T_0)$, where $T_{m+1} \sim \mathcal{M}(\cdot|T_m)$ for all $0 \leq m \leq K - 1$.

## 3.3 Improving the Tracking Controller via Iterative Optimization

We adopt an iterative approach that alternates between training the tracking controller with abundant robot tracking demonstrations and curating more diverse, higher-quality demonstrations using the controller. Our method is divided into three stages. In the first stage, we sample a small set of tracking tasks and generate an initial demonstration set by applying RL to obtain single-trajectory tracking results for each task. Using these demonstrations, we train the tracking controller with both RL and IL. At this stage, we do not train the homotopy path generator, as the model's generalization ability would be limited by the small number of effective homotopy paths available for training. In the second stage, we sample a dataset of trajectories from the remaining tasks, weighted according to the controller's tracking error. We then use RL, incorporating tracking priors, to optimize per-trajectory tracking, search for homotopy paths, and train a homotopy path generator based on the resulting data. The best tracking results from all successfully tracked trajectories are curated into a new demonstration set, which is used to re-train the tracking controller. In the third stage, we resample tracking tasks from the remaining set and leverage RL, the tracking controller, and the homotopy generator to curate another set of tracking demonstrations. This final set is used to optimize the tracking controller, resulting in our final model.

## 4 Experiments

We conduct extensive experiments to evaluate the effectiveness, generalizability, and robustness of our tracking controller. Tested on two HOI datasets featuring complex daily manipulation tasks, our method is assessed through both simulation and real-world evaluations (see Sec. 4.1). We compare our approach to strong baselines, showing its superiority. Our controller successfully handles novel manipulations, including intricate movements, thin objects, and dynamic contacts (see Sec. 4.2), while previous methods fail to generalize well. On average, our method improves the tracking success rate by over 10% compared to the best prior methods. Additionally, we analyze its robustness to significant kinematic noise, such as unrealistic states and large penetrations (see Sec. 4.3).

## 4.1 Experimental Settings

**Datasets.** We test our method on two public human-object interaction datasets: GRAB (Taheri et al., 2020), featuring daily interactions, and TACO (Liu et al., 2024b), containing functional tool-use interactions. In simulation, we use the Allegro hand, with URDF adapted from Isaac-GymEnvs (Makoviychuk et al., 2021), and in real-world experiments, the LEAP hand (Shaw et al., 2023), due to hardware constraints. Human-object interaction trajectories are retargeted to create robot hand-object sequences using PyTorch_Kinematics (Zhong et al., 2024). We fully retargeted the GRAB and TACO datasets, producing 1,269 and 2,316 robot hand manipulation sequences, respectively. The evaluation on GRAB focus on testing the model's generalization to unseen interaction sequences. Specifically, we use sequences from subject `s1` (197 sequences) as the test data while the remaining trajectories as the training set. For the TACO dataset, we follow the generalization evaluating setting suggested by the authors (Liu et al., 2024b) and split the dataset into a training set with 1,565 trajectories and four distinct test sets at different difficulty levels. The primary quantitative results are reported on the first-level set. More details are provided in Appendix C.

**Metrics.** We introduce five metrics to evaluate the tracking accuracy and task success: 1) Per-frame average object rotation error: $R_{\text{err}} = \frac{1}{N+1} \sum_{n=0}^{N} \text{Diff\_Angle}(\mathbf{q}_n, \hat{\mathbf{q}}_n)$, where $\hat{\mathbf{q}}_n$ and $\mathbf{q}_n$ are reference and tracked orientation. 2) Per-frame average object translation error: $T_{\text{err}} = \frac{1}{N+1} \sum_{n=0}^{N} \|\mathbf{t}_n - \hat{\mathbf{t}}_n\|$, where $\mathbf{t}_n$ and $\hat{\mathbf{t}}_n$ are the tracked and reference translations. 3) Per-frame average wrist position and rotation error: $E_{\text{wrist}} = \frac{1}{N+1} \sum_{n=0}^{N} \left( 0.5 \, \text{Diff\_Angle}(\mathbf{q}_n^{\text{wrist}}, \hat{\mathbf{q}}_n^{\text{wrist}}) + 0.5 \|\mathbf{t}_n^{\text{wrist}} - \hat{\mathbf{t}}_n^{\text{wrist}}\| \right)$, where $\mathbf{q}_n^{\text{wrist}}$ and $\hat{\mathbf{q}}_n^{\text{wrist}}$ are wrist orientations, and $\mathbf{t}_n^{\text{wrist}}$ and $\hat{\mathbf{t}}_n^{\text{wrist}}$ are translations. 4) Per-frame per-joint

Table 1: **Quantitative evaluations.** **Bold red** and *italic blue* values for best and the second best-performed ones respectively. "Ours (w/o) data" and "Ours (w/o data, w/o homotopy)" are two ablated versions w.r.t. quality of robot tracking demonstrations used in imitation learning (see Section 5 for details).

| Dataset | Method | $R_{\text{err}}$ (rad,↓) | $T_{\text{err}}$ (cm,↓) | $E_{\text{wrist}}$ (↓) | $E_{\text{finger}}$ (rad,↓) | Success Rate (%,↑) |
|---|---|---|---|---|---|---|
| GRAB | DGrasp | 0.4493 | 6.75 | 0.1372 | 0.6039 | 34.52/52.79 |
| | PPO (OmniGrasp rew.) | 0.4404 | 6.69 | 0.1722 | 0.6418 | 35.53/54.82 |
| | PPO (w/o sup., tracking rew.) | 0.3945 | 6.11 | **0.1076** | 0.5899 | 38.58/54.82 |
| | Ours (w/o data, w/o homotopy) | 0.3443 | 7.81 | 0.1225 | *0.5218* | 39.59/57.87 |
| | Ours (w/o data) | *0.3415* | *4.97* | 0.1483 | 0.5264 | *43.15/62.44* |
| | Ours | **0.3303** | **4.53** | *0.1118* | **0.5048** | **46.70/65.48** |
| TACO | DGrasp | 0.5021 | 5.04 | **0.1129** | 0.4737 | 38.42/47.78 |
| | PPO (OmniGrasp rew.) | 0.5174 | 5.43 | 0.1279 | 0.4945 | 33.5/46.31 |
| | PPO (w/o sup., tracking rew.) | *0.4815* | 4.82 | *0.1195* | *0.4682* | 34.98/57.64 |
| | Ours (w/o data, w/o homotopy) | **0.4444** | 2.33 | 0.1782 | 0.5438 | 44.83/67.00 |
| | Ours (w/o data) | 0.4854 | *2.21* | 0.1698 | 0.4772 | *47.78/72.41* |
| | Ours | 0.4953 | **2.10** | 0.1510 | **0.4661** | **48.77/74.38** |

average position error: $E_{\text{finger}} = \frac{1}{N+1}\sum_{n=0}^{N}\left(\frac{1}{d}\|\theta_n^{\text{finger}} - \hat{\theta}_n^{\text{finger}}\|_1\right)$, where $\theta$ denotes finger joint positions, $d$ is the degrees of freedom. 5) Success rate: A tracking attempt is successful if $T_{\text{err}}$, $R_{\text{err}}$, and $0.5E_{\text{wrist}} + 0.5E_{\text{finger}}$ are all below the thresholds. Success is calculated with two thresholds: 10cm-20°-0.8 and 10cm-40°-1.2.

**Baselines.** To our knowledge, no prior model-based methods have directly tackled tracking control for dexterous manipulation. Most existing approaches focus on single goal-driven trajectory optimization with simplified dynamics models (Jin, 2024; Pang et al., 2023; Pang & Tedrake, 2021), limiting their adaptability for our framework's generalizable tracking controller. Thus, we primarily compare our method with model-free approaches: 1) DGrasp (Christen et al., 2022): Adapted to track by dividing sequences into subsequences of 10 frames, with each subsequence solved incrementally. 2) PPO (OmniGrasp rew.): We re-implemented OmniGrasp's reward (Luo et al., 2024) to train a policy for tracking object trajectories. 3) PPO (w/o sup., tracking rew.): We trained a policy using PPO with our proposed tracking reward and observation design.

**Training and evaluation settings.** We use PPO (Schulman et al., 2017), implemented in rl_games (Makoviichuk & Makoviychuk, 2021), with Isaac Gym (Makoviychuk et al., 2021) for simulation. Training is performed with 8192 parallel environments for both the per-trajectory tracker and the tracking controller. For the dexterous hand, the position gain and damping coefficient are set to 20 and 1 per finger joint. Evaluation results are averaged across 1000 parallel environments, and for real-world evaluations, we use LEAP (Shaw et al., 2023) with a Franka arm and Foundation-Pose (Wen et al., 2023) for object state estimation. Additional details are provided in Appendix C.

## 4.2 GENERALIZABLE TRACKING CONTROL FOR DEXTEROUS MANIPULATION

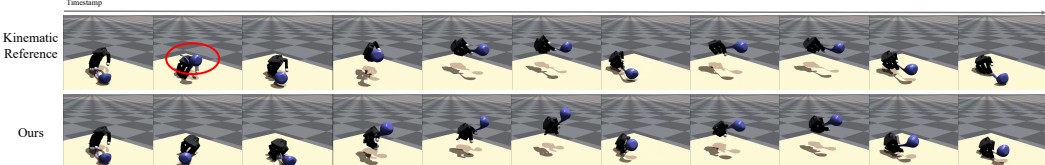

Figure 3: **Robustness w.r.t. unreasonable states.** Please check **our website** and **video** for animated results.

We demonstrate the generalization ability and robustness of our tracking controller on unseen trajectories involving challenging manipulations and novel, thin objects. Our controller has no difficulty in handling intricate motions, subtle in-hand re-orientations, and expressive functional manipulations, even when dealing with thin objects. As shown in Table 1, we achieve significantly higher success rates, calculated under two different thresholds, compared to the best-performing baseline across both datasets. Figure 4 provides qualitative examples and comparisons. We show the real-world effectiveness of our method and the superiority over best-performing baselines (Figure 4,Table 2). For animated results, please visit our project website and the accompanying video.

**Intriguing in-hand manipulations.** Our method effectively generalizes to novel, complex, and challenging functional manipulations, featured by subtle in-hand re-orientations, essential for precise tool-use tasks. For instance, in Figure 4a, the shovel is lifted, tilted, and reoriented with intricate finger movements to complete a stirring motion. Similarly, in Figure 4c, the small shovel is reori-

Table 2: **Real-world quantitative comparisons.** **Bold red** numbers for best values.

| Method | apple | banana | duck | elephant | flashlight | flute | hammer | hand | phone | waterbottle |
|---|---|---|---|---|---|---|---|---|---|---|
| PPO (w/o sup., tracking rew) | 0/0/0 | 25.0/25.0/0.0 | 50.0/25.0/0 | 50.0/0.0/0.0 | 50.0/0/0 | 0/0/0 | 25.0/0/0 | 66.7/33.3/0 | 25.0/0/0 | 33.3/33.3/0 |
| Ours | **25.0/0/0** | **50.0/50.0/25.0** | **75.0/50.0/25.0** | **75.0/50.0/50.0** | **50.0/25.0/25.0** | **25.0/25.0/25.0** | **50.0/50.0/50.0** | **66.7/33.3/33.3** | **50.0/50.0/25.0** | **50.0/33.3/33.3** |

ented using minimal wrist adjustments. These results demonstrate the robustness and generalization of our controller, outperforming the PPO baseline, which struggles with basic lifting.

**Intricate manipulations with thin objects.** Our method also generalizes well to challenging manipulations involving thin objects. In Figure 4b, despite the thin `shovel`'s complexity and missing CAD model parts, our method successfully lifts and controls it using a firm grasp with the second and third fingers. Similarly, in Figure 4e, our controller adeptly lifts and manipulates a thin `flute`, while the best-performing baseline struggles with the initial grasp. These results highlight the advantages of our approach in handling complex and delicate manipulations.

**Real-world evaluations and comparisons.** We directly transfer tracking results to the real world to assess tracking quality and evaluate the robustness of the state-based controller against noise in the state estimator. Success rates are measured under three thresholds and compared with the best baseline. Table 2 summarizes the per-object success rates averaged over their manipulation trajectories in the transferred controller setting. As demonstrated in Figures 4f and 4g, we enable the robot to track complex object movements and successfully lift a hard-to-grasp round apple in real-world scenarios. While the baseline fails. Further details can be found in Appendix B.2.

### 4.3 Further Analysis and Discussions

**Robustness to noise in the kinematic reference motions.** Despite severe hand-object penetrations in Figure 4c (frames 2 and 3) and Figure 4a (frame 2), the hand still interacts effectively with the object, highlighting the resilience of our tracking controller in challenging scenarios.

**Robustness to unreasonable references.** As shown in Figure 3, our method remains unaffected by significant noise in the kinematic references with unreasonable states. We effectively track the entire motion trajectory, demonstrating the controller's robustness in handling unexpected noise.

## 5 Ablation Studies

**Diversity and quality of robot tracking demonstrations.** We propose enhancing the diversity and quality of tracking demonstrations using the tracking controller and homotopy generator. We ablate these strategies by creating two variants: "Ours (w/o data, w/o homotopy)", where the dataset is built by optimizing each trajectory without prior knowledge, and "Ours (w/o data)", which uses only the homotopy optimization scheme to improve demonstrations. Despite using the same number of demonstrations, both variants produce lower-quality data. As shown in Table 1, they underperform compared to our full method, underscoring the importance of data quality in training the controller.

**Scaling the number of demonstrations.** To investigate the relationship between the tracking controller's performance and the number of demonstrations, we vary the size of the demonstration dataset during training and tested performance on the TACO dataset. Specifically, in the final training iteration, we down-sampled the dataset to 0.1, 0.3, 0.5, and 0.9 of its original size and re-trained the model. As shown in Figure 5, there is a clear correlation between the amount of demonstrations and model performance. Since the curve has not plateaued, we hypothesize that increasing the amount of high-quality data could further improve performance.

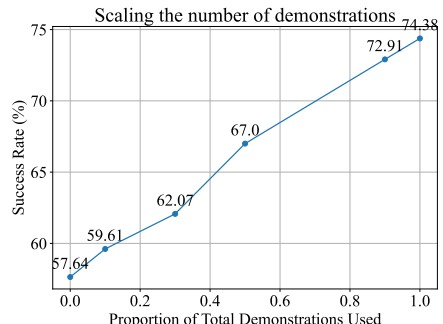

Figure 5: Scaling the amount of demonstrations.

## 6 Conclusions and Limitations

We propose DexTrack to develop a generalizable tracking controller for dexterous manipulation. Leveraging high-quality tracking demonstrations and a pre-trajectory tracking scheme, we refine the controller through bootstrapping. Extensive experiments confirm its effectiveness, establishing a strong foundation for future advancements. **Limitations.** A key limitation is the time-consuming process of acquiring high-quality demonstrations. Future work could explore faster, approximate methods for homotopy optimization to speed up training.

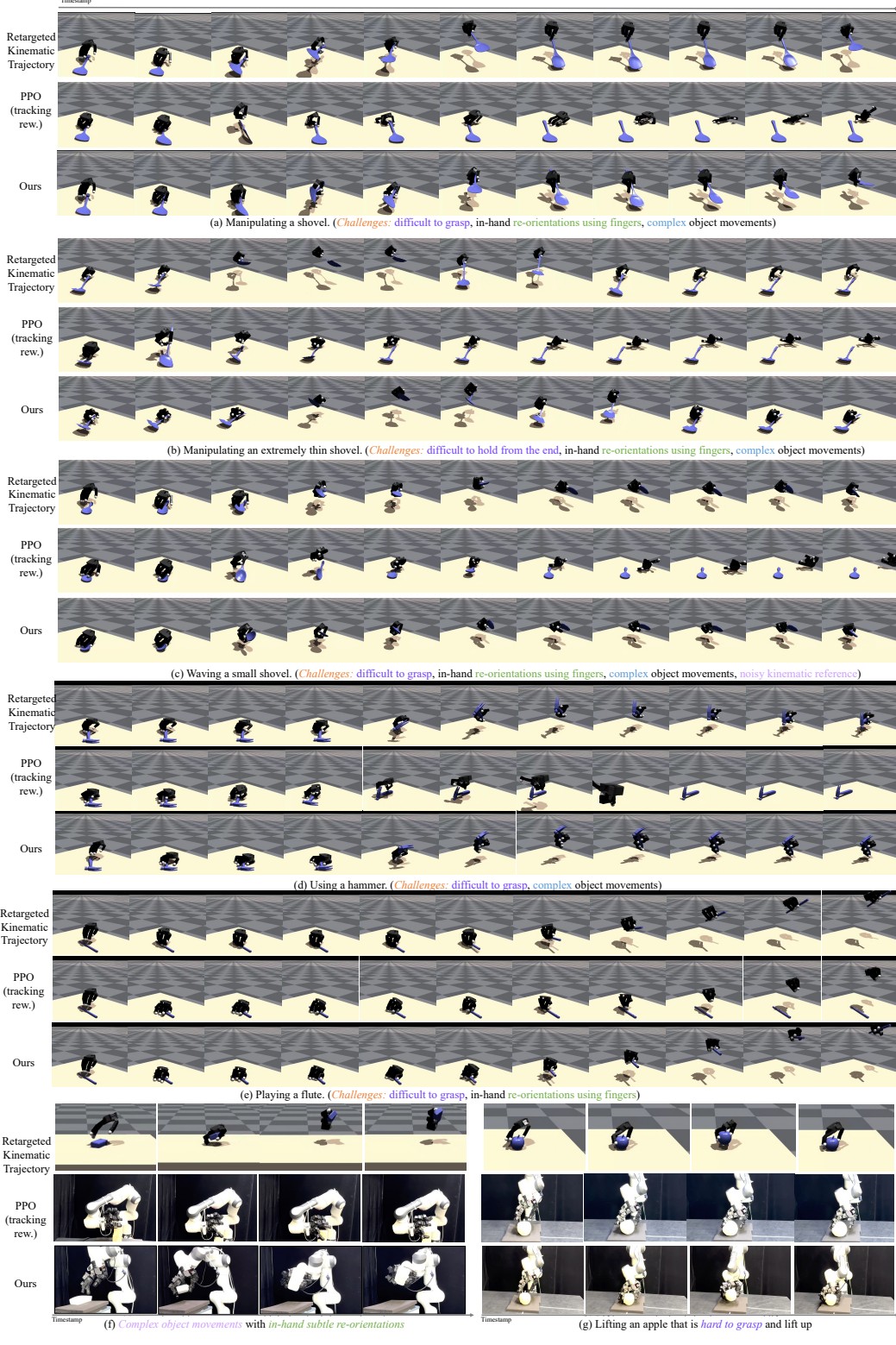

Figure 4: **Qualitative comparisons.** Please check **our website** and **the accompanying video** for animated results.

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

Sheng Zhong, Thomas Power, Ashwin Gupta, and Peter Mitrano. PyTorch Kinematics, February 2024. 7, 15

**Overview.** The **Appendix** provides a list of materials to support the main paper.

- **Additional Technical Explanations (Sec. A).** We give additional explanations to complement the main paper.
    - *Detailed Method Overview Figure*. We include an overview figure for the method (Figure 6) that illustrates the method more detailed than the figure in the method section.
    - *Data Preprocessing (Sec. A.1)*. We present details of the kinematics retargeting strategy we leverage to create a dexterous kinematic robot hand manipulation dataset from human references.
    - *Tracking Controller Training (Sec. A.2)*. We explain additional details in the RL-based training scheme design, including the observation space and the reward. We also explain the control strategy for a floating base dexterous hand.
    - *Homotopy Generator Learning (Sec. A.3)*. We explain details in the homotopy generator learning.
    - *Additional Details (Sec. A.4)*. We present additional details w.r.t. the techniques.
- **Additional Experimental Results (Sec. B).** We include more experimental results in this section to support the effectiveness of the method, including
    - *Dexterous Manipulation Tracking Control (Sec. B.1)*. We present additional experiments of our methods as well as additional comparisons, including results achieved by using different training settings and additional qualitative results. We will also discuss more generalization ability evaluation experiments.
    - *Real-World Evaluations (Sec. B.2)*. We include more results of the real-world evaluations. We will discuss failure cases in the real-world evaluation as well.
    - *Analysis on the Homotopy Optimization Scheme (Sec. B.3)*. We present qualitative results achieved by the proposed homotopy optimization method and comparisons to demonstrate the capability of the homotopy optimization as well as the effectiveness of the homotopy optimization path generator. Additionally, we discuss the generalization ability of the homotopy path generator.
    - *Failure Cases (Sec. B.4)*. We discuss the failure cases for a comprehensive evaluation and understanding w.r.t. the ability of our method.
- **Experimental Details (Sec. C).** We illustrate details of datasets, models, training and evaluation settings, simulation settings, real-world evaluation settings, and the running time as well as the complexity analysis. Additionally, we attempt to ground some key properties of the tracking controller computationally and present quantified evaluations regarding these properties.

We include a **video** and an **website** to introduce our work. The website and the video contain animated results. We highly recommend exploring these resources for an intuitive understanding of the challenges, the effectiveness of our model, and its superiority over prior approaches.

## A   ADDITIONAL TECHNICAL EXPLANATIONS

**Overview figure of the method with details.** In figure 6, to present a comprehensive overview of our method, we draw a detailed overview figure that includes more important details than the figure of the original method in Section 3.

### A.1   DATA PREPROCESSING

**Kinematic retargeting.** We curate kinematic robot-object interaction data from human references by retargeting robot hand manipulation sequences from human hand trajectories. For instance, given a human hand-object interaction trajectory describing the human hand pose sequences represented in MANO and the object pose sequences ($\mathbf{H}^{\text{human}}$), $\mathbf{O}$ as well as the description of the articulated robot hand, we retarget $\mathbf{H}^{\text{human}}$ to acquire the robot hand trajectory $\mathbf{H}$. We manually define correspondences between the robot hand mesh and the MNAO hand mesh. After that, the sequence of the robot hand DoF positions is optimized so that the resulting robot hand mesh sequence is close to the

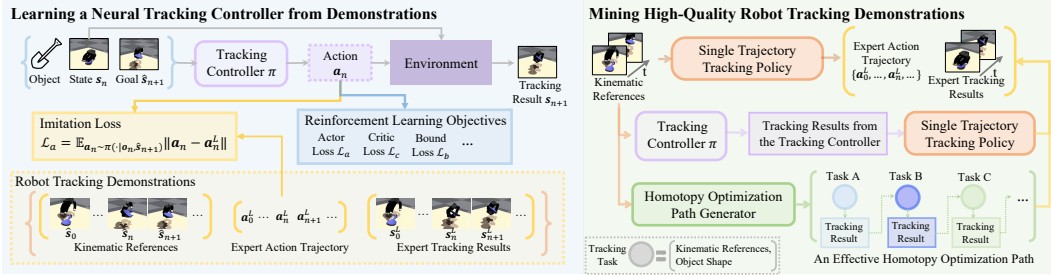

Figure 6: DexTrack learns a generalizable neural tracking controller for dexterous manipulation from human references. It alternates between training the tracking controller using abundant and high-quality robot tracking demonstrations, and improving the data via the tracking controller through a homotopy optimization scheme.

Table 3: Weights of different reward components.

|  | $w_{o,p}$ | $w_{o,q}$ | $w_{\text{wrist}} \cdot w_{\text{trans}}$ | $w_{\text{wrist}} \cdot w_{\text{ornt}}$ | $w_{\text{finger}}$ |
|---|---|---|---|---|---|
| Weight | 1.0 | 0.33 | 0.3 | 0.05 | 0.05 |

human hand sequence. Formally, let $\mathbf{K}^{\text{human}}$ and $\mathbf{K}$ denote the human hand keypoint sequence and the robot hand keypoint sequence respectively, the optimization objective is:

$$\text{minimize} \|\mathbf{K} - \mathbf{K}^{\text{human}}\|. \tag{5}$$

We use PyTorch_Kinematics (Zhong et al., 2024) to calculate the forward kinematics. Specifically, given the robot hand per-joint DoF position $\theta_n$ at the timestep $n$, we calculate $\mathbf{h}_n$ and $\mathbf{k}_n$ as follows:

$$\mathbf{h}_n = \text{Forward\_Kinematics}(\theta_n), \tag{6}$$
$$\mathbf{k}_n = \text{KeyPoints}(\text{Forward\_Kinematics}(\theta_n)), \tag{7}$$

where Forward_Kinematics($\cdot$) computes the forward kinematics using the function provided in Py-Torch_Kinematics, KeyPoints($\cdot$) reads out keypoints from the converted articulated mesh. We use a second-order optimizer, *i.e.,* L-BFGS implemented in PyTorch, to solve the optimization problem 5.

## A.2 TRACKING CONTROLLER TRAINING

**Controlling a floating-base articulated hand.** The articulated hand is represented in the reduced coordinate $\theta^{\text{finger}}$. We additionally add three translation joints and three revolute joints to control the global position and orientation of the hand, resulting in $\theta = (\theta)^{\text{trans}}, \theta^{\text{rot}}, \theta^{\text{rot}})$. For the Allegro hand and the LEAP hand that we use in our experiments, $\theta^{\text{finger}}$ is a 16-dimensional vector. Therefore, $\theta$ is a 22-dimensional vector.

**Observations.** The observation at each timestep $n$ encodes the current hand and object state, the next goal state, baseline trajectory, actions, and the object geometry:

$$\mathbf{o}_n = \{\mathbf{s}_n, \dot{\mathbf{s}}_n, \hat{\mathbf{s}}_{n+1}, \mathbf{s}_n^b, \mathbf{a}_n, \text{feat}_{\text{obj}}, \text{aux}_n\}. \tag{8}$$

where $\text{aux}_n$ is the auxiliary features, computed as follows:

$$\text{aux}_n = \{\hat{\mathbf{s}}_{n+1}, \mathbf{f}_n, \hat{\mathbf{s}}_{n+1} \ominus \mathbf{s}_n, \}, \tag{9}$$

where $\hat{\mathbf{s}}_{n+1} \ominus \mathbf{s}_n$ calculates the difference between two states, including the hand state difference and the object state difference, $\mathbf{f}_n$ indicates the hand finger positions in the world space.

**Reward.** Our reward for manipulating tracking encourages the transited hand state and the object state to be close to their corresponding reference states and the hand-object affinity:

$$r = w_{o,p}r_{o,p} + w_{o,q}r_{o,q} + w_{\text{wrist}}r_{\text{wrist}} + w_{\text{finger}}r_{\text{finger}} + w_{\text{affinity}}r_{\text{affinity}}, \tag{10}$$

Table 4: **Quantitative evaluations and comparisons. Bold red** numbers for best values. Models are trained on training tracking tasks from both the GRAB and the TACO datasets.

| Dataset | Method | $R_{\text{err}}$ (rad, ↓) | $T_{\text{err}}$ ($cm$, ↓) | $E_{\text{wrist}}$ (↓) | $E_{\text{finger}}$ (rad, ↓) | Success Rate (%, ↑) |
|---------|--------|------|------|------|------|------|
| GRAB | PPO (w/o sup., tracking rew.) | 0.5813 | 6.03 | 0.1730 | 0.5439 | 36.04/55.84 |
|  | Ours | **0.4515** | **4.82** | **0.14574** | **0.4574** | **42.64/61.42** |
| TACO | PPO (w/o sup., tracking rew.) | 0.6751 | 6.37 | **0.1264** | 0.5443 | 21.67/50.25 |
|  | Ours | **0.4782** | **3.94** | 0.1329 | **0.4228** | **32.02/62.07** |

where $r_{o,p}, r_{o,q}, r_{\text{wrist}}, r_{\text{finger}}$ are rewards for tracking object position, object orientation, hand wrist, hand fingers, $w_{o,p}, w_{o,q}, w_{\text{wrist}}, w_{\text{finger}}, w_{\text{affinity}}$ are their weights. $r_{o,p}, r_{o,q}, r_{\text{wrist}}, r_{\text{finger}}$ are computed as follows:

$$r_{o,p} = 0.9 - \|\mathbf{p}_n^o - \hat{\mathbf{p}}_n^o\|_2, \tag{11}$$

$$r_{o,q} = \text{np.pi} - \text{Diff\_Angle}(\mathbf{q}_n^o - \hat{\mathbf{q}}_n^o)), \tag{12}$$

$$r_{\text{wrist}} = -(w_{\text{trans}}\|\mathbf{s}_n^h[:3] - \hat{\mathbf{s}}_n^h[:3]\|_1 + w_{\text{ornt}}\|\mathbf{s}_n^h[3:6] - \hat{\mathbf{s}}_n^h[3:6]\|_1 \tag{13}$$

$$r_{\text{finger}} = -w_{\text{finger}}\|\mathbf{s}_n^h[6:] - \hat{\mathbf{s}}_n^h[6:]\|_1 \tag{14}$$

where $\mathbf{p}_n^o$ and $\mathbf{q}_n^o$ denote the position and the orientation, represented in quaternion, of the current object, $\mathbf{s}_n^h$ denotes the current hand state. In addition to these rewards, we would add an additional bonus reward 1 if the object is accurately tracked, *i.e.,* with the rotation error kept in 5-degree and the translation error kept in 5-cm. Table 3 summarizes weights of different reward components we use in our experiments.

**Pre-processing object features.** We train PonintNet-based auto-encoder on all objects from the two datasets we considered, namely GRAB and TACO. After that, we use the latent embedding of each object as its latent feature feeding into the observation of the tracking controller. The object feature dimension is 256 in our experiments.

## A.3 Homotopy Generator Learning

**Mining effective homotopy optimization paths.** The maximum number of iterations $K$ is set to 3 in our method to balance between the time cost and the effectiveness.

We need to identify neighbors for each tracking task so that we can avoid iterating over all tasks and reduce the time cost. We use the cross-kinematic trajectory similarity to filter neighboring tasks. We pre-select $K_{\text{nei}} = 10$ neighbouring tasks for each tracking task.

## A.4 Additional Explanations

In the reward design, we do not include the velocity-related terms since it is impossible for us to get accurate velocities from kinematic references. One can imagine calculating the finite differences between adjacent two frames as the velocities. However, it may not be accurate. Therefore, we do not use them to avoid unnecessary noise.

## B Additional Experiments

### B.1 Dexterous Manipulation Tracking Control

**Training the tracking controller on two datasets.** In the main experiments, the training data and the test data come from the same dataset. We adopt such a setting considering the large cross-dataset trajectory differences. Specifically, GRAB mainly contains manipulation trajectories with daily objects, while TACO mainly covers functional tool-using trajectories. However, jointly using the trajectories from such two datasets to train the model can potentially offer us a stronger controller considering the increased labeled data coverage. Therefore, we additionally conduct this experiment where we train a single model using trajectories provided by the two datasets and test the performance on their test sets respectively. Results are summarized in Table 4, which can still demonstrate the effectiveness of our approach.

Table 5: Generalizability evaluations on the TACO dataset.

| Test set | $R_{err}$ (rad, ↓) | $T_{err}$ ($cm$, ↓) | $E_{wrist}$ (↓) | $E_{finger}$ (rad, ↓) | Success Rate (%, ↑) |
|---|---|---|---|---|---|
| S1 | 0.5787 | 2.43 | 0.1481 | 0.4703 | 35.97/67.63 |
| S2 | 0.6026 | 2.46 | 0.1455 | 0.4709 | 30.83/65.00 |
| S3 | 0.6508 | 8.06 | 0.1513 | 0.4683 | 10.18/46.32 |

Table 6: Performance comparisons across models trained with different amount of demonstration data.

| Proportion | $R_{err}$ (rad, ↓) | $T_{err}$ ($cm$, ↓) | $E_{wrist}$ (↓) | $E_{finger}$ (rad, ↓) | Success Rate (%, ↑) |
|---|---|---|---|---|---|
| 0.0 | 0.4985 | 4.42 | 0.1435 | 0.4767 | 31.03/57.64 |
| 0.1 | **0.4730** | 3.86 | 0.1502 | 0.4921 | 36.45/59.61 |
| 0.3 | 0.4903 | 2.94 | **0.1256** | 0.4804 | 40.89/62.07 |
| 0.5 | 0.4749 | 2.51 | 0.1680 | 0.4682 | 41.38/67.00 |
| 0.9 | 0.4776 | 2.29 | 0.1437 | 0.4839 | 44.83/72.91 |
| 1.0 | 0.4953 | **2.10** | 0.1510 | **0.4661** | **48.77/74.38** |

**Comparisons with labeling the whole dataset for training the controller.** In our method we only try to label a small fraction of data in a progressive way leveraging the power of the tracking prior provided by the tracking controller and the homotopy paths proposed by the homotopy generator so that we can get high-quality demonstrations within an affordable time budget. One may wonder whether it is possible to label all the training dataset trajectories and train the tracking controller using those demonstrations. And if possible, what's the performance of the model trained via this approach? We therefore manage to label each trajectory in the training dataset of the GRAB dataset by running the per-trajectory tracking experiments in parallel in 16 GPUs using two machines. We complete the optimization within one week. After that, we use the resulting demonstrations to train the trajectory controller. The final model trained in this way achieves 42.13% and 60.41% success rates under two thresholds. The performance can still not reach our original method where we only try to optimize high-quality demonstrations for part of the data (see Table 1 for details). We suppose it is the quality difference between the two labeled datasets that causes the discrepancy. This further validates our assumption that both the quantity and the quality of the labeled dataset matter to train a good tracking controller.

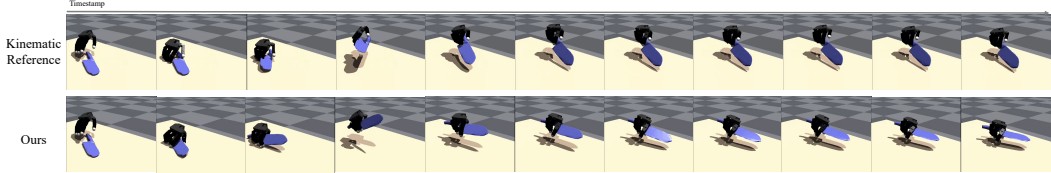

Figure 7: **Robustness towards out-of-distribution objects and manipulations.** Please refer to **our website** and **the accompanying video** for animated results.

**Further generalization ability evaluations on TACO dataset.** We further evaluate the model's generalization ability across various test sets within the TACO dataset. As shown in Table 5, the controller performs well in the category-level generalization setting (S1), where object categories are known but manipulation trajectories and object geometries are novel. Performance on S2, involving novel interaction triplets, is satisfactory, demonstrating the controller's capacity to handle new manipulation sequences. However, results from S3 reveal challenges when dealing with new object categories and unseen interaction triplets. For instance, generalizing from interactions with shovels and spoons to using bowls for holding objects is particularly difficult. As shown in Figure 7, despite unfamiliar objects and interactions, we successfully lift the knife and mimic the motion, though execution is imperfect, highlighting areas for improvement and adaptability in challenging scenarios.

**Additional results.** We present additional qualitatively results in Figure 8 to further demonstrate the capability of our method.

**Scaling the number of demonstration data.** In Table 6, we present the full evaluation results on all five types of metrics of each model traiend in the ablation study regarding the influence of the amount of demonstration data on model's performance (see section 5 for details).

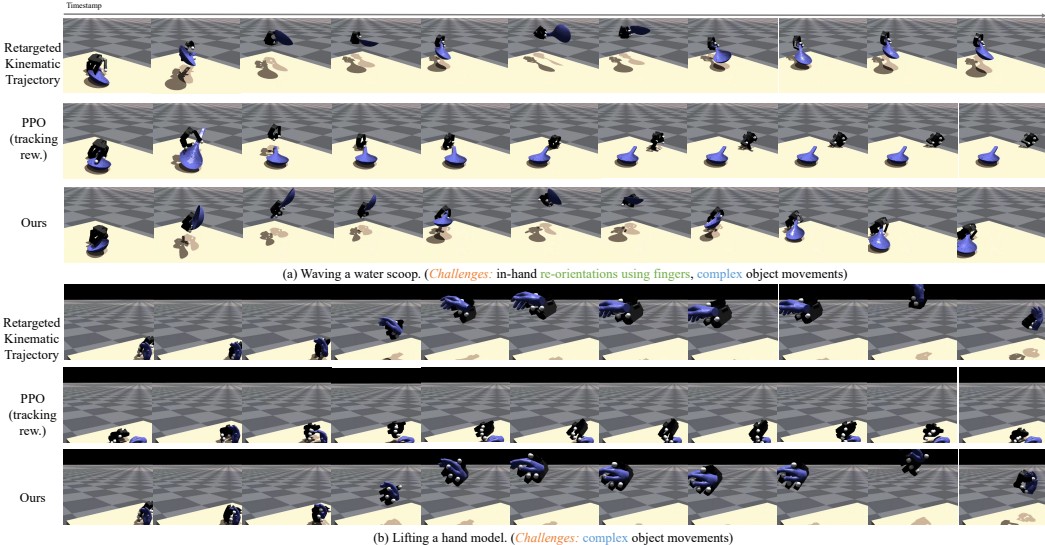

(a) Waving a water scoop. (*Challenges:* in-hand re-orientations using fingers, complex object movements)

(b) Lifting a hand model. (*Challenges:* complex object movements)

Figure 8: **Additional qualitative comparisons.** Please refer to **our website** and **the accompanying video** for animated results.

Table 7: **Real-world quantitative comparisons (GRAB dataset). Bold red** numbers for best values.

| Method | apple | banana | duck | elephant | flashlight | flute | hammer | hand | phone | waterbottle |
|---|---|---|---|---|---|---|---|---|---|---|
| PPO (w/o sup., tracking rew) | 0/0/0 | 25.0/25.0/0.0 | 50.0/50.0/0.0 | 25.0/0.0/0.0 | 50.0/25.0/0 | 0/0/0 | 25.0/25.0/0 | 33.3/33.3/0 | 25.0/25.0/0 | 33.3/0/0 |
| Ours | **50.0/50.0/25.0** | **50.0/50.0/25.0** | **75.0/50.0/50.0** | **50.0/50.0/50.0** | **75.0/50.0/25.0** | **25.0/25.0/0.0** | **50.0/25.0/25.0** | **66.7/66.7/66.7** | **25.0/25.0/25.0** | **50.0/50.0/50.0** |

## B.2 REAL-WORLD EVALUATIONS

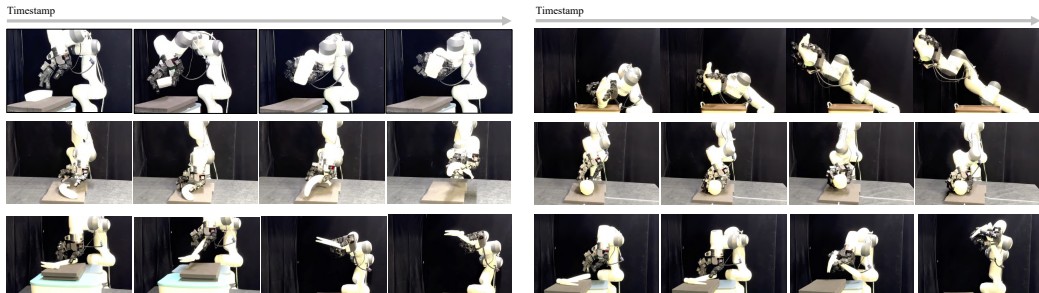

Figure 9: **Additional real-world qualitative results.** Please refer to **our website** and **the accompanying video** for animated results.

**Success thresholds.** We define three levels of success rates. The first level of success is defined as reaching the object, finding a good grasp pose, and exhibiting the potential movements to lift the object up, *i.e.,* one side of the object is successfully lifted up from the table. The second level of success is defined as finding a way to manage to lift the whole object up from the table. The third level of success is lifting the object up, followed by keeping tracking the object's trajectory for more than 100 timesteps.

**More results.** For direct tracking results transferring setting, we present the quantitative success rates evaluated on our method and the best-performed baseline in Table 7 (for the dataset GRAB) and Table 8 (for the dataset TACO). As observed in the table, the tracking results achieved by our method can be well transferred to the real-world robot, helping us achieve obviously better results than the baseline methods. It validates the real-world applicability of our tracking results.

Please refer to the main text (Sec. 4) for the quantitative comparisons between the transferred controllers. We include more qualitative results in Figure 9 to demonstrate the real-world application value of our method.

Table 8: **Real-world quantitative comparisons (TACO dataset).** **Bold red** numbers for best values.

| Method | soap | shovel | brush | roller | knife | spoon |
|---|---|---|---|---|---|---|
| PPO (w/o sup., tracking rew) | 33.3/0/0 | 25.0/0.0/0.0 | 25.0/0/0 | 25.0/25.0/0.0 | 0/0/0 | 25.0/0/0 |
| Ours | **100.0/66.7/66.7** | **50.0/25.0/25.0** | **25.0/25.0/0.0** | **50.0/25.0/25.0** | **25.0/25.0/0.0** | **50.0/50.0/25.0** |

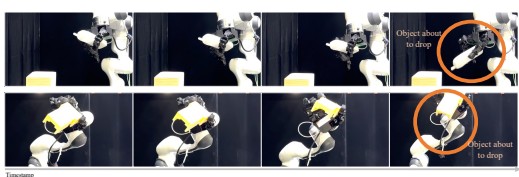

Figure 10: **Failure cases in real-world experiments.** Please refer to **our website** for animated results.

**Failure cases.** A typical fail mode is that the object tends to drop from the hand as contact varies when attempting the in-hand manipulations, as shown in Figure 10.

## B.3 ANALYSIS ON THE HOMOTOPY OPTIMIZATION SCHEME

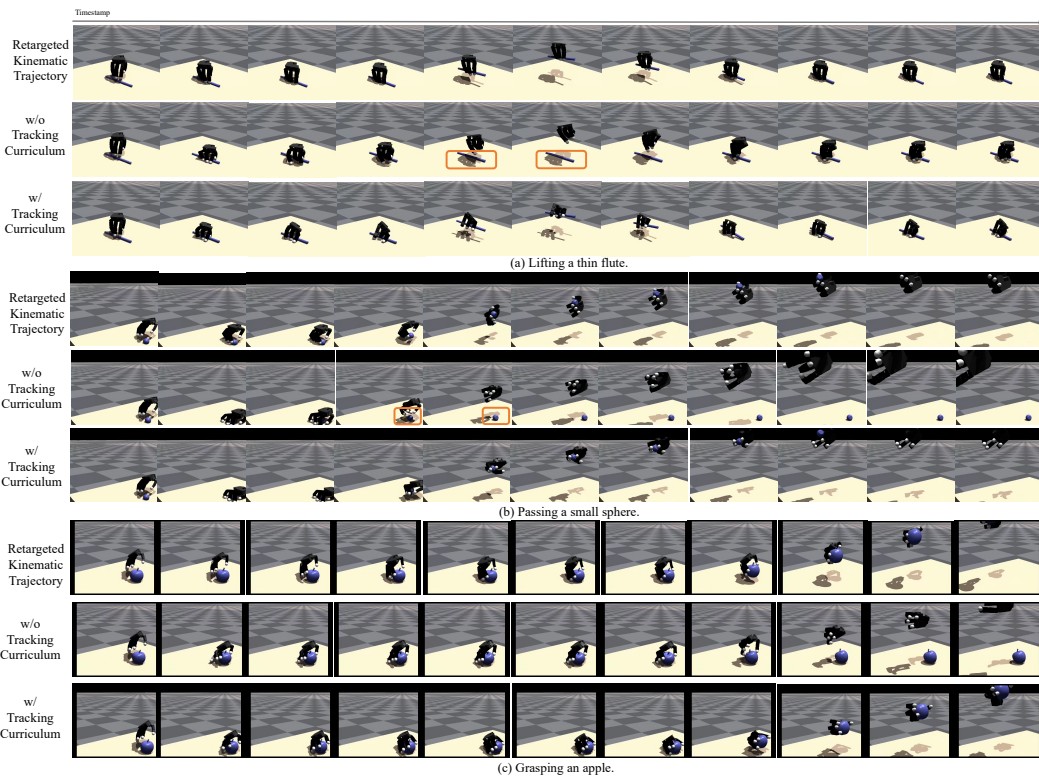

Figure 11: **Effectiveness of the homotopy optimization scheme.** Please refer to **our website** and **the accompanying video** for animated results.

We conduct further analysis of the proposed homotopy optimization scheme and the homotopy path generator to demonstrate their effectiveness. As shown in Figure 11, by optimizing through the homotopy optimization path, we can get better results in per-trajectory tracking.

**Lifting thin objects.** As demonstrated in Figure 11a, for the originally unsolvable tracking problem where we should manage to lift a very thin flute up from the table, we can finally ease the tracking

Table 9: Generalization experiments on the homotopy path generator.

| | Homotopy test (a) | Homotopy test (b) | Homotopy test (c) | Homotopy test (d) |
|---|---|---|---|---|
| Effectiveness Ratio (%) | 64.0 | 56.0 | 28.0 | 52.0 |

difficulty by gradually solving each tracking problem in the homotopy optimization path proposed by the generator.

**Grasping small objects.** As shown in Figure 11b, the original per-trajectory tracker fails to find a proper way to grasp the small sphere and lift it up from the table. However, empowered by the homotopy optimization, we can finally find a way to lift it up from the table.

**Lifting a round apple.** Figure 11c demonstrates an effective homotopy optimization path that can let us lift an apple up from the table which would previously challenge the policy due to the round surface.

**Generalization experiments on the homotopy path generator.** To further understand the generalization ability of the homotopy generator, we conduct the following tests:

- (a) Train the path generator via homotopy paths mined from GRAB's training set and evaluate it on the first test set that contains 50 tracking tasks uniformly randomly selected from remaining tracking tasks in GRAB's training set that are not observed by the homotopy generator.
- (b) Evaluate the path generator trained in (a) on the second test set that contains 50 tracking tasks uniformly randomly selected from the test tracking tasks of GRAB's test set.
- (c) Evaluate the path generator trained in (a) on 50 tracking tasks uniformly randomly selected from the test tracking tasks of TACO's first-level test set.
- (d) Train the path generator via homotopy paths mined from both GRAB's and TACO's training set and evaluate it on the test set used in (c).

For each tracking task, if the tracking results obtained by optimizing through the optimization path are better than the original tracking results produced by RL-based per-trajectory tracking, we regard the generated homotopy optimization path as an effective one. Otherwise, we regard it as ineffective. We summarize the ratio of the effective homotopy optimization paths in Table 9.

In summary:

- (a) the homotopy path generator can perform relatively well in the in-distribution test setting;
- (b) the performance would decrease slightly as the manipulation patterns shift a bit (please refer to section 4.1 for the difference between GRAB's training split and the test split);
- (c) the path generator would struggle to generalize to relatively out-of-distribution tracking tasks involving brand new objects with quite novel manipulation patterns;
- (d) Increasing the training data coverage for the homotopy path generator would let it get obviously better.

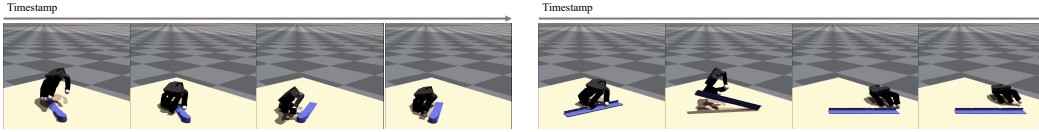

Figure 12: **Failure Cases.** Please refer to **our website** and **the accompanying video** for animated results.

## B.4 FAILURE CASES

Our method may fail to perform well in some cases where the object is from a brand new category with challenging thin geometry, as demonstrated in Figure 12.

## C    ADDITIONAL EXPERIMENTAL DETAILS

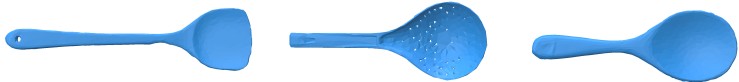

Figure 13: Examples of novel objects from the seen object category (TACO).

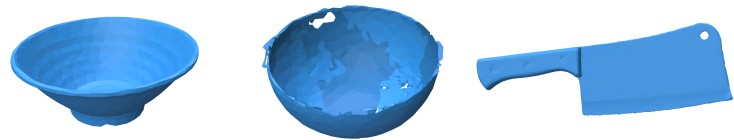

Figure 14: Examples of objects from new object categories (TACO).

**Datasets.** Our dexterous robot hand-object manipulation dataset is created by retargeting two public human-object datasets, namely GRAB Taheri et al. (2020), containing single-hand interactions with daily objects, and TACO Liu et al. (2024b), featured by functional tool using interactions. We retarget the full GRAB dataset and the fully released TACO dataset, obtaining 1269 and 2316 robot hand manipulation sequences respectively. For GRAB, we use sequences of the `subject s1`, with 197 sequences in total, as the test dataset. The training dataset is constructed by remaining sequences from other subjects. For the TACO dataset, we create one training set with four distinct test sets with different generalization levels for a detailed evaluation of the model's generalization performance. Specifically, the whole dataset is split into 1) a training dataset, containing 1565 trajectories, 2) test set S0 where both the tool object geometries and the interaction triplets are seen during training containing 207 trajectories in total, 3) test set S1 where the tool geometry is novel but the interaction triplets are seen during training with 139 trajectories, 3) test set S2 with novel interaction triplets but seen object categories and geometries, containing 120 trajectories in total, and 4) test set S3 with 285 trajectories where both the object category and interaction triplets are new to the training dataset. Figure 13 and Figure 14 draw the examples of unseen objects from seen categories and the objects from new categories respectively. The original data presented in TACO often contains noisy initial frames where the hand penetrates through the table or the object. Such noise, though seems subtle, would affect the initial dynamics, however. For instance, if the hand initially penetrates through the table, a large force would be applied to the hand at the beginning, which would severely affect the simulation in subsequent steps. Moreover, if the hand initially penetrates through the object, the object would be bounced away at the start of the simulation. To get rid of such phenomena, we make small modifications to the original sequences. Specifically, we interpolate the `phone pass` sequence of the subject `s2` from the GRAB dataset with such TACO sequences as the final modified sequence. Specifically, we take hand poses from the initial 60 frames of the GRAB sequence. We then linearly interpolate the hand pose in the 60th frame of the GRAB sequence with the hand pose in the 60th frame of the TACO sequence. For details, please refer to code in the supplementary material (refer "README.md" for instructions).

**Training and evaluation settings.** For both GRAB and TACO, in the first stage, we first sample 100 trajectories from the training dataset. We train their per-trajectory trackers to obtain their action-labeled data to construct our first version of the labeled dataset. After that,

Table 10:    Total training time consumption (TACO dataset).

|  | PPO (w/o sup) | Ours (w/o prior., w/o curri.) | Ours (w/o prior) | Ours |
|---|---|---|---|---|
| Time | ~1 day | ~2 days | ~4 days | ~4 days |

the first tracking controller is trained and evaluated on all trajectories. We then additionally sample 100 trajectories from the remaining trajectories using the weights positively proportional to the tracking object position error. These sampled trajectories and those sampled in the first stage then form our second version of the dataset to label. We leverage both per-trajectory tracker optimization and the tracking prior from the first version of the trained tracking controller to label the data aiming to get high-quality labeled trajectories. After that, we search tracking curricula from such 200 trajectories and train a tracking curriculum scheduler using the mined curricula. We then construct

the second version of the action-labeled dataset using the final best-optimized trajectories. We then re-train the tracking controller and evaluate its performance on each trajectory. In the third stage, we sample additionally 200 trajectories from the remaining not selected trajectories. We then label them using the joint power of per-trajectory tracking optimization, tracking prior from the tracking controller, and the curriculum scheduled by the curriculum scheduler. After that, our third version is constructed using best-optimized labeled trajectories. We then re-train the tracking controller using this version of the labeled dataset. When training the tracking controller, we set a threshold in the reward, *i.e.,* 50. Only trajectories with a reward above the threshold is used to provide supervision. Both the simulation and the policy run at 60Hz. The hand's gravity is ignored in the simulation. Please refer to the code in the supplementary materials for detailed settings. We train all models in a Ubuntu 20.04.6 LTS with eight NVIDIA A10 cards and CUDA version 12.5. All the models are trained in a single card without multi-gpu parallelization.

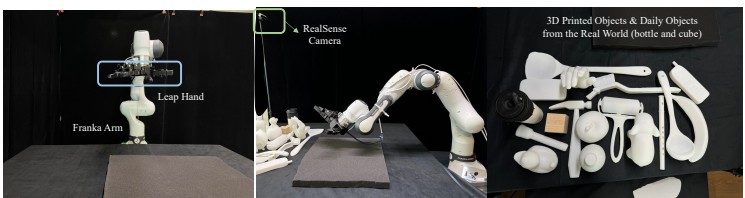

Figure 15: Real-world experiment setup.

**Real world experiment setup.** We use the Franka arm and LEAP hand to conduct real-world evaluations (Figure 15). When transferring the state-based policy, we use FoundationPose Wen et al. (2023) to estimate object poses. We use finite difference to estimate both the hand joint velocities as well as the object linear and angular velocities. Considering the gap between the control strategy we use in the simulator and the control of the Franka arm and the LEAP hand, we devise a strategy to mitigate the discrepancy between these two control methods. Specifically, instead of directly applying the control signal to the LEAP hand and the Franka arm, we set up a simulator with physical and control-related parameters same as our simulation settings during training. Then, in each timestep, we first apply the control commands to the simulated LEAP hand. We then simulate the hand. After that, we read the simulated state out from the simulator. We then calculate the positional target signal that should be applied to the real LEAP hand using the current obtained state from the simulator and the current state of the real LEAP hand. The calculated positional target signal is then directly fed to the real LEAP hand controller. In our observation, once the real hand has been commanded, it can almost reach exactly the same position in the command. Thereby, in practice, we directly use the state obtained from the simulator as the positional target command fed to the real LEAP controller. Experiments demonstrate the effectiveness of this control strategy.

**Time consumption and time complexity.** Table 10 summarizes the total time consumption of different methods on the TACO dataset. Directly training PPO without any supervision is the most efficient approach while the performance lagged behind due to no proper guidance. Solving the per-trajectory tracking problem for providing high-quality data for training the general tracking controller would additionally increase the time consumption due to the requirement in optimizing per-trajectory trackers. Since we only select a subset from the whole training dataset, the time consumption is still affordable. Improving the per-trajectory trackers via mining tracing curricula would introduce additional time costs. Since the number of trajectories we consider for learning the curriculum scheduler is still controlled to a relatively small value. The final time cost is still relatively affordable. Experiments are conducted on a Ubuntu 20.04 machine with eight A10 GPU cards. For per-trajectory tracker optimization, we train eight trackers in parallel at one time.

The overall time complexity of the training process is $\mathcal{O}(|\mathcal{S}| + KK_{\mathrm{nei}}|\mathcal{S}|)$. $\mathcal{S}$ denotes the training dataset.

**"Robustness", "Generalization Ability", and "Adaptivity".** We try to give computational definitions of some crucial characteristics of the neural tracking controller. Please note that there is no standard formal computational definitions of such concepts, to our best knowledge. The definition and the quantification we present here is only from our perspective.

Table 11: **Generlaization score (GRAB dataset).** **Bold red** numbers for best values.

| Method | DGrasp | PPO (OmniGrasp rew.) | PPO (w/o sup., tracking rew.) | Ours (w/o data, w/o homotopy) | Ours (w/o data) | Ours |
|---|---|---|---|---|---|---|
| $s_g$ | 2.424 | 2.389 | 2.688 | 2.725 | 3.050 | **3.251** |

Table 12: **Robustness score (GRAB dataset).** **Bold red** numbers for best values.

| Method | PPO (w/o sup., tracking rew.) | Ours |
|---|---|---|
| $s_r$ | 2.665 | **3.276** |

- To quantify the "generalization ability", we need to first quantify 1) the distribution gap between two trajectory distributions, so that we can define the levels of generalization, 2) the generalization ability from the training distribution to the test distribution via the gap between the model performance:

Denote $\mathcal{E}$ as the distribution of the tracking task of the training set, and $\mathcal{D}$ as the test distribution, we define their gap as follows:

$$d(\mathcal{D}; \mathcal{E}) = \mathbb{E}_{\mathbf{T} \sim \mathcal{D}} \left[ \min_{\mathbf{T_{train}} \sim \mathcal{E}} (\text{Tracking\_Task\_Diff}(\mathbf{T}, \mathbf{T_{train}})) \right], \quad (15)$$

where Tracking_Task_Diff$(\cdot, \cdot)$ measures the differnece between two manipulation trajectory tracking problem. For a trajectory tracking task described by the kinematic hand state sequence $\{\mathbf{s}_n^h\}_{n=0}^N$, the kinematic object pose sequence $\{\mathbf{p}_n^o, \mathbf{q}_n^o\}_{n=0}^N$, and the object geometry PC, *e.g.,* represented as the point cloud, we calculate the trajectory tracking task difference between two tracking tasks, *i.e.,* $\mathbf{T}_A = \{\{\mathbf{s}_n^{h,A}\}, \{\mathbf{p}_n^{o,A}, \mathbf{q}_n^{o,A}\}, \text{PC}^A\}$ and $\mathbf{T}_B = \{\{\mathbf{s}_n^{h,B}\}, \{\mathbf{p}_n^{o,G}, \mathbf{q}_n^{o,G}\}, \text{PC}^B\}$, as a weighted sum of the hand trajectory difference, object pose sequence difference and the object geometry difference:

$$\text{Tracking\_Task\_Diff}(\mathbf{T}_A, \mathbf{T}_B) = \frac{1}{N+1} \sum_{n=0}^N (w_{diff}^h \|\mathbf{s}_n^{h,A} - \mathbf{s}_n^{h,B}\| + w_{diff}^{o,p} \|\mathbf{p}_n^{o,A} - \mathbf{p}_n^{o,B}\| \tag{16}$$

$$+ w_{diff}^{o,q} \|\mathbf{q}_n^{o,A} - \mathbf{q}_n^{o,B}\|) \tag{17}$$

$$+ w_{diff}^{pc} \text{Chamfer-Distance}(\text{PC}^A, \text{PC}^B), \tag{18}$$

where $w_{diff}^h = 0.1, w_{diff}^{o,p} = 1, w_{diff}^{o,q} = 0.3, w_{diff}^{pc} = 0.5$. For a tracking policy $\pi$, define its performance on tracking tasks from the distribution $\mathcal{E}$ via the expectation of tracking errors:

$$L_{\mathcal{E}}(\pi) = \mathbb{E}_{\mathbf{T} \sim \mathcal{E}} [\text{Tracking\_Error}_\pi(\mathbf{T})], \tag{19}$$

where Tracking_Error$_\pi(\cdot)$ evaluates the tracking error of the tracking policy $\pi$ on a trajectory tracking problem. It is a weighted sum of the difference between the tracking result and the reference trajectory of the robot hand and that of the object:

$$\text{Tracking\_Error}_\pi(\mathbf{T}) = w_{err}^{o,p} T_{err} + w_{err}^{o,q} R_{err} + w_{err}^{h,wrist} E_{wrist} + w_{err}^{h,finger} E_{finger}, \tag{20}$$

where $w_{err}^{h,wrist} = 0.1, w_{err}^{h,finger} = 0.1, w_{err}^{o,p} = 1.0, w_{err}^{o,q} = 0.3$.

Therefore, the genralization ability of the policy $\pi$ trained to solve the trajectories from trianing distribution $\mathcal{E}$ to the test distribution $\mathcal{D}$ is measured as:

$$s_g = \frac{d(\mathcal{D}; \mathcal{E})}{\min(L_{\mathcal{D}}(\pi), \epsilon)}, \tag{21}$$

where $\epsilon$ is a small value to avoid numerical issue. The score $s_g$ increases as the train-test distribution gap increases or the tracking error on the test distribution decreases.

Using the above quantification w.r.t. the tracking task distribution gap and the generalization ability score, we summarize the generalization ability score of different kinds of models achieved on the GRAB dataset in Table 11.

Table 13: **Adaptativity score (GRAB dataset).** **Bold red** numbers for best values.

| Method | PPO (w/o sup., tracking rew.) | Ours |
|--------|-------------------------------|------|
| $s_a$ | 0.317/0.098/0.0 | **0.537/0.415/0.293** |

- To quantify the "robustness", since it is quite difficult to analyze the dynamic function with a neural controller, we measure the robustness by the performance discrepancy between the tracking policy's performance on the tracking tasks with relatively high-quality kinematic trajectories to those with disturbed kinematic references.

  To measure the "quality" of kinematic manipulation trajectories, we introduce three trends of quantities:

  - Smoothness: it calculates the difference between state finite differentiations :

  $$t_s(\mathbf{T}) := \frac{1}{N-1} \sum_{i=1}^{N-1} \frac{1}{\Delta t} \left( \frac{\mathbf{w_s} \cdot (\mathbf{s}_{i+1} - \mathbf{s}_i)}{\Delta t} - \frac{\mathbf{w_s} \cdot (\mathbf{s}_i - \mathbf{s}_{i-1})}{\Delta t} \right), \quad (22)$$

  where $\mathbf{w}_s$ is the weight vector of each state DoF, $\Delta t$ is the time between two neighbouring frames.

  - Consistency: it calculates the hand object motion consistency:

  $$t_c(\mathbf{T}) := \frac{1}{N-1} \sum_{i=1}^{N} \left\| \frac{\mathbf{p}_i^o - \mathbf{p}_{i-1}^o}{\Delta t} - \frac{\mathbf{p}_i^h - \mathbf{p}_{i-1}^h}{\Delta t} \right\|, \quad (23)$$

  where $\mathbf{p}_i^o$ is the object position at frame $i$ while $\mathbf{p}_i^h$ is the hand wrist position at the frame $i$.

  - Penetrations: it calculates the penetration between hand and object across all frames:

  $$t_p(\mathbf{T}) := \frac{1}{N+1} \sum_{i=0}^{N} \text{Pene\_Depth}(\mathbf{s}_i, \mathbf{P}^o), \quad (24)$$

  where $\mathbf{P}^o$ represents the object point cloud, Pene_Depth calculates the maximum penetration depth between hand and the object.

  "Quality" of kinematic manipulation references can be grounded using such three measurements. Combining the above quantities, define the overall "quality" of the test distribution $\mathcal{L}$ as:

  $$s_{quality}(\mathcal{L}) = \mathbb{E}_{\mathbf{T} \sim \mathbf{L}} \left[ \frac{t_s(\mathbf{T}) + t_c(\mathbf{T}) + t_p(\mathbf{T})}{3} \right]. \quad (25)$$

  The "robustness" can be measured by the performance gap of the model on tracking tasks with "high-quality" references and those with "perturbed" thus "low-quality" tracking tasks. Denote the "high-qualtiy" tracking task distribution as $\mathcal{H}$ while those with low quality as $\mathcal{L}$. Thus we can quantify the "robustness" as:

  $$s_r(\pi) := \frac{s_{quality}(\mathcal{L})}{\min(L_{\mathcal{L}}(\pi), \epsilon)}. \quad (26)$$

  As the quality of the trajectory distribution gets worse and the tracking error decreases, the "robustness score" would increase.

  To evaluate this, we construct a disturbed test set by adding random noise to the hand trajectory and the object position trajectory to test the trajectories of the GRAB dataset. After that, we test the performance of our method and the best-performed baseline, PPO (w/o sup., tracking rew.). We summarize the results in Table 12.

- We mainly focus on the real-world adaptivity to evaluate the "adaptivity" of the tracking controller. Since it is hard to quantitatively measure the discrepancy between the real-world dynamics and that in the simulator, we use the model's performance gap as the metric to directly evaluate the adaptivity. We use real-world per-trajectory average success rate to measure the "adaptivity". For our real-world test on trajectories from the GRAB dataset, we summarize the result in Table 14.

Table 14: **Generlaization score (GRAB dataset). Bold red** numbers for best values.

| Method | PPO (w/o sup., tracking rew.) | Ours |
|---|---|---|
| $s_a$ | 0.317/0.098/0.0 | **0.537/0.415/0.293** |

Table 15: **Trajectory difficulty statistics.**

| | $s^o_{smooth}(m \cdot s^{-2})$ | $v_{contact}$ | $s_{shape}(cm^{-1})$ |
|---|---|---|---|
| GRAB | 3.426 | 1.641 | 0.275 |
| TACO | 1.978 | 2.285 | 0.497 |

**Computational interpretation of "hard-to-track".** The most direct quantification is defining a score related to the tracking method's performance. For the tracking method $\mathcal{M}$ and the trajectory $\mathbf{T}$, define $\pi$ is the best tracking policy that we can optimize for the trajectory $\mathbf{T}$ (*i.e.,* $\pi = \mathcal{M}(\mathbf{T})$. Given the tracking error Tracking_Error$_\pi(\mathbf{T})$, the "hard-to-track" score could be defined as: $s_{ht} = \frac{1}{\min(\text{Tracking\_Error}_\pi(\mathbf{T}),\epsilon)}$, where $\epsilon$ is a small value to avoid numerical issue.

Besides, we can use some statistics of the hand and object kinematic trajectories to quantify the "hard-to-track" characteristic. Here we introduce three types of statistics: 1) object movement smoohtness $s^o_{smooth}$: it quantifies the motion smoothness by calculating the per-frame average object accelerations, *i.e.,* $s^o_{smooth} = \frac{1}{N-1}\sum_{i=1}^{N-1}\|\frac{1}{\Delta t}(\frac{\mathbf{p}^o_{i+1}-\mathbf{p}^o_i}{\Delta t} - \frac{\mathbf{p}^o_i-\mathbf{p}^o_{i-1}}{\Delta t})\|$, 2) hand-object contact shifting velocity $v_{contact}$: it quantifies the per-frame velocity of the contact map change, *i.e.,* $v_{contact} = \frac{1}{N}\sum_{i=1}^{N}\|\frac{\mathbf{c}_i-\mathbf{c}_{i-1}}{N_p\Delta t}\|$, where $\mathbf{c}_i \in \{0,1\}^{N_p}$ is the binary contact map which encodes the contact flag between sampled points from the hand surface and the object, $N_p$ is the number of points sampled from the hand, 3) object shape score $s_{shape}$: it is the z-axis extent of the object's bounding box to quantify the shape of the object: $s_{shape} = \frac{1}{\min(\text{extent}_z,\epsilon)}$, where extent $\in \mathbb{R}^3$ is the extent of the bounding box of the object. We can jointly use these three trends of scores to quantify the "hard-to-track" characteristic. As $s^o_{smooth}$ increases, $v_{contact}$ increases, and $s_{shape}$ increases, the trajectory would get more "difficult", thus "hard" to train a policy to track it. As shown in Table 15, the test trajectories in GRAB are more "difficult" than those in TACO w.r.t. the trajectory smoothness, while test trajectories in TACO are more difficult than those in GRAB regarding the contact velocity and the shape difficulty.

**Additional details.** For tracking error metrics, we report the medium value of per-trajectory result in the test set, under the consideration that the average value may be affected by outliers.

