# OpenReview forum: "DexTrack: Towards Generalizable Neural Tracking Control for Dexterous Manipulation from Human References"
_ICLR.cc/2025/Conference — ICLR 2025 Poster_

### Official Review · Reviewer_4aMi · 2024-11-02

**Soundness:** 3
**Presentation:** 2
**Contribution:** 3
**Rating:** 8
**Confidence:** 3

**Summary:**

The authors proposed a neural tracking controller that integrates the reinforcement learning and imitation learning by iteratively improving the controller and generating demonstrations in simulation. Specifically, they proposed a per-trajectory tracking scheme to generate diverse and high-quality tracking demonstrations through a homotopy optimization framework. The authors further show qualitative and quantitive results to evaluate their proposed method.

**Strengths:**

This work has solid contributions including novel approach and substantial experimental results. The work is well-motivated to present a generalizable tracking controller (policy learning) especially that can tackle dexterous manipulation. The framework the authors proposed was well-formulated. The experiments are thorough and results are well-presented and analyzed.

**Weaknesses:**

I found the method section was hard to follow and understand, and I would suggest to improve the writing.
1) It would be great to clearly define each terminology at the beginning such as for kinematic reference trajectory, expert trajectory, baseline trajectory, kinematic reference sequence, tracking prior and etc and use them consistently through the paper.
2) I found the pipeline in section 3.3 helpful for me to understand the whole framework, so I would suggest to move it to the beginning of the section 3.
3) Figure 2 has a lot information, but it was hard for me to find a proper order to read it. Maybe it can be improved with a more clear flowchart. It would be great to use consistent icons (multiple different ones for object) or remove redundant icons (such as what is the magnifying glass icon?)

Typo in section 4.1 Metric: racking -> tracking

**Questions:**

1) In section 3.1 Imitation Learning: Does expert state-action trajectory come from human demonstration? If so, how to address the kinematic and dynamic gaps here?

2) In section 3.2 Finding effective homotopy optimization paths: Could the authors clarify “better baseline trajectory” in the sentence “For a specific task, we consider a neighbor that provides … parent task."?

3) The real-world deployment seems impressive. I wonder beyond grasping, have the authors tried in-hand manipulation on real-world robot or were there any failure cases?

---

> ### Author Response · Authors · 2024-11-19
> **Author Response (Part 1 of 3)**
>
> Dear Reviewer 4aMi,
>
>
>
> Thank you so much for your detailed and constructive review. We greatly appreciate your recognition of our motivations, contributions, method design rationale, as well as the thorough experiments and well-presented results. In the responses below, we hope to adequately address any of your concerns and questions.
>
>
>
> **Improvements in the presentation of the method section**
>
> > - "..clearly define each terminology at the beginning such as for **kinematic reference trajectory**, **expert trajectory**, **baseline trajectory**, kinematic reference sequence, **tracking prior** and etc and use themconsistently through the paper. "
> > - "I found the pipeline in section 3.3 helpful for me to understand the whole framework, so I would suggest to move it to the beginning of the section 3."
> > - "Figure 2 has a lot information, but it was hard for me to find a proper order to read it...a more clear flowchart...use consistent icons (multiple different ones for object) or remove redundant icons (such as what is the magnifying glass icon?)"
>
> Thank you so much for your detailed and constructive suggestions on improving the method section. They are quite valuable for us. We've **revised the method section accordingly** in the revision **(highlighted in blue)**, trying our best to improve its clarity. Below is a summary of our modifications:
>
> - **Terminologies and notations:** We add a **"*Terminologies and notations*"** paragraph at the beginning of the method section. As suggested, we clearly explained key concepts such as *"kinematic reference trajectory," "expert trajectory," "kinematic reference sequence,"* and *"tracking demonstration."* The term *"baseline trajectory"* requires the context of our policy and action space design, so we have retained its introduction in the ***"Reinforcement Learning"*** paragraph of ***section 3.1***. We verified that *"baseline trajectory"* does not appear earlier in the text. Similarly, *"tracking prior"* is explained when it first appears since it needs the context of the tracking controller. We also revised the paragraph ***"Transferring 'tracking prior'"*** for greater clarity.
> - **Section 3.3 placement:** Section 3.3 assumes the knowledge of the training process of the tracking controller, the mining of tracking demonstrations, and the homotopy optimization scheme. Moving it to the beginning might confuse readers unfamiliar with these contexts. Instead, we refined the method overview paragraph to better articulate the interdependence between *"tracking controller training"* and *"mining tracking demonstrations."* We hope this offers a clearer overview of our method.
>
> - **Method figure:** Thanks for your feedback and suggestions. We revised the method figure to improve its structure and clarity. It now has two parts: the left part illustrates the tracking demonstration mining strategy while the right part depicts how we learn a tracking controller from demonstrations. Our method then iterates between these two steps to train a powerful tracking controller.
>
>   **Questions regarding the original method figure:** While we initially aimed to present both the overall structure and detailed components, we recognize this made the figure overly complex. In the original figure, the "magnifying glass" icon represents the single trajectory tracking policy. Using this icon can simplify our repeated use of single trajectory tracking when explaining the homotopy optimization scheme. For object icons, we previously used multiple icons to represent different tasks. For example, when illustrating the homotopy optimization path generator, we show the transformation from "manipulating a hammer" to "manipulating a flashlight." Thus, we use different icons to represent "hammer" and "flashlight", respectively.

---

> ### Author Response · Authors · 2024-11-19
> **Author Response (Part 2 of 3)**
>
> > **"Does expert state-action trajectory come from human demonstration? If so, how to address the kinematic and dynamic gaps here?"**
>
> Thanks for your detailed review and this great question. Our expert state-action trajectory comes from human demonstration. ***The key problem*** that we solve in the robot demonstration mining process ***is exactly how to address the kinematic and dynamic gaps***. To address the kinematic and dynamic gaps, we need to convert such kinematic-only human hand-object manipulation trajectories into dynamic-aware **joint action commands for the robotic hand** that can control the robot hand to precisely track the human demonstration. Given the morphological differences between human and robotic hands, we infer the robotic hand's joint command sequence through the following steps:
>
> - **Kinematic retargeting:** We retarget human hand-object interaction trajectories to generate kinematic robot hand-object manipulation trajectories (details in *Appendix A.1*).
> - **Tracking the retargeted manipulation sequences:** To generate high-quality robotic tracking demonstrations from retargeted kinematic-only sequences, we employ three strategies (details in *Section 3.2*):
>   - **RL-based per-trajectory tracking:** We train a policy RL to convert kinematic-only manipulation trajectories into dynamic robot joint command sequences. The policy takes the kinematic reference trajectory of the robotic hand and the object as input, outputting joint commands to minimize the tracking error.
>   - **Enhancing per-trajectory tracking via the tracking controller:** For challenging trajectories, the RL-based scheme alone may struggle to close the kinematic and dynamic gaps. To address this, we utilize the tracking controller to initialize the per-trajectory tracking policy, leveraging tracking priors to reduce tracking difficulty.
>   - **Improving per-trajectory tracking via cross-task relations:** We devise a homotopy optimization scheme to leverage cross-task relations. By employing an effective homotopy optimization path, it eases the difficulty of a specific per-trajectory tracking task by initiating its tracking policy using the tracking results of other tasks.
>
>
>
> > **"In section 3.2 Finding effective homotopy optimization paths: Could the authors clarify “better baseline trajectory” in the sentence “For a specific task, we consider a neighbor that provides … parent task."?"**
>
> Thank you for your careful review and the insightful question. The "better baseline trajectory"  is indeed an important concept in our homotopy optimization scheme. It is related to both our policy design and the details regarding how we leverage cross-tracking task relations to help improve a specific tracking task's solving. We will begin by introducing the concept of a "baseline trajectory," then explain how cross-task relations are leveraged to improve per-trajectory tracking policy learning, and finally clarify what's a "better baseline trajectory":
>
> - **Baseline trajectory in policy design:** As explained in the ***"Reinforcement Learning"*** paragraph of Section 3.1, we introduce a *baseline trajectory* and let the policy predict the residual relative target at each timestep. The joint targets are then computed based on the baseline trajectory and the predicted residual relative targets.  The baseline trajectory is usually set as the kinematic robot hand trajectory of the kinematic reference trajectory we wish to track. This design offers two advantages:
>   - Reformulating the action space as residual relative targets simplifies policy learning and accelerates convergence.
>   - It adds an optimization dimension—beyond optimizing model weights for better actions, we can enhance the policy by finding *a better baseline trajectory*. By using this improved baseline trajectory in conjunction with RL-optimized residuals, we achieve better tracking performance compared to the policy trained using the original kinematic references as the baseline trajectory.
> - **Leveraging cross-tracking task relations for per-trajectory tracking:** Assume we want to solve the single trajectory tracking problem for task $A$. When trying to improve task $A$ via task $B$, we set $A$'s baseline trajectory to $B$'s tracking results. We then re-train the tracking policy to predict the residual relative targets, aiming to solve the task $A$.
> - **Better baseline trajectory:** We say $B$ can provide **a better baseline trajectory** for $A$ than $A$'s kinematic trajectory if the tracking results of $A$ produced by the policy trained with the baseline trajectory set to $B$'s tracking results can track $A$ better than those produced by the policy trained with the baseline trajectory set to $A$'a kinematic hand trajectory.

---

> ### Author Response · Authors · 2024-11-19
> **Author Response (Part 3 of 3)**
>
> > **"The real-world deployment seems impressive. I wonder beyond grasping, have the authors tried in-hand manipulation on real-world robot or were there any failure cases?"**
>
> Thank you for your appreciation and insightful question. Demonstrating intriguing in-hand manipulations in the real world would be quite interesting. The current in-hand manipulation that can be achieved in the real world can only cover relatively slight re-orientations. It is still challenging for us to achieve large in-hand manipulations such as re-orienting the object for a large angle against gravity. The primary reason comes from the control gap between the simulation and the real world. We are actively exploring this side.
>
> **In-hand manipulations on real robots:** The *most successful case that has a subtle in-hand slight re-orientation* is the case that we show in the first line of real-world experiment videos on the [website](https://projectwebsite7.github.io/gene-dex-manip/) (the hand re-orient the object for an angle using the *little finger*).
>
> **Failure cases:** We have updated the [project website](https://projectwebsite7.github.io/gene-dex-manip/) with a section on **Real-World Evaluations -- Failure Cases** to provide more insights. A primary failure reason for in-hand manipulation against gravity is the difficulty in maintaining a firm grip when contact points shift, often leading to the object falling off.
>
>
>
> > **racking -> tracking** :
>
> Thanks. We've fixed it in the revised version.
>
>
>
>
>
>
>
>
>
> Finally, thank you again for your time and detailed constructive review. We are more than willing to address any further questions and will make every effort to resolve any concerns you may have. Please feel free to let us know if there is anything specific we can provide or clarify.
>
> Looking for your response!
>
> Best regards, Authors

---

> ### Author Response · Authors · 2024-11-23
> **Thank you for your constructive review and look forward to post-rebuttal feedback**
>
> Dear Reviewer 4aMi,
>
> The weekend is coming or already here! Wish you a nice, pleasant, and relaxing weekend!
>
> We'd like to express our sincere gratitude for your detailed and careful reviews, insightful questions, and constructive suggestions.
>
> We have provided responses to all of your questions and have incorporated your suggestions in our revision (highlighted in blue). We hope our explanations and revisions could fully answer your questions and address your concerns.
>
> As the discussion phase will conclude on November 26th, please don’t hesitate to let us know if you have any further questions. We are committed to exerting every effort to address any remaining concerns you may have. We appreciate your suggestions.
>
> Thank you for your time and constructive reviews! Sincerely look forward to your responses.
>
> Best regards,
>
> Authors

---

> > ### Author Response · Authors · 2024-12-02
> > **Thank you for your valuable feedback and look forward to post-rebuttal responses**
> >
> > Dear Reviewer 4aMi,
> >
> > We would like to kindly remind you that tomorrow is **the last day** of the discussion period. If there are any remaining questions or points you would like us to address, please feel free to let us know. We are fully available to discuss any aspects further.
> >
> > Thank you again for your valuable feedback.
> >
> > Best regards, Authors

---

### Official Review · Reviewer_gJWM · 2024-11-03

**Soundness:** 4
**Presentation:** 4
**Contribution:** 3
**Rating:** 6
**Confidence:** 3

**Summary:**

This paper presents DexTrack, a generalizable neural tracking controller for dexterous manipulation, trained on large-scale robot tracking demonstrations based on human-object interactions. The method integrates reinforcement learning, imitation learning, and homotopy optimization to improve controller performance, diversity, and adaptivity in dynamic environments. DexTrack achieves over 10% higher success rates than leading baselines in both simulation and real-world tests, demonstrating robust manipulation capabilities across various objects and tasks.

**Strengths:**

Originality: The method is original, and combining RL and IL makes perfect sense for training a controller with high-quality demonstrations. I especially like the use of homotopy optimization as a path generator to enhance data quality.

Quality: The experimental results are thorough and solid, both qualitatively and quantitatively. A substantial number of simulation experiments are conducted, and I appreciate the inclusion of real-world results as well.

Clarity: The paper is clear, well-written, and easy to understand.

Significance: This work represents a solid advancement toward solving the challenging task of dexterous hand manipulation, which is a significant objective in robotics.

**Weaknesses:**

The real-world robot results are not very impressive, as the grasp does not appear to be consistently robust, which may limit the method’s applicability in practical settings. It’s understandable, however, that real-world evaluations can be influenced by various factors, such as hardware limitations, sensor noise, and environmental variability, all of which can impact performance.

**Questions:**

Are there any limitations to the proposed homotopy generator beyond its speed? While speed is certainly a factor, there may also be other limitations, such as the generator’s ability to generalize across a diverse range of objects and trajectories. It would be valuable to understand if the homotopy generator maintains high performance across these variations or if specific configurations present challenges.

---

> ### Author Response · Authors · 2024-11-19
> **Author Response (Part 1 of 3)**
>
> Dear Reviewer gJWM,
>
>
>
> Thank you sincerely for your careful and constructive review. We'd like to express our huge gratitude for your recognition of the originality of the methodology, results' quality, solid experiments, clarity, and the significance of our achievements. In the following text, we respond to your questions and aim to address your concerns comprehensively.
>
>
>
> > **"The real-world robot results are not very impressive...real-world evaluations can be influenced by various factors..."**
>
> Thank you so much for your thoughtful and nice comments on real-world evaluations.
>
> - **Difficulty in learning a generalizable tracking controller:** The focus of this work lies in how to effectively learn a generalizable tracking controller for dexterous manipulations. Tackling the learning problem in simulation is *very difficult* where previous works have limitations in either task complexity [OmniGrasp, DGrasp, UniDexGrasp] or are restricted in developing specific skills [HORA, GenRot, TwistingLids, QuasiSim, ComplementaryFree].
> - **Real-world deployment for dexterous manipulations:** Real-world deployment in this work primarily aims at demonstrating the potential of equipping a real robot hand with the ability of generalizable dexterous manipulation skills. We highly agree that transferring dexterous manipulation skills to the real world faces numerous sim-to-real challenges, such as the misalignment in dynamics properties, control strategies, physics parameters and observation modality, limitations in sensor capability, as well as variations in the environment, etc. Fully addressing sim-to-real challenges requires serious efforts and is a crucial point in our future work. We are committed to making steady improvements in our real-world performance and hope to demonstrate significantly better results in the near future.
> - **Achievements in prior works:** Besides, ***demonstrating intriguing manipulation skills on a real robot,*** such as those involving in-hand manipulations with contact variations in the in-hand manipulation with contact changes in the presence of gravity, is extremely difficult. We haven't observed satisfactory results from previous work yet. For instance, in a [recent work](https://cypypccpy.github.io/obj-dex.github.io/) [ObjCentricDexManip], even with two hands, they do not show interesting manipulations against gravity on the real robot. Others demonstrating interesting in-hand manipulations always assume the object is already held in an up-facing hand (*e.g.,* [[HORA](https://haozhi.io/hora/)] [[GenRot](https://haozhi.io/rotateit/)]) with re-orienting or with two hands where one of them can hold the object against gravity (*e.g.,* [[TwistingLids](https://toruowo.github.io/bimanual-twist/)]).
>
> ​
>
> *[OmniGrasp] Luo, Zhengyi et al. “Grasping Diverse Objects with Simulated Humanoids.” ArXiv abs/2407.11385 (2024): n. pag.*
>
> *[DGrasp] Christen, Sammy Joe et al. “D-Grasp: Physically Plausible Dynamic Grasp Synthesis for Hand-Object Interactions.” 2022 IEEE/CVF Conference on Computer Vision and Pattern Recognition (CVPR) (2021): 20545-20554.*
>
> *[UniDexGrasp] Xu, Yinzhen et al. “UniDexGrasp: Universal Robotic Dexterous Grasping via Learning Diverse Proposal Generation and Goal-Conditioned Policy.” 2023 IEEE/CVF Conference on Computer Vision and Pattern Recognition (CVPR) (2023): 4737-4746.*
>
> *[HORA] Qi, Haozhi et al. “In-Hand Object Rotation via Rapid Motor Adaptation.” Conference on Robot Learning (2022).*
>
> *[GenRot] Qi, Haozhi et al. “General In-Hand Object Rotation with Vision and Touch.” ArXiv abs/2309.09979 (2023): n. pag.*
>
> *[TwistingLids] Lin, Toru et al. “Twisting Lids Off with Two Hands.” ArXiv abs/2403.02338 (2024): n. pag.*
>
> *[QuasiSim] Liu, Xueyi et al. “QuasiSim: Parameterized Quasi-Physical Simulators for Dexterous Manipulations Transfer.” ArXiv abs/2404.07988 (2024): n. pag.*
>
> *[ComplementaryFree] Jin, Wanxin. “Complementarity-Free Multi-Contact Modeling and Optimization for Dexterous Manipulation.” ArXiv abs/2408.07855 (2024): n. pag.*

---

> ### Author Response · Authors · 2024-11-19
> **Author Response (Part 2 of 3)**
>
> >  **"...any limitations to the proposed homotopy generator beyond its speed?...such as the generator’s ability to generalize across a diverse range of objects and trajectories...It would be valuable to understand if the homotopy generator maintains high performance across these variations or if specific configurations present challenges."**
>
> Thanks for your insightful question. As a learning-based model, the homotopy path generator would have limitations in other aspects. The generalization ability you've mentioned is quite a great point. In the following text, we will detailedly analyze its generalization ability and will propose its other limitations beyond the generalization.
>
> - **Performance in our experiments:** Since the path generator is expected to be able to generalize to propose effective optimization paths for a new tracking task, it should have some generalization ability. We are initially really concerned about its generalizability in the early stage of the project, considering the very limited amount of data we can leverage to train the path generator. Luckily, it can work well in our experiments. That's primarily because it only needs to tackle relatively in-distribution cases in our problem. We leverage the path generator to improve the single trajectory tracking for tasks from the training split of each dataset. Thus, the tracking tasks it faces during the inference time and those involved in its training data all come from the same split.
>
> - **Generalization experiments:** The generalization ability of the path generator, though is enough in our experiments, would still be restricted, considering the limited amount of its training data (optimization paths mined from several hundred tracking tasks).  To further understand the generalization ability of the homotopy generator, we conduct the following tests:
>
>   - (a) Train the path generator via homotopy paths mined from GRAB's training set and evaluate it on the first test set that contains 50 tracking tasks uniformly randomly selected from remaining tracking tasks in GRAB's training set that are not observed by the homotopy generator.
>   - (b) Evaluate the path generator trained in (a) on the second test set that contains 50 tracking tasks uniformly randomly selected from the test tracking tasks of GRAB's test set.
>   - (c) Evaluate the path generator trained in (a) on 50 tracking tasks uniformly randomly selected from the test tracking tasks of TACO's first-level test set.
>   - (d) Train the path generator via homotopy paths mined from both GRAB's and TACO's training set and evaluate it on the test set used in (c).
>
>   For each tracking task, if the tracking results obtained by optimizing through the optimization path are better than the original tracking results produced by RL-based per-trajectory tracking, we regard the generated homotopy optimization path as an effective one. Otherwise, we regard it as ineffective. We summarize the ratio of the effective homotopy optimization paths as follows:
>
>   |                         | Homotopy test (a) | Homotopy test (b) | Homotopy test (c) | Homotopy test (d) |
>   | ----------------------- | ----------------- | ----------------- | ----------------- | ----------------- |
>   | Effectiveness Ratio (%) | 64.0              | 56.0              | 28.0              | 52.0              |
>
>   In summary:
>
>   - (a) The homotopy path generator can perform relatively well in the in-distribution test setting;
>   - (b) The performance would decrease slightly as the manipulation patterns shift a bit (please refer to section 4.1 for the difference between GRAB's training split and the test split);
>   - (c) The path generator would struggle to generalize to relatively out-of-distribution tracking tasks involving brand-new objects with quite novel manipulation patterns;
>   - (d) Increasing the training data coverage for the homotopy path generator would let it get better.

---

> ### Author Response · Authors · 2024-11-19
> **Author Response (Part 3 of 3)**
>
> - **Limitations in generalization:** From the above experiments, we make the following observations:
>
>   - (a) The homotopy path generator can perform relatively well in the in-distribution test setting;
>   - (b) The performance would decrease slightly as the manipulation patterns shift a bit (please refer to section 4.1 for the difference between GRAB's training split and the test split);
>   - (c) The path generator would struggle to generalize to relatively **out-of-distribution tracking tasks** involving brand-new objects with quite novel manipulation patterns;
>   - (d) Increasing the training data coverage for the homotopy path generator would let it get better.
>
>   Though currently limited, the experiments above point out the role training data played in the generalization ability of the homotopy path generator. Its generalizability would get stronger as we expand its training data to cover effective optimization paths mined from more and more diverse tracking tasks.
>
>   **Other reasons that may restrict the generalization ability:** Apart from the training data, as we do not rigorously investigate the model architecture of the homotopy path generator as well as the tracking task representation, it is possible that their current details are not the most suitable choice to train a generalizable homotopy path generator.
>
>   How to design better architectures and tracking task representations, as well as how to utilize data in the most efficient and effective way to improve the generalization ability of the homotopy path generator, is an interesting future research direction and would bring opportunities for improving the robot tracking demonstrations.
>
> - **Other limitations:** Another issue comes from the multi-modal behavior of the homotopy path generator. We use a conditional diffusion model to describe the cross-task relations in the homotopy optimization path. On one side, it is quite suitable to describe the multi-modal cross-task relations, *i.e.,* one tracking task may have multiple different "parent" tasks. On the other hand, during the inference, it is possible that the model is not predicting the most promising homotopy path. This behavior brings us both opportunities and challenges:
>
>   - On one side, we can query the path generator to generate multiple possible homotopy paths. By optimizing through each of them, we get multiple tracking results. After that, we can pick the best one for the tracking demonstration. This strategy, in a similar flavor to the "bagging" algorithm philosophically, can go beyond the ability of a single homotopy path and provide us with a better demonstration.
>   - On the other side, we have no idea how many samples we need to get the most promising homotopy path. The "bagging" strategy, together with the additional inference time cost of the diffusion model over feed-forward architectures, would introduce additional inference time costs.
>   - **Possibilities to solve this issue** include 1) replacing the diffusion model with a deterministic backbone and 2) introducing guided sampling techniques to reduce the variance, which guides the model to sample trajectories towards certain criteria, such as the simplicity of the trajectory.
>
>
>
> Finally, thank you again for your valuable and constructive review. We are more than willing to address any further questions and will make every effort to resolve any concerns you may have. Please feel free to let us know if there is anything specific we can provide or clarify.
>
> Looking for your response!
>
> Best regards, Authors

---

> ### Author Response · Authors · 2024-11-23
> **Thank you for your constructive review and look forward to post-rebuttal feedback**
>
> Dear Reviewer gJWM,
>
> The weekend is coming or already here! Wish you a nice, pleasant, and relaxing weekend!
>
> We'd like to express our sincere gratitude for your detailed, careful, nice reviews and insightful questions.
>
> We have provided responses to all of your questions and concerns and hope they adequately address them.
>
> As the discussion phase will conclude on November 26th, please don’t hesitate to let us know if you have any further questions. We are dedicated to making every effort to resolve any remaining concerns you may have and sincerely value your suggestions.
>
> Thank you for your time and constructive reviews! We genuinely look forward to your response.
>
> Best regards,
>
> Authors

---

> > ### Author Response · Authors · 2024-12-02
> > **Thank you for your valuable feedback and look forward to post-rebuttal responses**
> >
> > Dear Reviewer gJWM,
> >
> > We would like to kindly remind you that tomorrow is **the last day** of the discussion period. If there are any remaining questions or points you would like us to address, please feel free to let us know. We are fully available to discuss any aspects further.
> >
> > Thank you again for your valuable feedback.
> >
> > Best regards, Authors

---

### Official Review · Reviewer_CSzf · 2024-11-04

**Soundness:** 2
**Presentation:** 1
**Contribution:** 2
**Rating:** 5
**Confidence:** 4

**Summary:**

The authors propose a neural IK controller for controlling dexterous robot hands. The results are validated in both simulated environments and the real world. The authors claim the proposed IK controller can achieve better tracking results than existing approaches.

(12/11/2024 - The score was updated to a five after discussion with other reviewers)

**Strengths:**

- **Real-world experiments are valuable**. I appreciate the authors' efforts to conduct real-world validations of their approach.
- **Baseline comparisons add value**. I appreciate the authors' efforts to compare against DGrasp and PPO baselines.

**Weaknesses:**

**A. Presentation**.

The paper's presentation is substantially below the ICLR standard in figure and writing. I had a very difficult time understanding the paper.

The teaser figure is confusing. First, it is too dense for the readers to quickly catch what the paper is doing. There are simply too many display items in the teaser figure, and the readers could be overwhelmed. More importantly, **I could not comprehend the input and output behavior of the work from the teaser figure**. I scrolled to the method section just to understand what the paper is tackling.

Now, the second figure of the method is just as dense. I don't know what viewing order the authors want to present, but it left me confused about the most essential input and output behavior of the method.

Regarding writing, the authors had trouble situating their contribution and the problem they are solving using the conventional language in robotics. **The authors could have communicated very clearly that they were building an inverse kinematics controller that takes Euclidean space coordinates as input and outputs joint space position-based commands, in just a few sentences in the abstract, introduction, and figures.**

Next, there is a mixture of overuse and incorrect usage of terms from deep (reinforcement) learning, multi-body dynamics, control theory, and robotics in general. This leads me to question the credibility of this work.

I already provided one example of the authors' clarity in communicating their input-output behavior.  Another example I will give is the use of "robustness of tracking controller" by the authors. It can be easily confused with robustness analyses from control theory in the context of trajectory tracking and IK, e.g., using toolsets from Lyapunov Analysis. This, in my view, irresponsible use of terms is also evidenced in line 176, when the authors casually throw out words like "adaptability," "generalizability," and "robustness" without grounding these properties in any computational ways, leaving the readers to interpret themselves. These terms should not be easily thrown into publications without computational interpretations. Another example - what are the "hard-to-track" trajectories mentioned in Figure 2 vs. the "Parent" trajectory? Can the authors computationally state what makes them hard to track?

Whether the approach can handle second-order transients is also not communicated effectively. The proposed approach seems to eventually use a PID-like controller to command the included robotic systems. I could not assess this part due to a lack of descriptions. Can the system perform torque-level control, which is crucial for contact-rich manipulations?

**B. Confusing related work section, Missing critical experiments**.

I am currently unconvinced and cannot fully assess the performance of the work. The related work section mentions the sample inefficiency problem of RL. Yet, the evaluation protocol doesn't ablatively analyze how the proposed approach can generalize other than a very fuzzy success rate criterion in Figure 5.

I am worried that this work will not inform the community why the proposed approach brings benefits. The proposed approach is a very complicated pipeline, piecing together existing toolsets. First, tracking is a highly continuous task and should not be solely evaluated in a single scalar of success rate when the authors want to show that their approach is more "sample-efficient." Together with the presentation issues I have listed above, I could not understand the take-home message of the work to the robotics community. What limitations of existing controllers (e.g., OmniGrasp & DGrasp) are addressed by this pipeline?

**Questions:**

Please see my comments in the weakness section. Thanks.

---

> ### Author Response · Authors · 2024-11-19
> **Author Response (Part 1 of 13)**
>
> Dear Reviewer CSzf,
>
>
>
> Thank you so much for your careful and detailed review. We sincerely appreciate your recognition of our experimental evaluations in both simulation and real-world settings. Your constructive feedback on our presentation, including the figures and terminology, is highly valuable. We also thank you for raising concerns regarding our contributions, the rationale behind our method, and its distinction from prior work.
>
> In this response, we first address your concerns on the presentation and detail the improvements we've made in the revision. We then clarify our contributions, elaborate on the rationale behind our method, and provide detailed responses to your questions. We hope our explanations and revisions can adequately address your concerns.
>
>
>
>
>
> ## Presentation: Figures, Terminologies, Concepts
>
> ### Teaser figure and method figure
>
> > **The teaser figure is confusing...too dense...too many display terms...could not comprehend the input and output behavior..."**
>
> We sincerely thank you for your valuable feedback on our teaser figure. Below we explain what we want to convey in the original teaser figure, and the efforts we've made in the revised version to improve its clarity.
>
> - **Input and output**: The tracking controller takes the kinematic robotic hand and object manipulation trajectory as input and outputs the predicted action trajectory for the robotic hand. Since "actions" are difficult to visualize, we instead present the tracking results: the actual hand and object states achieved when the robotic hand is controlled using these actions. This approach is common in related works such as [OmniGrasp], [DGrasp], [MaskedMimic], [PHC], [DTC], and [DeepMimic]. The primary aim of the original teaser is not to elaborate on the problem setting but to showcase ***the capabilities of our tracking controller*** from *various perspectives*, as demonstrated in notable works like [SDEdit], [SDDIM], [MaskedMimic], [BiLoco], [PenSpin], [HORA], and [GeneRot], some of which are quite famous.
> - **Revised teaser:** We appreciate the reviewer’s emphasis on clearly communicating the problem setting and the model’s input/output behavior in the teaser, as this ensures readers quickly understand the problem we aim to address. While we initially used "kinematic references" to represent the input and "results" for the output, we've realized that these terms may lack clarity. We've updated the teaser in the revision:
>   - Added a subfigure (**Fig. 1 (a)**) to *illustrate the workflow of the neural tracking controller* at each timestep. The controller takes the current observation—composed of the current state, goal state, and object geometry—as input and outputs the action. Applying this action to the robotic hand leads the hand and object to transition to the next state. The kinematic reference trajectory provides the tracking goal state at each timestep.
>   - Replaced "robustness" with a more concrete description for the example in the teaser figure: *"Tracking noisy interactions with unreachable goal states."*
>
> Thank you again for your feedbacks on our teaser figure. We sincerely hope you find the current version more clear and satisfactory.
>
>
>
> *[OmniGrasp] Luo, Zhengyi et al. “Grasping Diverse Objects with Simulated Humanoids.” ArXiv abs/2407.11385 (2024): n. pag.*
>
> *[DGrasp] Christen, Sammy Joe et al. “D-Grasp: Physically Plausible Dynamic Grasp Synthesis for Hand-Object Interactions.” 2022 IEEE/CVF Conference on Computer Vision and Pattern Recognition (CVPR) (2021): 20545-20554.*
>
> *[SDEDit] Meng, Chenlin et al. “SDEdit: Guided Image Synthesis and Editing with Stochastic Differential Equations.” International Conference on Learning Representations (2021).*
>
> *[SDDIM] Lukoianov, Artem et al. “Score Distillation via Reparametrized DDIM.” ArXiv abs/2405.15891 (2024): n. pag.*
>
> *[MaskedMimic] Tessler, Chen et al. “MaskedMimic: Unified Physics-Based Character Control Through Masked Motion Inpainting.” ArXiv abs/2409.14393 (2024): n. pag.*
>
> *[BiLoco] Li, Zhongyu et al. “Reinforcement Learning for Versatile, Dynamic, and Robust Bipedal Locomotion Control.” ArXiv abs/2401.16889 (2024): n. pag.*
>
> *[PenSpin] Wang, Jun et al. “Lessons from Learning to Spin "Pens".” ArXiv abs/2407.18902 (2024): n. pag.*
>
> *[HORA] Qi, Haozhi et al. “In-Hand Object Rotation via Rapid Motor Adaptation.” Conference on Robot Learning (2022).*
>
> *[GeneRot] Chen, Tao et al. “A System for General In-Hand Object Re-Orientation.” ArXiv abs/2111.03043 (2021): n. pag.*
>
> *[PHC] Luo, Zhengyi et al. “Perpetual Humanoid Control for Real-time Simulated Avatars.” 2023 IEEE/CVF International Conference on Computer Vision (ICCV) (2023): 10861-10870.*
>
> *[DTC] Jenelten, Fabian et al. “DTC: Deep Tracking Control.” Science Robotics 9 (2023): n. pag.*
>
> *[DeepMImic] Peng, Xue Bin et al. “DeepMimic.” ACM Transactions on Graphics (TOG) 37 (2018): 1 - 14.*

---

> ### Author Response · Authors · 2024-11-19
> **Author Response (Part 2 of 13)**
>
> > **"...figure of the method is just as dense...what viewing order ...confused about the most essential input and output behavior..."**
>
> Thanks for your feedback. Thank's quite valuable to us. We acknowledge that the original method figure may not have been the most effective in presenting readers with a quick and clear overview of our method. To address this, we have revised the figure to make it more concise and comprehensible. Below, we briefly explain the design logic of the original figure and the key improvements made in our revision.
>
> - **Original design logic:** Our initial aim was to present not only the overall structure of the method but also the *necessary details* of each component. This approach was intended to help readers understand both the working flow and the internal mechanics of each component and their interactions. To achieve this, we carefully designed the layout to include every critical detail. However, we have realized that this level of detail may compromise clarity, especially for readers trying to grasp the method quickly. Thus, we have revised it.
> - **Revised method figure:** The updated figure adopts a more abstract design, focusing on key ideas rather than exhaustive details. Specifically, it emphasizes the *data flywheel*—the iterative bootstrapping framework that alternates between improving the tracking controller through high-quality, diverse demonstration data and enhancing demonstrations using the tracking controller. Key revisions include:
>   - **Highlighting the data flywheel:** The revised figure prominently features the iterative interaction between training the tracking controller and refining tracking demonstrations.
>   - **Streamlining the framework:** The framework is organized into two parts:
>     1. The **left part** outlines the single trajectory tracking scheme, showcasing how the tracking controller improves the quality and diversity of demonstrations through the homotopy optimization scheme.
>     2. The **right part** highlights the workflow for training the tracking controller, abstracting unnecessary details and focusing on the skeleton of each process.
> - **Placement of the original method figure:** To retain the comprehensive details of the original design, we have redesigned the figure and included it in *Appendix A*. This detailed figure is intended to help readers, after reviewing the full method, gain a deeper understanding of how individual components work and how the various parts of the framework interact.
>
> We would like to express our heartfelt gratitude once again for your thoughtful feedback on our method figure. We sincerely hope that you find the revised version clearer and easier to follow. If there are any aspects that you feel could be further improved or if you have additional comments, please do not hesitate to let us know. We would be more than happy to make further refinements to enhance clarity and effectiveness.

---

> ### Author Response · Authors · 2024-11-19
> **Author Response (Part 3 of 13)**
>
> ### Terminologies, concepts, and a clarification of the problem setting
>
> > **"...a mixture of overuse and incorrect usage of terms from deep (reinforcement) learning, multi-body dynamics, control theory, and robotics in general..."**
> > **"...robustness of tracking controller...robustness analyses  from control theory...using toolsets from Lyapunov Analysis"**
> > **"...casually throw out words like 'adaptability,' 'generalizability,' and 'robustness' without grounding these properties in any computational ways...These terms should not be easily thrown into publications without computational interpretations."**
>
> Thanks for the reviewer’s feedback on these terms. We are sorry for any confusion they may have caused. However, we would like to clarify that the issue arises not from misuse or incorrect application but rather from cross-disciplinary differences in the interpretation of their meanings. To address the reviewer’s concerns, we first provide an explanation of the *widely acknowledged meanings* of these terms within the robot learning community and their typical usage in the literature. We've made two lines of effort in our revision to avoid further confusion, including adding *a terminology paragraph* that explains their meanings and *further computational interpretations and analysis* of these concepts.
>
> **"Robustness", "adaptability", and "generalizability" of a neural controller in the robot learning community:** We deeply appreciate and respect the rigorous definitions of controller properties such as "robustness", along with their computational and mathematical foundations in robotics and control theory. Within the robot learning community, neural controllers—where neural networks are trained to take observations as input and output agent actions—are also expected to exhibit certain properties. Some of these properties, like "robustness", are adapted from robotics, while others, such as "generalization ability" ("generalizability"), have emerged specifically due to the use of neural networks. These properties, whether adapted or newly introduced, have acquired well-recognized meanings in the robot learning field and are widely used in the literature [BiLoco, HORA, VMP, UniDexGrasp, OmniGrasp, PHC, DTC, GenRot]. In the following text, we will elaborate on the meanings of these properties in robot learning, provide examples of their usage, and reference relevant analyses from established works to justify the reasonableness of our usage and analysis.
>
> **1) Robustness:** In the robot learning domain, robustness refers to a neural controller's ability to perform reliably under unexpected situations and disturbances [BiLoco, HORA, VMP, DTC]. For instance, in [VMP], a reinforcement learning-based humanoid motion tracking framework demonstrates robustness by handling infeasible inputs such as unreachable states (*e.g.*, a humanoid jumping into the air) and input discontinuities (*e.g.*, unsmooth kinematic references, as shown in Fig. 5 of [VMP]). Our work demonstrates the robustness of our neural controller in handling challenging scenarios, including unreachable states where the object moves unexpectedly (e.g., flying off up from the hand, as shown in Fig. 1 and Fig. 3 of our paper). This is analogous to evaluating robustness against infeasible inputs in [VMP]. Additionally, as seen in the videos on our [project website](https://projectwebsite7.github.io/gene-dex-manip/), the input kinematic references exhibit obvious discontinuities between adjacent frames, such as jittering fingers (e.g., unnatural thumb motion in the first case video involving a shovel). Despite these irregularities, our approach maintains stability and produces reliable outputs. For manipulation tasks, our method also shows resilience to substantial kinematic noise, such as hand-object penetrations in the reference trajectory (e.g., Fig. 4a and 4c, "Using a short shovel," on our [website](https://projectwebsite7.github.io/gene-dex-manip/)). Even under these conditions, our controller remains not affected.

---

> ### Author Response · Authors · 2024-11-19
> **Author Response (Part 4 of 13)**
>
> **Analysis of the robustness:** In Section 4.3, we analyze the robustness of our neural controller using qualitative evaluations. Given the challenges in mathematically analyzing neural controllers, presenting qualitative results and comparisons is a widely adopted approach in the literature to demonstrate "robustness". For example, [VMP] focuses on qualitative robustness analysis, while [BiLoco] emphasizes similar evaluations (see Fig. 10, 11, 12, and Section IX). Similarly, other works, such as [HORA], also utilize this strategy to showcase robustness.
>
> **The usage of the term "robustness":** The term "robustness" is widely used in the literature to describe the ability of a neural controller or policy, often without formal or computational definitions. To illustrate this, we provide examples from previous works:
>
> - **[BiLoco]**: "significant *robustness* to unexpected disturbances," "how to effectively structure the learning process to harness these advantages, such as adaptivity and *robustness*," and "enhance adaptivity and *robustness* in *RL-based* locomotion control."
>
> - **[HORA]**: "This paradigm has enabled *robust* and adaptive locomotion policies," and "This gives a policy which is *robust*, instead of adaptive, to all shape and physical property variations."
>
> **2) Adaptivity:** The adaptivity of a neural controller refers to its ability to adapt to environmental changes, such as time-invariant shifts in dynamics and time-varying contact events [BiLoco, HORA]. In [BiLoco], the authors demonstrate adaptivity to changing contact events, like walking on various terrains, as well as adaptability to dynamics, such as transferring their policy from a simulator to a real-world robot. Similarly, in our work, we showcase adaptivity to contact events, such as tracking trajectories with shifting contacts (e.g., clips of *playing with the banana and flute* in the `demo_video` on our [website](https://projectwebsite7.github.io/gene-dex-manip/) and supplementary materials). Additionally, we demonstrate the policy’s adaptivity to dynamic parameters by deploying it from the simulator to the real LEAP hand.
>
> **The usage of the term "adaptivity":** Like "robustness," "adaptivity" is widely recognized in the robot learning community and is often used without formal or computational definitions. To illustrate this, we provide examples from previous works:
>
> - **[HORA]**: "it to *adaptively* manipulate a diverse set of objects," "train an *adaptive* policy with it as an input," "an *adaptive* and smooth finger gait emerges from the learning process," "we instead use model-free reinforcement learning to train an *adaptive* policy."
>
> - **[BiLoco]**: "This ablation study highlights the *adaptivity* of the proposed controller," "The study also delves into the *adaptivity* and robustness introduced by the proposed RL system," "enhance *adaptivity* and robustness in RL-based locomotion control," "which enhances its *adaptivity* to uncertain dynamics and external perturbations."
>
> **3)** **Generalization ability** is a key concept in machine learning, with a rigorous mathematical definition in learning theory [Gene1, Gene2, Gene3, Gene4]. Training a neural network to achieve high generalization, particularly out-of-domain generalization, is a long-standing challenge and critical for enhancing the network’s applicability in real scenarios. For neural-based dexterous manipulation policies, **generalizing to new objects and input commands** is essential and has been a major focus, as seen in works like the generalizable grasping policy in [UniDexGrasp] and the trajectory-following policy in [OmniGrasp]. In our work, we also aim to achieve these two types of generalization: adapting to different objects and unseen manipulation trajectories.
>
> **The usage of the term "generalization ability"/"generalizability":** "Generalization ability" and "generalizability" are widely recognized concepts in machine learning and are often used without formal or computational definitions. To illustrate this, we provide examples from previous works:
>
> - **[OmniGrasp]**: "following object trajectories and *generalizing* to unseen objects," "presents a scalable approach that *generalizes* to unseen object shapes and trajectories," and "our *generalization* to unseen objects."
>
> - **[UniDexGrasp]**: "our highly *generalizable* goal-conditioned grasping policy," "and *generalize* across hundreds of categories," and "can *generalize* to novel object instances."
>
> - **[HORA]**: "which are critical to our *generalization* ability," "we instead use model-free reinforcement learning to train an adaptive policy, and use adaptation to achieve *generalization*," "Our approach focuses on *generalization* to a diverse set of objects and can be trained within a few hours," and "but also gives a much better *generalization to out-of-distribution* object parameters..."

---

> ### Author Response · Authors · 2024-11-19
> **Author Response (Part 5 of 13)**
>
> **Computational interpretations for robustness, adaptivity, and generalization ability:** In addition to the conceptual definitions, we have followed the reviewer's suggestions to provide computational interpretations and quantified analyses of these properties. These updates are included in the revised version, and further details can be found in *Appendix C*.
>
> **Our revisions**: To prevent cross-disciplinary confusion, we have included explanations of key concepts from the robot learning community in the revised paper (in the first paragraph of the method section). Additionally, we have provided computational interpretations and analyses of these concepts in the Appendix. Given the difficulty of mathematically analyzing the properties of a ***neural*** tracking controller, we quantify these concepts based on its performance across different test sets. It is important to note that, as there is no universally accepted computational interpretation for these concepts in robot learning, our definitions and analyses only reflect our own perspective on the tracking problem.
>
> - We have added a **"Terminologies and notations"** section at the beginning of the method, similar to [BiLoco]. In this section, we provide conceptual definitions for key terms, including **"robustness," "generalization ability,"** and **"adaptivity."**
> - We have attempted to provide computational interpretations of concepts such as **"robustness", "generalization ability", and "adaptivity"** in *Appendix C*. However, even with these quantified concepts, it is important to note that, as a neural controller requiring training, we can hardly provide formal mathematical proof of whether our method is guaranteed to be **"robust", "generalizable"**, or exhibit high **"adaptivity"**. The theoretical framework for deep neural networks is still largely limited to simpler architectures and traditional tasks, such as image classification, which remains behind the more advanced experimental developments in the field.

---

> ### Author Response · Authors · 2024-11-19
> **Author Response (Part 6 of 13)**
>
> > **"...what are the "hard-to-track" trajectories mentioned in Figure 2 vs. the "Parent" trajectory"**
> >
> > **"Can the authors computationally state what makes them hard to track?"**
>
> Thanks for the reviewer's feedback. In the following text, we will clarify the specific meanings of these concepts, provide a computational interpretation of the term "hard-to-track", and summarize the revisions we have made to address any potential confusion.
>
> - **"hard-to-track" trajectories** refer to challenging trajectories that are difficult for RL-based tracking algorithms to solve directly. These trajectories are both influenced by the inherent difficulty of the kinematic reference trajectory and are closely related to the problem-solving capabilities of the RL-based single trajectory tracking method. A "hard-to-track" trajectory may involve complex object geometries, such as thin shovels, or intricate kinematic motions, like in-hand manipulations with fingers (as discussed in Section 4.2). The challenge in a trajectory is not solely due to its complexity, but also depends on the capability of the tracking method used. For example, in the **"A homotopy optimization scheme"** section 3.2, we highlight that such trajectories are tracking problems that cannot be solved using the basic **"RL-based single trajectory tracking"** method.
>
> - **"Parent" trajectory** is an important concept in our homotopy optimization scheme. As stated in section 3.2, if the trajectory $B$ is trajectory $A$'s parent trajectory, we mean that solving the tracking task of $B$ and using its tracking results as the baseline trajectory in the per-trajectory tracking policy of $A$, we can ultimately solve the tracking problem for $A$ more effectively than by directly solving it using $A$'s kinematic hand trajectory as the baseline.
>
> - **Computational interpretation of "hard-to-track":** The most direct quantification is defining a score related to the tracking method's performance. Besides, we can use some statistics of the hand and object kinematic trajectories to quantify the "hard-to-track" characteristic. Here we introduce three types of statistics: **1)** object movement smoothness: it quantifies the motion smoothness by calculating the per-frame average object accelerations, **2)** hand-object contact shifting velocity: it quantifies the per-frame velocity of the contact map change, and **3)** the object shape score: it is the z-axis extent of the object's bounding box to quantify the shape of the object. (Since we frequently encounter "math errors" when writing formulas in OpenReview's text editor, please refer to *Appendix C* for details). We can jointly use these three trends of scores to quantify the "hard-to-track" characteristic.
>
> - **Our revisions:** In the original method figure, we used the term "hard-to-track" to provide an intuitive description of these trajectories. We now recognize that this may cause confusion and have removed it from the revised figure. Similarly, we initially used the term "parent" trajectory, as defined in Section 3.2. However, we agree that this term may not be clear to readers without the proper context, and we have removed it from the revised version.
>
>   Besides, we have incorporated the computational interpretations into *Appendix C*. Furthermore, we have included the per-tracking trajectory average statistics for the three metrics calculated on the test sets in the revised paper.

---

> ### Author Response · Authors · 2024-11-19
> **Author Response (Part 7 of 13)**
>
> > **"...had trouble situating their contribution and the problem they are solving using the conventional language in robotics."**
> >
> > **"...they were building an inverse** **kinematics controller that takes Euclidean space coordinates as input and outputs joint space position-based commands..."**
>
> Thanks for the reviewer’s effort in summarizing the problem setting of our work. We are sorry but we could not fully grasp what you mean by **"IK controller"**. If possible, could you kindly provide a detailed definition? We will proceed with our response based on our current understanding.
>
> **Our understanding of the "IK controller":** Based on our basic knowledge of inverse kinematics, as well as *information from sources such as Google and GPT-4o*, we interpret the "IK controller" as a controller designed to compute the joint configurations required for an agent to reach a specific position and orientation in space. Once the joint configuration is calculated, it is treated as the desired target to achieve in joint space. Subsequently, torques applied to the joints can be computed using techniques like PID control with additional strategies such as feed-forward compensation, gain scheduling, adaptive control, or model predictive control (MPC).
>
> **Difference from IK in our problem setting:** If our understanding of the "IK controller" aligns with the reviewer, we would like to clarify some key differences in our problem setting:
>
> - **Controller’s duty:** Our controller does not aim to solve an inverse kinematics (IK) problem. In fact, the tracking targets for the robotic hand are already provided in the joint space.
> - **Problem setting:** Given a kinematic reference trajectory that includes a sequence of robotic hand states in joint space and object pose states, our tracking controller generates an action trajectory for the robotic hand. This action trajectory consists of per-joint positional target trajectories, with the goal of controlling the hand to track the reference kinematic trajectory.
> - **Related literature:** The task of tracking is widely studied in the fields of robot learning and physics-based character animation, as demonstrated by works such as [VMP, BiLoco, DTC, PHC, PULSE, QuasiSim, OmniGrasp], including for dexterous hand control [QuasiSim, OmniGrasp].
> - **Core challenge:** The difficulty in our tracking problem does not lie in solving IK for joint space states. Rather, the challenge is in inferring control commands—such as per-joint positional targets—from both joint space states and object states that the system aims to achieve.
>
> If there are any misalignments between our interpretation of the "IK controller" and the meaning intended by the reviewer, we wish for a kind correction. We are more than willing to engage in further discussions and will make every effort to address any concerns raised.
>
>
> > **"...handle  second-order transients...PID-like controller ...could not assess this part...Can the system perform torque-level control..."**
>
> Thank you for your detailed question regarding our control strategy.
>
> - **Low-level PD control:** As stated in the "Reinforcement learning" paragraph in Section 3.1, we use a **PD controller** to control the robotic hand:
>
>   > "We control the robotic hand using a proportional derivative (PD) controller, following previous literature (**Luo et al.,2024; 2023b; Christen et al.,2022; Zhang et al., 2023**)."
>
>   The low-level PD controller receives positional target commands (output from our neural tracking controller) as input to calculate appropriate joint forces/torques.
>
> - **Independence from low-level controller:** Our method and contributions do not depend on the specific low-level controller used. The core design of our method, including the learning scheme and the data feedback loop, can still be applied to train a neural tracking controller with a different low-level controller. For example, if we were to replace the current PD controller with a torque-level controller, we would only need to modify the action space of our neural tracking controller to align with the input requirements of the torque-level controller.
>
> - **Reason for using a low-level PD controller:** We selected the PD controller because, compared to torques, positional targets are more transferable across simulators and from simulation to real-world applications [Sim2Real, Sim2Real2]. This facilitates the sim-to-real transfer process.
>
>   Additionally, the choice of action space would also influence the training of a generalizable tracking controller. Using torque-level controls may limit the policy’s ability to learn shared tracking knowledge across different trajectories, hindering generalization.

---

> ### Author Response · Authors · 2024-11-19
> **Author Response (Part 8 of 13)**
>
> We would like to express our sincere gratitude once again for your thoughtful feedback. In response, we have made corresponding revisions in the revised version to address potential confusion arising from the different usages of terms across disciplines. Please let us know if you have any concerns or comments. We will spare no effort in improving the paper and addressing your concerns.
>
>
>
>
>
> *[VMP] Serifi, Agon et al. “VMP: Versatile Motion Priors for Robustly Tracking Motion on Physical Characters.” Computer Graphics Forum (2024): n. pag.*
>
> *[BiLoco] Li, Zhongyu et al. “Reinforcement Learning for Versatile, Dynamic, and Robust Bipedal Locomotion Control.” ArXiv abs/2401.16889 (2024): n. pag.*
>
> *[HORA] Qi, Haozhi et al. “In-Hand Object Rotation via Rapid Motor Adaptation.” Conference on Robot Learning (2022).*
>
> *[Gene1] Brennan, R. L. (2001). Generalizability Theory. New York: Springer-Verlag.*
>
> *[Gene2] Crocker, L., & Algina, J. (1986). Introduction to Classical and Modern Test Theory. New York: Harcourt Brace.*
>
> *[Gene3] Shavelson, R.J., & Webb, N.M. (1991). Generalizability Theory: A Primer. Thousand Oaks, CA: Sage.*
>
> *[Gene4] Shalev-Shwartz, Shai and Shai Ben-David. “Understanding Machine Learning - From Theory to Algorithms.” (2014).*
>
> *[UniDexGrasp] Xu, Yinzhen et al. “UniDexGrasp: Universal Robotic Dexterous Grasping via Learning Diverse Proposal Generation and Goal-Conditioned Policy.” 2023 IEEE/CVF Conference on Computer Vision and Pattern Recognition (CVPR) (2023): 4737-4746.*
>
> *[OmniGrasp] Luo, Zhengyi et al. “Grasping Diverse Objects with Simulated Humanoids.” ArXiv abs/2407.11385 (2024): n. pag.*
>
> *[QuasiSim] Liu, Xueyi et al. “QuasiSim: Parameterized Quasi-Physical Simulators for Dexterous Manipulations Transfer.” ArXiv abs/2404.07988 (2024): n. pag.*
>
> *[QuasiStatic] Pang, Tao et al. “Global Planning for Contact-Rich Manipulation via Local Smoothing of Quasi-Dynamic Contact Models.” IEEE Transactions on Robotics 39 (2022): 4691-4711.*
>
> *[ComplementaryFree] Jin, Wanxin. “Complementarity-Free Multi-Contact Modeling and Optimization for Dexterous Manipulation.” ArXiv abs/2408.07855 (2024): n. pag.*
>
> *[DIAL-MPC] Xue, Haoru et al. “Full-Order Sampling-Based MPC for Torque-Level Locomotion Control via Diffusion-Style Annealing.” ArXiv abs/2409.15610 (2024): n. pag.*
>
> *[GeneRot] Chen, Tao et al. “A System for General In-Hand Object Re-Orientation.” ArXiv abs/2111.03043 (2021): n. pag.*
>
> *[PHC] Luo, Zhengyi et al. “Perpetual Humanoid Control for Real-time Simulated Avatars.” 2023 IEEE/CVF International Conference on Computer Vision (ICCV) (2023): 10861-10870.*
>
> *[DTC] Jenelten, Fabian et al. “DTC: Deep Tracking Control.” Science Robotics 9 (2023): n. pag.*
>
> *[PULSE] Luo, Zhengyi et al. “Universal Humanoid Motion Representations for Physics-Based Control.” ArXiv abs/2310.04582 (2023): n. pag.*
>
> *[Sim2Real] Zhao, Wenshuai et al. “Sim-to-Real Transfer in Deep Reinforcement Learning for Robotics: a Survey.” 2020 IEEE Symposium Series on Computational Intelligence (SSCI) (2020): 737-744.*
>
> *[Sim2Real2] Chebotar, Yevgen et al. “Closing the Sim-to-Real Loop: Adapting Simulation Randomization with Real World Experience.” 2019 International Conference on Robotics and Automation (ICRA) (2018): 8973-8979.*

---

> ### Author Response · Authors · 2024-11-19
> **Author Response (Part 9 of 13)**
>
> ## **Contributions, differences from previous works, evaluation process, and performance**
>
> ### Contributions, differences from previous works
>
> > **"I am currently unconvinced and cannot fully assess the performance of the work..."**
> >
> > **"...worried that this work will not inform the community why the proposed approach brings benefits. The proposed approach is a very complicated pipeline, piecing together existing toolsets."**
> >
> > **"... could not understand the take-home message of the work to the robotics community. What limitations of existing controllers (e.g., OmniGrasp & DGrasp) are addressed by this pipeline?"**
>
> We are sorry if the cross-disciplinary differences in the meanings of these terms have caused confusion, impacting the communication of our key ideas, contributions, and the rationale behind our method design. To address these concerns, in the following text, we will summarize our contributions, method design rationale, take-home message, and what makes us distinct from prior works. We'll answer your specific questions as we clarify these points.
>
> **Terminologies:** Please allow us to make clear the terminologies we will use in the following text: *Tracking task* refers to the task of imitating a given manipulation trajectory. *Versatile tracking policy/controller* refers to a policy or a *neural* controller capable of handling a diverse range of manipulation tracking tasks (the term *versatile* is borrowed from [BiLoco]). *Generalization ability* refers to the policy's capacity to track novel interaction trajectories involving new objects and manipulation types. A manipulation trajectory is considered *complex/difficult* for a tracking controller if it involves challenging objects, non-trivial object movements, or manipulations (e.g., beyond basic grasping, pick-and-place, or trajectory following without significant contact variations).
>
> **Key contributions:** To the best of our knowledge, we are the first to develop *a versatile and generalizable dexterous manipulation neural controller* capable of handling *a diverse set of non-trivial manipulation skills*. These tasks are defined by *per-frame dense goals* that involve *thin* objects, *complex* object movements, and *subtle in-hand reorientations*—tasks that go beyond simpler grasping, pick-and-place, or trajectory following without significant contact variations. As acknowledged by other reviewers, our work is recognized for *"tackling an important problem of learning a tracking controller for dexterous manipulation..."* (Reviewer 4aCE), *"...representing a solid advancement toward solving the challenging task of dexterous hand manipulation, a significant objective in robotics..."* (Reviewer gJWM), and *"...offering solid contributions, including a novel approach and substantial experimental results, with a well-motivated generalizable tracking controller for dexterous manipulation..."* (Reviewer 4aMi).

---

> ### Author Response · Authors · 2024-11-19
> **Author Response (Part 10 of 13)**
>
> In the problem-solving aspect, compared to prior works such as [OmniGrasp] and [DGrasp] mentioned by the reviewer, our contributions address two main dimensions that are central to *robot learning*:
>
> - **Manipulation skill difficulty:** The most difficult manipulation skills presented in previous literature (to the best of our knowledge) mainly lie in two types:  1) **Manipulation complexity**: This includes in-hand manipulations such as in-hand re-orientations (e.g., [HORA], [ReOrnt]), pen spinning (e.g., [PenSpin], [Eureka]), or dynamic hand-over with contact variations. However, these works can only deal with a single specific goal, such as rotating the object along the z-axis for 90 degrees, and they cannot handle time-varying commands. This limits their ability to accomplish tasks with rich semantics that cannot be described by a single goal. Therefore, a new type of complexity arises: *goal complexity*. 2) **Goal complexity**: Beyond the single or sparse goal-driven tasks, dexterous manipulations also involve tasks with rich semantics, such as writing a sentence, cleaning a bowl using brushes, or cutting vegetables with a knife. These tasks require dynamic handling of time-varying goals.
>
>   **Previous works**: [OmniGrasp] demonstrated the ability to follow a trajectory with dense per-frame object and hand states as goals but did not handle in-hand manipulations. Once an object is grasped, hand-object contacts would not change, limiting its ability to track trajectories with intricate in-hand manipulations involving dynamic contact variations. On the other hand, [DGrasp] focused on simpler tasks, primarily re-orienting an object to a final pose, without considering per-frame dense goals or in-hand manipulations.
>
>   **Our problem setting**: In contrast, our work aims to address manipulation problems that combine both high **manipulation complexity** (e.g., intricate in-hand re-orientations, dynamic contacts, or subtle object motions) and **goal complexity** (e.g., tasks involving rich semantics with time-varying goals). For example, our tasks include playing with a flute or banana with dynamic contacts, or waving a shovel involving subtle in-hand reorientations using fingers. These tasks cannot be described by a single goal and require robust tracking of complex manipulation trajectories.
>
> - **Generalization ability**: Humans are dexterous manipulation generalists, meaning we can master a variety of diverse skills and perform new tasks in a zero-shot manner—e.g., opening a box we’ve never seen before or placing a stuffed toy we’ve never interacted with into it. The ability to generalize (solving new tasks) and be versatile (mastering a diverse set of skills) has long been a goal in the robot learning community. **Previous work** demonstrated some level of versatility and generalization on relatively simple tasks, such as trajectory following in [OmniGrasp] or prehensile grasping in [UniDexGrasp].
>
>   **Our aim**: We take this a step further by developing a tracking controller that achieves both **generalization ability** and **versatility on more complex manipulation tracking tasks**. These tasks involve handling thin objects, changing contacts, and performing subtle in-hand re-orientations. Our work aims to push the boundaries of generalization and versatility in the context of more difficult and varied manipulation tasks than those tackled in prior work.
>
> **Core techniques:**  1) We develop a technique for curating a large-scale set of high-quality tracking demonstrations, composed of paired kinematic references and robot actions, to train a tracking controller by carefully integrating Reinforcement Learning (RL) and Imitation Learning (IL).  2) We utilize a data flywheel to iteratively enhance the performance of the tracking controller and the quality and diversity of tracking demonstrations.

---

> ### Author Response · Authors · 2024-11-19
> **Author Response (Part 11 of 13)**
>
> **Rationale behind method designs:**
>
> - **Integrating RL and IL**: The reason for integrating RL and IL techniques to train the neural tracking controller is to combine the strengths of both methods to achieve the goal of a generalizable controller. Solely relying on RL’s trial-and-error approach is inadequate because it struggles with heterogeneous tracking targets that result from the need to master a diverse set of tracking tasks. IL can effectively guide RL’s exploration by providing high-quality demonstrations, enabling the controller to learn faster and scale better with the quality and quantity of tracking data. This integration also improves the controller’s generalization ability, while maintaining robustness to external disturbances. The combination of IL and RL is essential for training a generalizable neural tracking controller.
>
> - **Data flywheel operation**: To make the data flywheel function effectively, we need a scalable per-trajectory tracking scheme that continuously improves the tracking demonstrations as the tracking controller becomes stronger. We devise three techniques:  a) A basic RL-based per-trajectory tracking scheme.  b) A strategy for improving single trajectory tracking by using the tracking controller to initiate its tracking policy.  c) A homotopy optimization scheme that explores cross-task relations to improve single trajectory tracking.  Among these, the most important aspect of the homotopy optimization scheme is finding effective optimization paths. A naive brute-force search is too slow and not feasible for real-world applications. Therefore, we propose a model trained to learn cross-task relations via a data-driven method, which is then used to efficiently generate optimization paths.
>
> **Take-home message**: By formulating dexterous manipulation skill acquisition as a unified manipulation trajectory tracking problem and leveraging abundant human manipulation trajectories as references, we can develop a versatile and generalizable neural tracking controller. This approach presents a promising direction for advancing generalizable and diverse dexterous manipulation skill learning.

---

> ### Author Response · Authors · 2024-11-19
> **Author Response (Part 12 of 13)**
>
> ### Our evaluation process and the performance
>
> > **"I am currently unconvinced and cannot fully assess the performance of the work..."**
> >
> > **"...tracking is a highly continuous task and should not be solely evaluated in a single scalar of success rate..."**
>
> - **Evaluation Protocol:**  Our *main evaluation* focuses on the generalizable tracking performance of our tracking controller.
>
>   - **Tracking performance evaluation:** To evaluate the tracking performance, we introduce **five types of metrics** that measure **per-frame average tracking error** for both the hand and the object. These metrics evaluate the tracking performance across the entire trajectory to assess the controller’s ability to perform a **highly continuous tracking task**, as well as the **overall tracking task success rates** calculated based on the hand and object per-frame average tracking errors. The metrics are summarized as follows:
>
>     - **Per-frame average object tracking error** includes a per-frame average object rotation error $R_{err}$ and a per-frame average object translation error $T_{err}$.
>
>     - **Per-frame average hand tracking error** includes a per-frame average  wrist tracking error $E_{wrist}$ and per-frame average per-finger joint position error $E_{finger}$.
>     - **Success rate** is evaluated using two sets of thresholds for $T_{err}$, $R_{err}$, and $0.5E_{wrist} + 0.5E_{finger}$. The tracking is considered successful only if all three values are below their respective thresholds.
>
>     For details, please refer to **"Metrics"** paragraph in section 4.1.
>
>   - **Tracking generalization ability evaluation:**  We adopt an out-of-distribution test setting to assess the model's generalization ability by evaluating its performance on out-of-distribution test sets. For details regarding dataset splitting, please refer to the **"Datasets"** section in **Section 4.1**.
>
>   **Evaluating robustness and adaptability:** These evaluations are secondary but important for the complete assessment of the model's performance. Initially, we focused on qualitative evaluations (see **Section 4.3** for robustness, and **real-world evaluations** for adaptability). Based on the reviewer’s suggestions, we’ve now quantified these properties and added their quantitative evaluations in **Appendix C**.
>
>   **Ablation studies** focus on evaluating 1) the model's perforamnce w.r.t. the quality of the demonstrations (see paragraph **Diversity and quality of robot tracking demonstrations"** in section 5 for details), and 2) the model's performance regarding the quantity of the demonstrations (see **"Scaling the number of demonstrations"** in section 5 for details). For 1) we presnet comparisons on all five metrics in the main text (Table 1). While for 2), we initially only draw ***the trend of the sucess rate to better illustrate the "scaling" behaviour***. We have added their results on all five metrics in ***Appendix B.1***.
>
> - **Performance:** The main generalizable tracking control evaluation results are summarized in **Table 1**. We compare our performance with strong baselines and also present the ablation study's results. Qualitative results are demonstrated in Figure 4 and on our [website](https://projectwebsite7.github.io/gene-dex-manip/). Our method can obviously outperform baseline methods both quantitatively (10%+ improvement) and qualitatively. Additional generalization evaluations regarding the controller's performance on different levels of test sets are presented in Table 4, B.1. Qualitative results for the robustness and adaptivity are shown in Figure 3,4,6,8, and our [website](https://projectwebsite7.github.io/gene-dex-manip/). We've newly added their quantitative results (Appendix C). To summarize the performance, our generalizable tracking controller can effectively track a diverse set of non-trivial dexterous manipulation trajectories and generalizes to track unseen manipulations, outperforming previous works by a large margin. It remains robust towards large kinematics noises such as unreachable states and penetrations. We've transferred the tracking controller to the real world and shown its better adaptation and performance in real robots compared to best-performed baselines.

---

> ### Author Response · Authors · 2024-11-19
> **Author Response (Part 13 of 13)**
>
> > **"...sample inefficiency problem of RL..."**
> >
> > **"...tracking is a highly continuous task and should not be solely evaluated in a single scalar of success rate when the authors want to show that their approach is more `sample-efficient` "**
>
> - **Sample efficiency** is neither a direct metric for evaluating tracking performance nor a desired property of a neural tracking controller. Instead, it characterizes how many trial-and-error interactions the model needs to improve. We highlight sample inefficiency in the RL algorithm to justify our integration of RL and IL for training the neural tracking controller. "Sample efficiency" is related to the final performance. Thus it is reflected by the tracking performance.
>
> - **Comparisons with pure RL:** The key difference between using pure RL and our approach lies in the final tracking performance. Due to the inherent sample inefficiency of RL, achieving satisfactory results is challenging (as seen in the performance of *PPO (w/o sup., tracking rew.)* in **Table 1, Figure 4**). Our method significantly outperforms it (**Table 1, Figure 4**).
>
> At last, we hope that the explanations provided above regarding the core ideas, contributions in problem-solving and technical aspects, as well as the differences and uniqueness compared to previous works, are clear. Should you have any further questions or concerns regarding our contributions, the distinctions from prior research, or the method design, please do not hesitate to let us know. We are more than happy to address any of your concerns.
>
>
>
>
>
>
>
> *[HORA] Qi, Haozhi et al. “In-Hand Object Rotation via Rapid Motor Adaptation.” Conference on Robot Learning (2022).*
>
> *[ReOrnt] Chen, Tao et al. “A System for General In-Hand Object Re-Orientation.” ArXiv abs/2111.03043 (2021): n. pag.*
>
> *[PenSpin] Wang, Jun et al. “Lessons from Learning to Spin "Pens".” ArXiv abs/2407.18902 (2024): n. pag.*
>
> *[Eureka] Ma, Yecheng Jason et al. “Eureka: Human-Level Reward Design via Coding Large Language Models.” ArXiv abs/2310.12931 (2023): n. pag.*
>
> *[OmniGrasp] Luo, Zhengyi et al. “Grasping Diverse Objects with Simulated Humanoids.” ArXiv abs/2407.11385 (2024): n. pag.*
>
>
>
>
>
> In summary, we deeply appreciate your time and detailed review. Your feedback on our figures, terminology, and concerns regarding our contributions, methods, evaluations, and performance are quite valuable to us. We have tried our best to address your questions and concerns in this response and have reflected these improvements in our revised submission. We hope you find the updated version clearer and more satisfactory than the original.
>
> We would love to have discussions, answer any further questions, and will do our best to address any concerns you may have. Please let us know if there is anything specific we can provide.
>
>
>
> Looking forward to your responses!
>
> Best regards, Authors

---

> > ### Comment · Reviewer_CSzf · 2024-11-21
> > **Reviewer's response (1/2)**
> >
> > I thank the authors for their time and efforts in writing the rebuttal.
> >
> > Unfortunately, I remain unconvinced of this work and would prefer not to change my rating. I want to highlight some of my deepest concerns.
> >
> > # Complicated pipeline and unclear long-term insights
> >
> > In a single sentence, I am afraid that the pipeline is too complicated to produce valuable long-term insights for the robotics community. I would like to briefly talk about how the presented method makes non-trivial assumptions, some of which are not explicitly reported by the authors. This makes me worried about the longer-term insights into the field.
> >
> > **A. On the perception side**, the design is highly complicated. The method requires point cloud encoding of an object. However, the authors do not report how point clouds are processed to obtain objectness in the first place. Is it a partial point cloud similar to the real world, or is it a full point cloud from privileged knowledge of the physics simulator - I do not know.
> >
> > Also, it seems that the proposed method is unable to take on perceptual inputs of the whole scene as the input, e.g., what if the input point cloud encapsulates the surface of the table, the robot hand, the robot arm, and the object?
> >
> > **B. On the physics / optimal control / system dynamics side**, the method implicitly assumes manipulating a single, rigid object. The author does not explicitly make this clear. This assumption is coupled with the author's explicit method design, as the policy network must use the 6-DoF Euclidean frame to represent the goal state.
> >
> > The authors' account of system dynamics is unclear and lacks rigor. Is it assumed that the system's state is always in force equilibrium and quasistatic assumptions are imposed? The authors' report blurs this, as the authors seem to ignore that kinematic references might not be achievable by just a single "action" or control command between two steps. How is the kinematic reference trajectory of both the object and the robot hand obtained in the first place during inference and planning?
> >
> >
> > **C. The data mining procedure (Sec. 3.2 / 3.3) adds more noise rather than making the work elegant.**
> >
> > *The pipeline extends beyond a trained point cloud module and an RL-trained goal-conditioned policy. The authors also propose to train a conditional diffusion model to "transform tasks."* There seem to be many trainable neural networks in this work that could all contribute to weakness in generalizations.
> >
> > I am also concerned with the terminologies, such as "homotopy optimization scheme" and "parent task." None of these are rigorously grounded in theory or computational terms and are marred by an overuse of human intuitions in writing. For example, are there guarantees that the paths generated by the diffusion model constitute homotopy? There was almost no care given to this in writing. The authors then go on to talk about how the homotopy optimization they used is similar to a "chain of thought" in a large language model (L301).
> >
> > This makes me deeply concerned with the writing of that entire section (Sec. 3.2 and Sec. 3.3). **I am worried about using these intuition-driven terms to report scientific findings and whether they actually mean anything to the community in the long term.**
> >
> > It is also very unclear what tasks mean in computational terms. Perhaps the authors are trying to talk about the distribution of a sequence of configurations (q's), which would be the object and hand configurations. It seems that the data-mining section can be more rigorously grounded in either exploration of the theory of RL or optimal control (e.g., how trees of an RRT are grown). Due to trouble understanding this work, I could not make further suggestions in this aspect.

---

> > > ### Comment · Reviewer_CSzf · 2024-11-21
> > > **Reviewer's response (2/2)**
> > >
> > > # Questionable professionalism and credibility in the characterization of the work
> > >
> > > All of the above made me question the professionalism and credibility of this report when I was reading it. I have highlighted that the authors seem to have trouble situating their contributions precisely without resorting to human-intuition-driven terms. I think the revisions needed for this paper are beyond the scope of just this conference rebuttal and would require a resubmission to another venue.
> > >
> > > The authors seem to suggest in their response that the "robot-learning community" might have developed special termonologies that should be allowed for the blurry usage of the terms "adaptability," "generalization," and "robustness." Without going into philosophical conservations, I do not agree with this view. As a robot learning practitioner and part of physical sciences (i.e., robot hardware and objects materialize in the physical world), I think it is essential to communicate clearly about the techniques we develop.
> > >
> > > My review aims to ensure that the papers we publish in the community clearly state what they are tackling and will bring long-term insights. Again, I sincerely thank the authors for their time and efforts in writing the response. I would also like to ask the authors not to resort to GPT answers when writing future responses.

---

> > > > ### Author Response · Authors · 2024-11-22
> > > > **Author Response (Part 1 of 7)**
> > > >
> > > > Dear Reviewer CSzf,
> > > >
> > > >
> > > >
> > > > We thank the reviewer for the responses and the follow-up questions. Below, we respond to concerns raised in the reviewer's new response.
> > > >
> > > >
> > > >
> > > > ## Long-term insights and the framework
> > > >
> > > > ### Long-term insights
> > > >
> > > > We sincerely thank the reviewer for raising this insightful concern. Below, we provide a brief explanation of the long-term insights of our work and how they were presented.
> > > >
> > > > **Long-term insights:**
> > > >  The central long-term insight of our work is leveraging the power of data to address the complex challenge of learning a generalizable tracking controller. Specifically, we emphasized the significance of using large-scale, high-quality robotic tracking demonstrations to tackle the complex problem of learning a generalizable tracking controller for dexterous manipulation. To acquire such tracking demonstrations, the ***key insight on the technical side*** is leveraging the data flywheel that iterates between:
> > > >
> > > > - Training the tracking controller using supervision from high-quality demonstrations.
> > > >
> > > > - Enhancing the quality and diversity of tracking demonstrations—used for further training the controller—through a trajectory-tracking scheme that benefits from the improvements in the tracking controller.
> > > >
> > > > **Presentation of the long-term insights:** The long-term insights are introduced at a high level in the introduction section, providing an overarching perspective. Detailed techniques and methodologies for operationalizing these insights are then thoroughly explained in the methods section.
> > > >
> > > > - **High-level explanation introduction section:**  In the fourth paragraph, after explaining the background, problem setting, and challenges, we highlight the insights of our work at a high level:
> > > >
> > > >   - "*leveraging large-scale, high-quality robot tracking demonstrations...supervise and significantly empower neural controllers...*" highlights an important key of our work: utilizing high-quality demonstrations with expert action trajectories to enhance the capabilities of neural controllers, which are typically trained via reinforcement learning (RL).
> > > >   - "*acquiring large and high-quality...is challenging but we would utilize the data flywheel...*" addresses the challenges in obtaining high-quality tracking demonstrations, which are critical to enabling the proposed learning scheme to truly work. We present our solution that utilizes a data flywheel to iteratively expand and refine these demonstrations.
> > > >
> > > > - **Method section:** We organize the method section to reflect and detailedly explain key insights.
> > > >
> > > >   - **Learning a neural tracking controller from demonstrations:** We elaborate our designs on the learning scheme to train the tracking controller, where we carefully combine reinforcement learning and imitation learning.
> > > >   - **Mining high-quality robot tracking demonstrations:** We detailedly explain how we mine high-quality robot tracking demonstrations continuously to make the data flywheel work. To improve the quality and diversity of tracking demonstrations, we need to go beyond the capability of the basic single trajectory tracking strategy. Therefore, we introduce two techniques to improve the quality and diversity of the tracking controller: 1) leveraging the tracking controller to improve single trajectory tracking, and 2) utilizing a homotopy optimization scheme that leverages cross-tracking task relations to enhance the single trajectory tracking.
> > > >
> > > > - **Experimental designs and evidence:** To demonstrate the role of leveraging large-scale and high-quality data to empower the tracking controller's learning, we carefully design these two trends of experiments beyond main experiments and comparisons:
> > > >
> > > >   - To validate the **importance of the data quality** in the tracking controller's learning and the role of our two techniques proposed for improving the data diversity and quality, we devise two ablated models:
> > > >
> > > >     - Ours (w/o data, w/o homotopy) to validate the role of leveraging the tracking controller and the homotopy optimization to improve the quality of demonstrations used in the imitation learning process,
> > > >     - and Ours (w/o data) to validate the role of leveraging the tracking controller to improve the quality of demonstrations.
> > > >
> > > >     By comparing the performance of Ours (w/p data) with Ours, we can validate the importance of leveraging the tracking controller to improve the tracking demonstration's quality. By comparing the performance of Ours (w/o data, w/o homotopy) with Ours (w/o data), we can validate the importance of leveraging the homotopy optimization scheme to improve the demonstration's quality.
> > > >
> > > >   - To validate **the amount of data** utilized to train the tracking controller, we ablate the data scale, training the model using demonstration data of different scales. By comparing their performance, we can validate the importance of leveraging a large amount of data to train the tracking controller.

---

> > > > ### Author Response · Authors · 2024-11-22
> > > > **Author Response (Part 2 of 7)**
> > > >
> > > > ### Framework
> > > >
> > > > Each component of our framework is essential and plays a critical role in the proposed method. The framework is *not unnecessarily complex*.
> > > >
> > > > Below, we will go through the overall framework via the logic drawn in the current method figure, along with explaining the important role of each component and how it aligns with and reflects the long-term insights of our work.
> > > >
> > > > - **High-level insights -- utilizing a data flywheel to mine high-quality demonstrations and leveraging the demonstrations to empower the training of a tracking controller:** At a high level, to tackle the difficult problem of training a generalizable tracking controller for dexterous manipulations, we leverage high-quality robot tracking demonstrations to empower the tracking controller's learning. However, how to obtain high-quality robot tracking demonstrations itself is difficult. So, we leverage a data flywheel where two important techniques are introduced that utilize the tracking controller and a homotopy optimization scheme so that we can continuously mine higher quality and more diverse data as the tracking controller's capability increases. By iterating between these two modules, we can finally train a strong tracking controller.
> > > > - **Mining high-quality tracking demonstrations:** In the tracking demonstration mining part, we introduce two important techniques: 1) a strategy that can let us improve the single trajectory tracking utilizing the tracking controller, and 2) a homotopy optimization scheme that explores cross-tracking task relations for improving the single tracking, especially for difficult tracking tasks (*e.g.,* which are originally not solvable by the basic single trajectory tracking strategy). We wish to highlight that:
> > > >   - **Importance of these two techniques:** Both of them play a vital role in enabling us to go beyond the capability of RL-based single trajectory tracking and to mine higher-quality and more diverse tracking demonstrations as the controller gradually gets stronger.
> > > >   - **Importance of the data mining part:** The data mining part is vital for us to make the data flywheel work: 1) To facilitate the tracking controller's training, we need abundant and high-quality tracking demonstrations that can provide supervision. 2) To improve the demonstration data's diversity and quality, the above two techniques that can utilize the tracking controller in a homotopy optimization scheme to improve the demonstration data is important. Otherwise, the tracking demonstration's quality would be restricted to the ability of RL-based single trajectory tracking strategy and will not be able to be further improved together with the tracking controller.
> > > > - **Training the tracking controller:** In the tracking controller learning process, to carefully combine RL with IL, we introduce an auxiliary imitation loss that biases the tracking controller's prediction towards expert action trajectories.

---

> ### Author Response · Authors · 2024-11-22
> **Author Response (Part 3 of 7)**
>
> ### Responses to specific questions
>
> > "**On the perception side**... authors do not report how point clouds are processed to obtain objectness..."
> >
> > "...a partial point cloud similar to the real world...full point cloud from privileged knowledge of the physics simulator..."
> >
> > "...unable to take on perceptual inputs of the whole scene as the input..."
>
> - **Problem setting:** We aim at tackling the dexterous manipulation tracking problem, where the whole kinematic trajectory with *the goal hand pose and the goal object pose at each timestep are assumed available*. It aligns with previously published "tracking" work for dexterous hands-object manipulations [QuasiSim, OmniGrasp]. "Tracking" is also a widely studied problem in humanoid robots, physics character animations, and quadrupedal locomotions [DTC, PHC, PULSE, DeepMImic].
>
> - **Usage of object point clouds:** We clarify that, rather than providing direct state information, object point clouds are used to extract object features. These features are then fed into the tracking policy to enable it to recognize *which object is involved in the trajectory* [UniDexGrasp, UniDexGrasp++] (as stated in Section 3.2, line 236, and Appendix A).
>
> - **"partial point cloud" "perceptual inputs of the whole scene"**: In this work, our focus is not on solving the entire dexterous manipulation problem, which involves taking raw observations as input and outputting actions. Instead, we address the "tracking problem", where *given the current state and the goal state*, the objective is to output an action that aims to track the goal.
>
>   Importantly, neither the goal object state nor the current object state is implicitly derived from nor explicitly estimated using object point clouds. Below, we explain how these states are obtained:
>
>   - **Current object states** are available from the simulator. In the real world, we obtain the object state via **Foundation-Pose,** as stated in section 4.1, line 411.
>
>   - **Goal object states** are available from the kinematic trajectory we aim to track.
>
> - **Trajectory source for the tracking problem:** We have lots of methods to acquire such kinematic references for the tracking problem leveraging modern computation vision techniques, *e.g.*, via motion synthesis models [ManiDext, DexHand], reconstruction models [MCC-HO], and the motion capture technique. In this work,  our trajectories that contain object pose trajectory and hand trajectory, as well as the object CAD models, come from MoCap datasets (GRAB and TACO datasets).
>
>   As the vision techniques gradually get developed, the trajectory source would be more and more abundant. It would expand the potential of our tracking control paradigm toward general dexterous manipulation skill learning.

---

> ### Author Response · Authors · 2024-11-22
> **Author Response (Part 4 of 7)**
>
> > "...implicitly assumes manipulating a single, rigid object..."
> >
> > "Is it assumed that the system's state is always in force equilibrium and quasistatic assumptions are imposed?... blurs this...ignore that kinematic references might not be achievable by just a single "action" or control command between two steps. "
> >
> > "How is the kinematic reference trajectory of both the object and the robot hand obtained in the first place during inference and planning?"
>
> - **"Manipulating a single, rigid object":** Yes, our focus is on tracking manipulation trajectories involving a single, rigid object. However, we do not consider this a significant limitation. Single rigid-object manipulation is common in daily life, and in many cases, the manipulation problems we aim to solve fall within this category.
>
> - **Physics and system dynamics:** We do not make any special assumptions regarding the dynamics of the physics involved in the manipulation. In our work, we rely on the physics encoded in the IsaacGym simulator for the simulated environment, while using real-world dynamics in our real robot experiments.
>
>   We do not assume that the system is always in force equilibrium, nor do we impose quasistatic assumptions. *We allow for dynamic and time-varying contacts during the manipulation.* As we do not introduce any additional assumptions on the system dynamics, we have not explicitly stated such assumptions in the manuscript.
>
> - **"...kinematic references might not be achievable by just a single "action"... or control command.."**: This is an insightful observation, as it is true that the kinematic references may not always be achievable with a single control command. The reference trajectory may contain physically implausible states (e.g., unreachable configurations) or be affected by various types of noise, such as jitter or penetration. Consequently, the goal of the tracking is not to match every frame exactly but to bring the resulting states close to the overall reference trajectory.
>
>   Furthermore, *the presence of physically implausible states in the reference trajectory represents one of the challenges of our tracking problem.* Our tracking policy is designed to bring the resulting states close to the reference, aligning with the semantics specified by the trajectory, while remaining robust to implausible states and noise. As demonstrated in [videos](https://projectwebsite7.github.io/gene-dex-manip/) and figures (e.g., case (a) and case (c) in Figure 3 and 4, first and fifth cases in the videos on the [website](https://projectwebsite7.github.io/gene-dex-manip/)), our method effectively handles such issues, including penetrations, jittering hand states, and unreachable object configurations. We do not assume that the tracking of the next frame depends on the precise tracking of previous states.
>
> - **Source of kinematic references:** We are addressing the "tracking" problem, where kinematic references are assumed to be available from sources such as MoCap datasets, video reconstructions, *etc.*, as mentioned in our response to the previous question. As vision techniques continue to advance, the variety and abundance of trajectory sources will increase, further expanding the potential of our tracking control paradigm for learning general dexterous manipulation skills.

---

> ### Author Response · Authors · 2024-11-22
> **Author Response (Part 5 of 7)**
>
> > "The pipeline extends beyond a trained point cloud module and an RL-trained goal-conditioned policy."
> >
> > "...work that could all contribute to weakness in generalizations..."
> >
> > "...are there guarantees that the paths generated by the diffusion model constitute homotopy..."
> >
> > "I am also concerned with the terminologies, such as "homotopy optimization scheme" and "parent task."...None of these are rigorously grounded in theory or computational terms and are marred by overuse of human intuitions in writing"
> >
> > "... how the homotopy optimization they used is similar to a "chain of thought" in a large language model..."
> >
> > "...using these intuition-driven terms to report scientific..."
>
> - **"...extends beyond a trained point cloud module and an RL-trained goal-conditioned policy...":** Thank you for your effort in summarizing our method pipeline. However, we would like to clarify that the core of our method lies not in the point cloud processing module or the goal-conditioned policy. As previously discussed, the essence of our approach is twofold: 1) Leveraging high-quality demonstrations to enhance the tracking controller's learning, and 2) Utilizing a data flywheel that continuously improves the quality and diversity of the tracking demonstrations, which are critical for training the tracking controller. The data mining aspect plays a pivotal role in refining the tracking demonstrations and is not merely an "extension".
>
> - **Homotopy path generator:** We use this module to efficiently propose effective homotopy paths. While the learning-based module cannot guarantee perfect generalization, it is employed to mine tracking demonstrations from the training dataset, rather than for generalization purposes in out-of-domain test settings.
>
>   **...are there guarantees... constitute homotopy...?** While we cannot theoretically guarantee that the proposed paths constitute a true homotopy, there is a high likelihood that a well-trained model can generate effective homotopy paths. Using this model to enhance single trajectory tracking is reasonable, as we *do not require it to be effective every time*.
>
>   For tracking a specific trajectory, the original tracking results are provided by the basic RL-based single trajectory tracking policy. After proposing an optimization path using the path generator and optimizing through it, we obtain new tracking results for the task. ***We only adopt the new tracking results when they outperform the original results produced by the basic tracking policy.***
>
>   *At test time*, only the final tracking controller is used. As illustrated in the inference flow of the teaser figure, homotopy optimization is not involved during the inference phase.
>
> - **Terminologies' definitions:** In section 3.2, line 299, we define the "homotopy optimization": "...*the homotopy optimization iteratively solves each tracking task in an optimization path...*". In line 311-312, we define the "parent task": "*we consider...a neighbor that provides a better baseline trajectory...an effective 'parent' task.*" In line 314-315, we define the "effective optimization path".   We've already clarified their definitions.
>
>   - **Homotopy optimization:** "Homotopy optimization" is a traditional approach for tackling complex optimization problems by solving a sequence of progressively harder surrogate subproblems [NumericalM, StabilityAna, HomotopyAna]. The key idea is to find an effective "path" that maps an easier problem to the more difficult one, allowing the solver to begin with the simplest problem, progressively address intermediate challenges, and ultimately solve the final, complex problem.
>
>     In our work, we apply "homotopy optimization" to address the challenging task of single trajectory tracking by starting with a simpler tracking problem. We then iteratively solve each problem along the optimization path, ultimately solving the original, more difficult problem. The meaning of "homotopy optimization" and "homotopy" in our work is consistent with their use in the optimization literature [NumericalM, StabilityAna, HomotopyAna].
>
>   - **Parent task:** We have defined the concept of the "parent" task on pages 311-212. This definition is sufficient to convey its meaning. However, if the reviewer feels it necessary, we can provide a more formal "computational" interpretation using mathematical language and include it in the revised manuscript.

---

> ### Author Response · Authors · 2024-11-22
> **Author Response (Part 6 of 7)**
>
> - **The reason for using intuition-driven terms:** We intentionally use intuition-driven terms to convey the meaning of some concepts in a way that is accessible to readers who may not be familiar with robotics or optimization. For example, readers with a deep learning background who are unfamiliar with "homotopy optimization" might more easily grasp its intuition through an analogy with "chain-of-thought," a widely recognized concept in foundation models. This approach helps bridge the gap between disciplines and makes the underlying ideas more relatable and understandable to a broader audience.
>
>   **"...intuition-driven terms to report scientific...":** A well-crafted scientific report should both have rigorous definitions for each newly introduced concept and offer intuitive explanations that can help readers grasp the meaning of these concepts, even for people who do not have the right background. This was the intention behind using the analogy of "chain-of-thought" to explain "homotopy optimization".
>
>   Using intuitive analogies does not imply that we lack rigorous definitions. On the contrary, our goal is to complement the formal definitions with accessible explanations that aid understanding, particularly for readers who may not have the relevant background.

---

> ### Author Response · Authors · 2024-11-22
> **Author Response (Part 7 of 7)**
>
> ## Professionalism and credibility
>
> We greatly appreciate the reviewer’s high standards regarding the professionalism of a scientific work's presentation. We would like to clarify that for newly introduced concepts, such as "parent task" and "effective homotopy path", we have rigorously defined them prior to or at their first usage in the manuscript. The introduction of intuitive terms aims to assist readers from diverse backgrounds in better understanding the meaning and context of specific concepts.
>
> Additionally, we wish to highlight that terms like "robustness", "adaptability", and "generalization ability" are indeed widely utilized across numerous published works in the robot learning community, including but not limited to [VMP, BiLoco, HORA, UniDexGrasp, OmniGrasp, GeneRot, PHC, DTC]. In our work, we have provided computational definitions for these properties in Appendix C, in the context of a tracking controller.
>
> Moreover, "generalization ability" is a fundamental concept in machine learning, with rigorous mathematical definitions rooted in learning theory [Gene1, Gene2, Gene3, Gene4]. This term is extensively employed not only within robot learning but across the broader deep learning community.
>
>
>
>
>
> Finally, thank you again for your time and the follow-up response. We are more than willing to address any further questions and concerns. Please feel free to let us know if there is anything specific we can provide or clarify.
>
> Best regards,
>
> Authors
>
>
>
>
>
> *[VMP] Serifi, Agon et al. “VMP: Versatile Motion Priors for Robustly Tracking Motion on Physical Characters.” Computer Graphics Forum (2024): n. pag.*
>
> *[BiLoco] Li, Zhongyu et al. “Reinforcement Learning for Versatile, Dynamic, and Robust Bipedal Locomotion Control.” ArXiv abs/2401.16889 (2024): n. pag.*
>
> *[HORA] Qi, Haozhi et al. “In-Hand Object Rotation via Rapid Motor Adaptation.” Conference on Robot Learning (2022).*
>
> *[Gene1] Brennan, R. L. (2001). Generalizability Theory. New York: Springer-Verlag.*
>
> *[Gene2] Crocker, L., & Algina, J. (1986). Introduction to Classical and Modern Test Theory. New York: Harcourt Brace.*
>
> *[Gene3] Shavelson, R.J., & Webb, N.M. (1991). Generalizability Theory: A Primer. Thousand Oaks, CA: Sage.*
>
> *[Gene4] Shalev-Shwartz, Shai and Shai Ben-David. “Understanding Machine Learning - From Theory to Algorithms.” (2014).*
>
> *[UniDexGrasp] Xu, Yinzhen et al. “UniDexGrasp: Universal Robotic Dexterous Grasping via Learning Diverse Proposal Generation and Goal-Conditioned Policy.” 2023 IEEE/CVF Conference on Computer Vision and Pattern Recognition (CVPR) (2023): 4737-4746.*
>
> *[UniDexGrasp++] Geng, Haoran and Yun Liu. “UniDexGrasp++: Improving Dexterous Grasping Policy Learning via Geometry-aware Curriculum and Iterative Generalist-Specialist Learning.” 2023 IEEE/CVF International Conference on Computer Vision (ICCV) (2023): 3868-3879.*
>
> *[OmniGrasp] Luo, Zhengyi et al. “Grasping Diverse Objects with Simulated Humanoids.” ArXiv abs/2407.11385 (2024): n. pag.*
>
> *[QuasiSim] Liu, Xueyi et al. “QuasiSim: Parameterized Quasi-Physical Simulators for Dexterous Manipulations Transfer.” ArXiv abs/2404.07988 (2024): n. pag.*
>
> *[QuasiStatic] Pang, Tao et al. “Global Planning for Contact-Rich Manipulation via Local Smoothing of Quasi-Dynamic Contact Models.” IEEE Transactions on Robotics 39 (2022): 4691-4711.*
>
> *[GeneRot] Chen, Tao et al. “A System for General In-Hand Object Re-Orientation.” ArXiv abs/2111.03043 (2021): n. pag.*
>
> *[PHC] Luo, Zhengyi et al. “Perpetual Humanoid Control for Real-time Simulated Avatars.” 2023 IEEE/CVF International Conference on Computer Vision (ICCV) (2023): 10861-10870.*
>
> *[DTC] Jenelten, Fabian et al. “DTC: Deep Tracking Control.” Science Robotics 9 (2023): n. pag.*
>
> *[PULSE] Luo, Zhengyi et al. “Universal Humanoid Motion Representations for Physics-Based Control.” ArXiv abs/2310.04582 (2023): n. pag.*
>
> *[DeepMImic] Peng, Xue Bin et al. “DeepMimic.” ACM Transactions on Graphics (TOG) 37 (2018): 1 - 14.*
>
> *[MCC-HO] Wu, Jane et al. “Reconstructing Hand-Held Objects in 3D.” ArXiv abs/2404.06507 (2024): n. pag.*
>
> *[ManiDext] Zhang, Jiajun et al. “ManiDext: Hand-Object Manipulation Synthesis via Continuous Correspondence Embeddings and Residual-Guided Diffusion.” ArXiv abs/2409.09300 (2024): n. pag.*
>
> *[DexHand] Jiang, Haiyan et al. “DexHand: dexterous hand manipulation motion synthesis for virtual reality.” Virtual Reality 27 (2023): 2341 - 2356.*
>
> *[NumericalM] Allgower, Eugene L. and Kurt Georg. “Numerical continuation methods - an introduction.” Springer Series in Computational Mathematics (1990).*
>
> *[StabilityAna] Seydel, Rüdiger. “Practical Bifurcation and Stability Analysis.” (1994).*
>
> *[HomotopyAna] Liao, Shijun. “Homotopy Analysis Method in Nonlinear Differential Equations.” (2012).*

---

> > ### Author Response · Authors · 2024-12-02
> > **Thank you for your valuable feedback and look forward to future responses**
> >
> > Dear Reviewer CSzf,
> >
> > We would like to kindly remind you that tomorrow is **the last day** of the discussion period. If there are any remaining questions or points you would like us to address, please feel free to let us know. We are fully available to discuss any aspects further.
> >
> > Thank you again for your valuable feedback and follow-up questions.
> >
> > Best regards, Authors

---

### Official Review · Reviewer_4aCE · 2024-11-05

**Soundness:** 3
**Presentation:** 3
**Contribution:** 3
**Rating:** 8
**Confidence:** 4

**Summary:**

This paper proposes an approach for training generalizable neural tracking controllers from human reference trajectories for dexterous manipulation tasks with robot hands. The approach consists of three steps: The first step involves using reinforcement learning to successfully track single, per-task trajectories for a subset of tasks. These policies are then distilled into a single policy using imitation learning. Next, a dataset of trajectories is sampled from the remaining tasks. The trained IL policy is now used to generate reference policies and residual RL is used to improve per-trajectory tracking. The best tracking results are used to re-train the IL policy. To enhance the diversity of trajectories tracked, a homotopy optimization scheme is introduced to facilitate “chain-of-thought” learning of the tracking controller over the set of related tasks. Once a sufficient set of high-quality tracking trajectories is generated, a conditional diffusion model is trained to serve as a homotopy generator. Finally, RL, the trained tracking and the trained homotopy generator are used to curate a final set of demonstrations and optimize the tracking controller.

**Strengths:**

- Meticulous and concise description of the details of the proposed algorithm.
- Tackles an important problem of learning a tracking controller for dexterous manipulation that is often overlooked, but could have significant impact as our data collection devices continue to improve with the rapid rise of AR/VR.
- Thorough evaluations of the proposed method in simulation as well as real-world setup.
- Paper provides useful insights into the importance of high-quality demonstration data for dexterous manipulation, and outlines a procedure for curating it from existing datasets.

**Weaknesses:**

- Approach relies on the presence of object states in the dataset which are difficult to obtain and are limited to very few datasets. It could be useful to investigate the effectiveness of this dataset without using object states, especially given the relatively small performance gap with the `(w/o data)` ablation. Based on the current results, it is unclear to me that the use of object state data, which is especially hard to come by in real-world datasets is significantly useful to the proposed approach.
- In the same vein, it might be insightful to present results with the `Ours (w/o data)` ablation in the real-world experiments to better understand the effectiveness of data when crossing the significant domain gap between simulation and real-world.

**Questions:**

- How does a simple open loop baseline perform? I would be interested in the authors’ experience with sampling a reference trajectory from the dataset based on a simple heuristic to determine a nearest neighbor trajectory.
- With human-object interaction, the trajectory of the object is generally the main variable of interest. Given the significant morphology gap between human hands and the robot hands used in this work, the error in position and orientation of the wrist as well as the robot hand fingertips can be a noisy signal resulting from artifacts in the trajectory retargeting scheme. I would like the authors to reflect on the relative weighting schemes attempted and any related conclusions that informed their design of the reward function as well as metrics.

---

> ### Author Response · Authors · 2024-11-19
> **Author Response (Part 1 of 5)**
>
> Dear Reviewer 4aCE,
>
>
>
> Thank you so much for your thorough and constructive review. We do appreciate your recognition of the importance of our problem, valuable insights, and our thorough evaluations conducted in the simulation and the real world. Below, we address your specific questions in the hope that our responses adequately address any of your concerns.
>
>
>
> > **"Approach relies...object states...difficult to obtain and are limited to very few datasets...effectiveness of this dataset without using object states...relatively small performance gap with the (w/o data) ablation. ...unclear to me that the use of object state data...hard to come by in real-world datasets is significantly useful to the proposed approach."**
>
> - **Object states are required in the tracking problem** since per-frame object states serve as **goals** that tell the policy "what to track" in the next timestep. Such per-frame goal states are assumed available in the tracking problem [Omnigrasp, BiLoco, HPC]. In each timestep, the policy observes the current hand and object states, as well as the goal, including the hand state and the **object state** *expected* to be achieved in the next timestep, and outputs the action. We've added a paragraph at the beginning of the method section in the revised version, where we explain the "tracking" problem from a high level.
> - **Object goal states usually cannot be omitted** in the policy's observation space. If we directly remove the guidance from the object state in the training, the object may fall out of hand but the policy can still get high rewards.
> - **Developing a tracking policy without the need for object goal states** is an interesting and valuable future research direction. Possible solutions are **1**) replacing state goals with vision goals (*e.g.,* specified via images), **2)** replacing the state goal guidance with high-level guidance such as text, and **3)** incorporating a forecasting model that can predict future object states to track on-the-fly under some high-level guidance.
> - **Source of manipulation trajectories with object states:** Considering the current policy formulation which requires the hand and object states to specify goals, our kinematic reference trajectories come from existing MoCap datasets. However, as the 3D vision domain continues to develop, many effective image/video-based object state estimation tools such as [MCC]  have emerged. Utilizing these tools to lift object state sequences from images or videos is a promising strategy to expand our tracking trajectories from existing MoCap datasets to *world-wild datasets*. As these techniques continue to develop, tracking control will become a more promising paradigm to develop a general dexterous manipulation policy.
> - **"...the use of object state data..."**: As explained above, object states serve as "goals" that are encoded as part of the observation passed to the RL module. We *do not use object states to provide supervision.*  It is different from the "tracking demonstration" data we use in imitation learning. We use ***expert robot hand action trajectories*** in tracking demonstrations to supervise actions predicted by the tracking controller.
> - Besides, **the (w/o data) ablation** denotes the model trained with demonstrations that are *only improved by the homotopy optimization scheme*, without trying to transfer tracking data prior from the tracking controller for improvement, as detailed in the ***first paragraph in section 5***. We suppose that the reviewer might regard this ablation as the model trained without supervision from robot tracking demonstrations (this model is denoted as ***PPO (w/o sup., tracking rew.)***, whose performance lags behind the model trained with demonstration data, evidencing the importance of utilizing demonstration data to train the tracking controller together with RL). If so, we are sorry, and we've updated the caption of Table 1 to avoid confusion.
>
>
>
> *[OmniGrasp] Luo, Zhengyi et al. “Grasping Diverse Objects with Simulated Humanoids.” ArXiv abs/2407.11385 (2024): n. pag.*
>
> *[BiLoco] Li, Zhongyu et al. “Reinforcement Learning for Versatile, Dynamic, and Robust Bipedal Locomotion Control.” ArXiv abs/2401.16889 (2024): n. pag.*
>
> *[PHC] Luo, Zhengyi et al. “Perpetual Humanoid Control for Real-time Simulated Avatars.” 2023 IEEE/CVF International Conference on Computer Vision (ICCV) (2023): 10861-10870.*
>
> *[MCC] Wu, Chaozheng et al. “Multiview Compressive Coding for 3D Reconstruction.” 2023 IEEE/CVF Conference on Computer Vision and Pattern Recognition (CVPR) (2023): 9065-9075.*

---

> ### Author Response · Authors · 2024-11-19
> **Author Response (Part 2 of 5)**
>
> > "**...insightful to present results with the Ours (w/o data) ablation in the real-world experiments to better understand the effectiveness of data...**"
>
> - As explained in the response to the previous question, the *Ours (w/o data)* ablation denotes the model trained with demonstrations that are improved by the homotopy optimization scheme only, without transferring tracking data prior to the tracking controller.
> - We suppose that the reviewer might regard ***Ours (w/o data)*** as the model trained without supervision from expert robot tracking demonstrations. If so, this model is exactly the best baseline we've compared with, which is denoted as ***PPO (w/o sup., tracking rew.)***. And it is also exactly the same baseline that **we've compared in our real-world experiments**. Please refer to Table 2,5,6, Figure 4, and videos on the [website](https://projectwebsite7.github.io/gene-dex-manip/) for details.
> - We've updated the caption of Table 1 (*highlighted in blue*) to include descriptions for ablated models.

---

> ### Author Response · Authors · 2024-11-19
> **Author Response (Part 3 of 5)**
>
> > **"How does a simple open loop baseline perform? I would be interested in the authors’ experience sampling a reference trajectory from the dataset based on a simple heuristic to determine the nearest neighbor trajectory."**
>
> Thanks for suggesting this baseline. That's quite an interesting baseline setup. It would be promising if we have a large number of expert demonstration data with high-quality action sequences so that we can form a large and relatively continuous expert action trajectory space. However, it might not be effective given the limited amount of our current demonstration data.
>
> In the following text, we'll describe the open-loop baseline setting, followed by presenting its results. In case we might misinterpret the "open loop baseline" you've described, please let us know if there is any misunderstanding. And we'll conduct further experiments accordingly.
>
> - **Open-loop baseline setting:**  In the evaluation scenario, for each test tracking task, sample a kinematic reference trajectory from the demonstration set that is the closest to the kinematic reference trajectory of the test tracking task. Directly apply the action sequence of the sampled kinematic reference trajectory to solve the test tracking task.
>
> - **Our experiments:** We test its performance on the GRAB test set.
>
>   - *Calculating the difference between two tracking tasks:* For a trajectory tracking task described by the kinematic hand state sequence, the kinematic object pose sequence, and the object geometry, e.g., represented as the point cloud, we calculate the trajectory tracking task difference between two tracking tasks as a weighted sum of the hand trajectory difference, object pose sequence difference and the object geometry difference (Please refer to Appendix C for details, as we frequently encounter "math errors" when using the OpenReview text editor).
>
>   - *Finding the nearest neighbour:* Since both the test dataset and the demonstration set only contain several hundreds tracking tasks (please see *Appendix C* for detaisl), we each test task, we find its nearest neighbour from the demonstration set through a brute-force manner. We compute the difference between the test task and all tracking tasks in the demonstration set, followed by picking the one with the smallest difference value.
>
>   - *Rollout the action trajectory:* For each test task $\mathbf{T}_A$ and its nearest neighbour found from the demonstration set $\mathbf{T}_A^{nei}$, we set the initial state of the hand to the initial state of $\mathbf{T}_A^{nei}$. It is a crucial point to avoid the hand from just "flying away" when the initial state of the test task $\mathbf{T}_A$ deviates largely from the first positional commands of $\mathbf{T}_A^{nei}$'s action trajectory. After that, we directly roll out the action sequence of $\mathbf{T}_A^{nei}$, trying to solve the tracking task $\mathbf{T}_A$.
>
> - **Results:** We summarize the performance achieved by this baseline and that of our method in the following table:
>
>   |           | $R_{\text{err}}$ ($\text{rad}, \downarrow$) | $T_{\text{err}}$ (${cm}, \downarrow$) | $E_{\text{wrist}}$ ($ \downarrow$) | $E_{\text{finger}}$ ($\text{rad}, \downarrow$) | Success Rate ($\%, \uparrow$) |
>   | --------- | ------------------------------------------- | ------------------------------------- | ---------------------------------- | ---------------------------------------------- | ----------------------------- |
>   | Open-loop | 0.9357                                      | 29.17                                 | 0.4524                             | 0.6003                                         | 10.15/14.21                   |
>   | Ours      | **0.3303**                                  | **4.53**                              | **0.1118**                         | **0.5048**                                     | **46.70/65.48**               |
>
>   As shown in the above table, the open-loop baseline performs very badly *(the worst one* among all the baselines we've compared). We suppose the reason is the gap between the test tracking task set and tracking tasks in the demonstration set.  Directly using the action sequence from task $\mathbf{T}_A^{nei}$ can hardly solve the tracking problem $\mathbf{T}_A$.

---

> ### Author Response · Authors · 2024-11-19
> **Author Response (Part 4 of 5)**
>
> > **"...the error in position and orientation of the wrist as well as the robot hand fingertips can be a noisy signal...reflect on the relative weighting schemes attempted and any related conclusions that informed their design of the reward function as well as metrics."**
>
> - **Reasons for introducing hand tracking metrics and rewards:** Thank you for your detailed and insightful question. Due to the morphological differences between the human hand and the robot hand, the retargeted robot hand trajectory may exhibit artifacts and noise. Consequently, hand-tracking errors or accuracies are not inherently reliable for designing metrics or rewards. However, the reason **we still include hand-tracking metrics and use them to design rewards** lies in the broader objective of manipulation tracking. While the primary goal is for the robot to manipulate objects and achieve their desired state sequences, we also aim for human-like robot hand behavior in the process.
>
> - **Reward design:** We design the reward to encourage accurate object tracking and hand tracking as outlined in *Eq.3, Section 3.1* and *Eq. 11-14, Appendix A*.  In our experiments, weights related to hand state and object pose tracking in the reward function $r=w_{o,p}r_{o,p}+w_{o,q}r_{o,q}+w_{\text{wrist}}(-(w_{\text{trans}}d_{\text{trans}}+w_{\text{ornt}}d_{\text{ornt}}))+w_{\text{finger}}r_{\text{finger}}+w_{\text{affinity}}r_{\text{affinity}}$, where $d_{\text{trans}}$ and $d_{\text{ornt}}$ are distances from the current state to the desired goal state to track respectively (details w.r.t. such two distances can be found in Appendix A), are set as follows:
>
>   |        | $w_{o,p}$ | $w_{o,q}$ | $w_{\text{wrist}}\cdot w_{\text{trans}}$ | $w_{\text{wrist}}\cdot w_{\text{ornt}}$ | $w_{\text{finger}}$ |
>   | ------ | --------- | --------- | ---------------------------------------- | --------------------------------------- | ------------------- |
>   | Weight | 1.0       | 0.33      | 0.3                                      | 0.05                                    | 0.05                |
>
>   **Experiences and observations in the balancing reward weights**:
>
>   - In our implementation, we assign a relatively large reward coefficient to object tracking, encompassing both position and orientation, compared to the coefficients for hand tracking. This aligns with the primary goal, as you mentioned, of accurately tracking object states. Specifically, we set a significantly higher reward coefficient for object position tracking to prioritize lifting the object off the table. In our practice, setting $w_{o,p}$ larger than $w_{o,q}$ is quite important to encourage correct early-stage manipulation involving mainly approaching and lifting. Another reason is that the object orientation error is measured by radian, whose absolute number tends to be larger than the position error measured via Euclidean distance (the unit is meters).
>   - For hand tracking rewards, we encourage tracking the wrist translations more than tracking the wrist orientation and fingers. This is because wrist translation tracking is important to encourage the hand to **reach** the object in the initial manipulation stage. In the following stage, encouraging wrist translation tracking can also help with tracking object positions.
>   - Even though the retargeted robot hand trajectory is clean, object tracking is still a vital point for manipulation. Otherwise, in some cases, the policy may get stuck into the local optimum by only tracking hands leaving the object not even being lifted up.
>
>   We have added the above table in *Appendix A.2*.
>
>   **Limitations of the current reward design:** Besides, we use a single suit of reward coefficients in the whole tracking process. However, we may need to balance different in different stages. For instance, in the early stage, the most important thing is picking up the object. In the following manipulation stage, tracking object orientation would become more crucial. That's a limitation in our current reward design. How to adapt it to a multi-stage reward is an interesting future direction.

---

> ### Author Response · Authors · 2024-11-19
> **Author Response (Part 5 of 5)**
>
> - **Metrics design:** As detailed in the *Metrics* paragraph in *Section 4.1*, we introduce five types of metrics for measuring hand and object tracking performance individually and to evaluate the overall task success taking both hand and object tracking results into consideration:  **1)** object rotation error $R_{\text{err}}$, **2)** object translation tracking error $T_{\text{err}}$, **3)** hand wrist tracking error $E_{\text{wrist}}$, **4)** hand finger tracking error $E_{\text{finger}}$, and 5) success rates:
>   - For object orientation and position tracking errors, as well as hand finger tracking errors, it is unnecessary to assign different weights to the individual terms.
>   - **Weighting hand wrist tracking error:** For hand wrist tracking error, we assign equal weights to the wrist translation error and the wrist orientation error so that the hand wrist tracking metric gives equal consideration to both translation and orientation performance. However, it is worth noting that in the reward design, we prioritize translation tracking.
>   - **Weighting in sucess rates:** For success rates, we consider a tracking task successful if both the object tracking error and the hand tracking error fall below specific thresholds. Additionally, we calculate a holistic hand tracking metric by averaging the wrist tracking error and the finger tracking error. The ***hand tracking threshold is set to a value that is less strict than those for object rotation and orientation***.
>
>
>
> Finally, thank you again for your detailed constructive review. We would love to address any further questions you may have and are committed to resolving any concerns you've raised. Please feel free to let us know if there is additional information we can provide.
>
>
>
> Looking forward to your responses!
>
> Best regards, Authors

---

> ### Author Response · Authors · 2024-11-23
> **Thank you for your constructive review and look forward to post-rebuttal feedback**
>
> Dear Reviewer 4aCE,
>
> The weekend is coming or already here! Wish you a nice, pleasant, and relaxing weekend!
>
> We'd like to express our sincere gratitude for your detailed and careful reviews, insightful questions, and constructive suggestions.
>
> We have provided responses to all of your questions and concerns and hope they sufficiently address them.
>
> As the discussion phase will conclude on November 26th, please don’t hesitate to let us know if you have any further questions. We are committed to making every effort to resolve any remaining concerns you may have and deeply appreciate your valuable suggestions.
>
> Thank you for your time and constructive reviews! Sincerely look forward to hearing from you.
>
> Best regards,
>
> Authors

---

> > ### Author Response · Authors · 2024-12-02
> > **Thank you for your valuable feedback and look forward to post-rebuttal responses**
> >
> > Dear Reviewer 4aCE,
> >
> > We would like to kindly remind you that tomorrow is **the last day** of the discussion period. If there are any remaining questions or points you would like us to address, please feel free to let us know. We are fully available to discuss any aspects further.
> >
> > Thank you again for your valuable feedback.
> >
> > Best regards, Authors

---

### Author Response · Authors · 2024-11-19
**General Responses and Revision Summary**

Dear All Reviewers:

We sincerely thank all the reviewers for their constructive feedback and valuable feedback. We are especially grateful for the recognition of the importance of our problem (*Reviewer 4aCE*), the significance of our achievements (*Reviewer gJWM*), our solid contributions (*Reviewer 4aMi*), the originality and design rationale of our method (*Reviewers gJWM, 4aMi*), as well as the solid (*Reviewer gJWM*) and thorough (*Reviewers 4aCE, gJWM, 4aMi*) experimental evaluations.

We appreciate your thoughtful questions and concerns, and we have tried our best to address them comprehensively in our individual responses.

We have carefully revised the paper to reflect our responses to reviewers' concerns and suggestions (highlighted in blue). Key revisions are:

- Improve the *teaser* figure and *method* figure.
- Add a paragraph at the beginning of the *method* section to clarify terminology and notations; and improve the section’s overview paragraph.
- Improve the caption of *Table 1* to describe ablated versions.
- Add a detailed method figure, additional results for ablation studies on demonstration scaling, failure cases from real-world experiments, generalizability analysis of the homotopy path generator, and computational interpretations of key concepts with corresponding quantitative evaluations to the *Appendix*.

We are truly grateful once again for your feedback. Please refer to our individual responses for our answers to specific questions and concerns. Please let us know if you have any further questions or comments. We would be delighted to discuss them and will spare no effort in addressing them.

Thank you,
Authors

---

### Author Response · Authors · 2024-11-29
**Thank you for all your insightful comments and look forward to post-rebuttal feedbacks**

Dear All Reviewers:

A new weekend is coming or already here! Wish you a nice, pleasant, and relaxing weekend!

We'd like to express our sincere gratitude again for your detailed reviews and valuable suggestions.

We have provided responses to all of your questions and have incorporated your suggestions in the revised paper (highlighted in blue). We hope our explanations and revisions could address your concerns.

As the discussion phase will conclude on December 2nd, please don’t hesitate to let us know if you have any further questions. We are committed to exerting every effort to address any remaining concerns you may have. We appreciate your suggestions.

Thank you for your time and constructive reviews! Sincerely look forward to your responses.

Best regards,

Authors

---

### Meta-Review · Area_Chair_6P8C · 2024-12-26

**Metareview:**

The paper introduces DexTrack, a neural tracking controller for dexterous manipulation, trained using large-scale curated demonstrations and enhanced with reinforcement learning, imitation learning, and homotopy optimization. The method achieves over a 10% improvement in success rates compared to baselines in both simulation and real-world evaluations.

The reviewers generally recognized the importance of the work, highlighting (1) the significance of the problem addressed, (2) the intuitive design of the method, (3) comprehensive experiments, and (4) its potential to leverage internet-scale videos for developing dexterous robotic policies. However, notable concerns were raised regarding the paper's presentation, its contextualization within the broader community, and the inconsistent and unclear use of technical terminology, which led to confusion in assessing the work's contributions.

After reviewing the reviews, rebuttal, and discussions, the AC has mixed feelings. While the paper’s presentation needs significant improvement, the contributions and experimental results provide a solid demonstration of the method's effectiveness. The AC recommends accepting the paper but strongly encourages the authors to carefully address all detailed comments from reviewers, ensuring proper articulation of the contributions, clearer positioning within related literature, and refinement of technical terminology in a future revision.

**Additional Comments On Reviewer Discussion:**

The Reviewer Discussion phase involved extensive debate. Reviewer CSzf raised concerns regarding (1) the technical soundness and imprecise use of mathematical terms, (2) the robustness of the method in handling implausible references, and (3) the limited demonstration of the method’s generalization capabilities. Reviewer 4aCE actively engaged in the discussion, helping to clarify the method’s contributions, which led Reviewer CSzf to raise their score to 5.

The AC believes Reviewer 4aCE’s comments provide valuable insights for improving the paper’s clarity and presentation. The AC has included these comments for the authors’ reference and urges them to carefully incorporate the suggested improvements to enhance the quality of the paper.

> Dear AC, reviewer CSzf and other reviewers,

> Thank you for taking the time to evaluate this work and for providing valuable feedback. I agree with Reviewer CSzf's comments regarding the quality of the writing and presentation, and it did take me some time to fully grasp the authors' point. However, I would still like to emphasize why the concept of a neural tracking controller is both significant and deserving of attention within the robot learning community.

> A key point often overlooked is the non-trivial nature of trajectory retargeting from humans to robots due to the morphological differences between human and robot hands. Even with highly accurate kinematic reference trajectories, this remains a challenging problem. For example, consider a scenario where a human uses high-fidelity motion capture gloves to generate precise trajectories for picking up a tennis ball on a table. Directly mapping these trajectories to a robot hand using heuristic retargeting functions often leads to failure in most cases. A neural tracking controller, however, learns this retargeting function, bridging the morphological gap.

> At this point, it is useful to draw a distinction from imitation learning works like ALOHA and diffusion policy, which rely on teleoperated demonstrations. Teleoperation involves a person watching the robot, providing online, corrective actions to achieve the task, and compensating for inaccuracies in retargeting functions during task execution. The neural tracking controller setting, on the other hand, mirrors the earlier tennis ball example: you have high-precision data of humans executing the task but the robot's ability to replicate the motion is uncertain without a reliable retargeting mechanism.

> Most imitation learning algorithms require robot state-action pairs for expert policy estimation. The value of a neural tracking controller lies in its ability to reliably map human state-action pairs to robot state-action pairs, enabling imitation learning algorithms like ALOHA and Diffusion Policy to leverage human-generated data. This is particularly compelling given the relative ease and accessibility of collecting human data compared to teleoperated robot demonstrations. At its most ambitious, a neural tracking controller could potentially utilize human state-action pairs derived from internet-scale videos to develop dexterous robotic policies. While this work does not yet achieve that vision, it addresses an important problem with substantial potential.

> Lastly, I believe the paper's focus on scenarios with access to high-quality kinematic reference trajectories justifies not addressing cases involving implausible trajectories, as they fall outside the intended scope.

> All of that being said, I do believe that some of the authors' defense of their work as well as their presentation in the paper misses essential emphasis on the relevance and contextualization of their work. Whether that is sufficient grounds for rejection, I would leave to the AC's judgement.

> Best, Reviewer 4aCE

---

### Decision · Program_Chairs · 2025-01-22

Accept (Poster)